# Efficient Last-Iterate Convergence in Solving Extensive-Form Games

**Linjian Meng**[1], **Tianpei Yang**[1†], **Youzhi Zhang**[2†], **Zhenxing Ge**[1], **Shangdong Yang**[3],
**Tianyu Ding**[4], **Wenbin Li**[1], **Bo An**[5], **Yang Gao**[1]

[1] National Key Laboratory for Novel Software Technology, Nanjing University
[2] Centre for Artificial Intelligence and Robotics, Hong Kong Institute of Science & Innovation, CAS
[3] Jiangsu Key Laboratory of Big Data Security and Intelligent Processing, Nanjing
University of Posts and Telecommunications
[4] Microsoft Corporation
[5] School of Computer Science and Engineering, Nanyang Technological University
menglinjian@smail.nju.edu.cn, tianpei.yang@nju.edu.cn, youzhi.zhang@cair-cas.org.hk,
zhenxingge@smail.nju.edu.cn, sdyang@njupt.edu.cn, tianyuding@microsoft.com,
liwenbin@nju.edu.cn, boan@ntu.edu.sg, gaoy@nju.edu.cn

## Abstract

To establish last-iterate convergence for Counterfactual Regret Minimization (CFR) algorithms in learning a Nash equilibrium (NE) of extensive-form games (EFGs), recent studies reformulate learning an NE of the original EFG as learning the NEs of a sequence of (perturbed) regularized EFGs. Hence, proving last-iterate convergence in solving the original EFG reduces to proving last-iterate convergence in solving (perturbed) regularized EFGs. However, these studies only establish last-iterate convergence for Online Mirror Descent (OMD)-based CFR algorithms instead of Regret Matching (RM)-based CFR algorithms in solving perturbed regularized EFGs, resulting in a poor empirical convergence rate, as RM-based CFR algorithms typically outperform OMD-based CFR algorithms. In addition, as solving multiple perturbed regularized EFGs is required, fine-tuning across multiple perturbed regularized EFGs is infeasible, making parameter-free algorithms highly desirable. This paper show that $CFR^+$, a classical parameter-free RM-based CFR algorithm, achieves last-iterate convergence in learning an NE of perturbed regularized EFGs. This is the first parameter-free last-iterate convergence for RM-based CFR algorithms in perturbed regularized EFGs. Leveraging $CFR^+$ to solve perturbed regularized EFGs, we get Reward Transformation $CFR^+$ ($RTCFR^+$). Importantly, we extend prior work on the parameter-free property of $CFR^+$, enhancing its stability, which is vital for the empirical convergence of $RTCFR^+$. Experiments show that $RTCFR^+$ exhibits a significantly faster empirical convergence rate than existing algorithms that achieve theoretical last-iterate convergence. Interestingly, $RTCFR^+$ show performance no worse than average-iterate convergence CFR algorithms. It is the first last-iterate convergence algorithm to achieve such performance. Our code is available at https://github.com/menglinjian/NeurIPS-2025-RTCFR.

## 1 Introduction

Extensive-form games (EFGs) are a foundational model for capturing interactions among multiple agents and sequential events, which are widely applied in simulating real-world scenarios, such as medical treatment [Sandholm, 2015], security games [Lisý et al., 2016], and recreational games [Brown and Sandholm, 2019b]. A common goal to address EFGs is to learn a Nash equilibrium (NE), where no player can unilaterally improve their payoff by deviating from the equilibrium.

---

† Corresponding authors.

39th Conference on Neural Information Processing Systems (NeurIPS 2025).

Recent research commonly employs regret minimization algorithms [Zhang et al., 2022b] to learn an NE in EFGs. Among them, Counterfactual Regret Minimization (CFR) algorithms are the most widely used ones for learning an NE in real-world EFGs [Bowling et al., 2015, Moravčík et al., 2017, Brown and Sandholm, 2018, 2019b, Pérolat et al., 2022]. They usually use Regret Matching (RM) algorithms [Hart and Mas-Colell, 2000, Gordon, 2006, Lanctot et al., 2009, Lanctot, 2013, Tammelin, 2014, Brown and Sandholm, 2019a, Farina et al., 2021, 2023, Xu et al., 2024b] as the local regularizer, since RM algorithms usually exhibit a faster empirical convergence rate than other local regret minimizers, such as Online Mirror Descent (OMD) [Nemirovskij and Yudin, 1983]. For convenience, we refer to the CFR algorithms that employ RM algorithms and OMD algorithms as local regularizers as RM-based CFR algorithms and OMD-based CFR algorithms, respectively.

However, most regret minimization algorithms, including CFR algorithms, typically only achieve average-iterate convergence and their strategy profile may diverge or cycle, even in normal-form games (NFGs) [Bailey and Piliouras, 2018, Mertikopoulos et al., 2018]. Average-iterate convergence implies that the averaging of strategies is necessary, which increases computational and memory overhead. Additionally, when strategies are parameterized via function approximation, a new approximation function must be trained to represent the average strategy, resulting in further approximation errors. Consequently, algorithms with last-iterate convergence to NE, which ensures that the sequence of strategy profiles converges to the set of NEs, are preferable.

To establish last-iterate convergence for CFR algorithms, recent studies [Pérolat et al., 2021, 2022, Liu et al., 2023] employ the Reward Transformation (RT) framework, which (i) transforms the task of learning an NE of the original EFG into learning the NEs of a sequence of (perturbed) regularized EFGs and (ii) ensures the sequence of the NEs of these (perturbed) regularized EFGs converges to the set of NEs of the original EFG. Therefore, to ensure last-iterate convergence in learning an NE of the original EFG, it is sufficient to establish last-iterate convergence in learning an NE of (perturbed) regularized EFGs. Unfortunately, these studies only establish last-iterate convergence in learning an NE of (perturbed) regularized EFGs for OMD-based CFR algorithms, incurring a poor empirical convergence rate to the set of NEs of the original EFG, as illustrated in our experiments.

To improve the empirical convergence rate, we propose Reward Transformation CFR$^+$ (RTCFR$^+$), utilizing CFR$^+$ [Tammelin, 2014], a classical parameter-free RM-based CFR algorithm, to solve perturbed regularized EFGs. RTCFR$^+$ is inspired by two observations: (i) RM-based CFR algorithms (the CFR algorithms that employ RM algorithms as the local regret minimizer) usually outperform OMD-based CFR algorithms, and (ii) parameter-free algorithms, implying no parameters need to be tuned [Grand-Clément and Kroer, 2021], are desirable to solve multiple perturbed regularized EFGs because fine-tuning across all perturbed regularized EFGs is infeasible. Notably, the parameter in CFR algorithms typically refers to the step sizes. Based on the RT framework, if CFR$^+$ has last-iterate convergence in learning an NE of perturbed regularized EFGs, then RTCFR$^+$ has last-iterate convergence in learning an NE of the original EFG. Unfortunately, it remains unknown whether CFR$^+$ achieves the parameter-free (i.e., holds for any step sizes) last-iterate convergence in learning an NE of perturbed regularized EFGs. It motivates a key question:

*Does CFR$^+$ have parameter-free last-iterate convergence*
*in learning an NE of perturbed regularized EFGs?*

To answer this question, we first provide the non-parameter-free (w.r.t. the step sizes) last-iterate convergence of CFR$^+$, i.e., for any initial accumulated counterfactual regrets, CFR$^+$ achieves last-iterate convergence in learning an NE of perturbed regularized EFGs when the step size exceeds a positive constant. We then extend this non-parameter-free result to establish the parameter-free result, i.e., CFR$^+$ achieves last-iterate convergence for any initial accumulated counterfactual regrets and step sizes. Note that our parameter-free result holds for any initial accumulated counterfactual regrets—not just the zero initialization in previous works [Farina et al., 2021][1]—enhancing the stability of CFR$^+$ [Farina et al., 2023], which is critical for the empirical convergence of RTCFR$^+$ in solving the original EFG. Without our parameter-free result, RTCFR$^+$ fails to empirically converge to the set of NEs of the original EFG! To the best of our knowledge, this is the first parameter-free last-iterate convergence guarantee for RM-based CFR algorithms in learning an NE of perturbed regularized EFGs. As a consequence, based on the convergences of the RT framework and CFR$^+$, RTCFR$^+$ achieves last-iterate convergence in learning an NE of the original EFG.

---

[1]While Tammelin et al. [2015] establish parameter-free average-iterate convergence of CFR$^+$ under any initialization, we show both last- and average-iterate convergence. Their proof techniques differ from ours and the recent RM-based CFR works, which are all based on Farina et al. [2021]. See details in Appendix B.

Specifically, we propose novel techniques to overcome the challenges in the above two steps of the proof. First, the primary challenge in proving the non-parameter-free result is that the smoothness of the instantaneous counterfactual regrets—the key property used in prior works [Liu et al., 2023] to establish the last-iterate convergence of CFR algorithms—cannot be leveraged, since RM algorithms update within the cone of the strategy space while the final output lies in the strategy space itself. To address this, we exploit the fact that an NE represents a best response to others at each infoset in perturbed EFGs. More specifically, this fact allows a term—related to the accumulated counterfactual regrets and the utility obtained by deviating from an NE of perturbed EFGs—can be added. It enables the smoothness of the instantaneous counterfactual regrets to be leveraged, ensuring that the cumulative squared distance between the iterated strategy profiles and the NE of perturbed regularized EFGs remains bounded by a constant across all iterations, thereby guaranteeing last-iterate convergence. Second, the main challenge of proving our parameter-free result is that the property used in prior proofs of the parameter-free property of $CFR^+$—the strategy sequence produced by $CFR^+$ remains invariant across different step sizes—holds only when the initial accumulated counterfactual regrets are zero [Farina et al., 2021]. We address this by leveraging the linearity of the projection alongside our non-parameter-free convergence result that holds for any initial accumulated counterfactual regrets. In particular, we use the linearity of projection to show that for any given initial accumulated counterfactual regrets and step sizes, there exists an alternative choice of these parameters that yields an identical strategy profile sequence. By then applying our non-parameter-free result to this alternative setting, we establish that the resulting strategy profile sequence converges to the set of NEs of perturbed regularized EFGs, thus proving the parameter-free last-iterate convergence. Notably, We only provide parameter-free last-iterate convergence results for $CFR^+$. In other words, $RTCFR^+$ is not a parameter-free algorithm.

Experimental results across nine instances from five standard EFG benchmarks—Kuhn Poker, Leduc Poker, Goofspiel, Liar's Dice, and Battleship, as well as two heads-up no-limit Texas Hold'em (HUNL) Subgames—demonstrate that $RTCFR^+$ achieves a significantly faster empirical convergence rate compared to existing algorithms with theoretical last-iterate convergence guarantees. Interestingly, $RTCFR^+$ even performs no worse well as average-iterate convergence CFR algorithms. Notably, it is the first last-iterate convergence algorithm to accomplish this level of performance.

## 2 Preliminaries

**Extensive-form games (EFGs).** EFG is a commonly used model for modeling tree-form sequential decision-making problems. An EFG can be formulated as $G = \{\mathcal{N}, \mathcal{H}, P, A, \mathcal{I}, \{u_i\}\}$. Here, $\mathcal{N}$ is the set of players. $\mathcal{H}$ is the set of all possible histories. The set of leaf nodes is denoted by $\mathcal{Z}$. For each history $h \in \mathcal{H}$, the function $P(h)$ represents the player acting at node $h$, and $A(h)$ denotes the actions available at node $h$. To account for private information, the nodes for each player $i$ are partitioned into a collection $\mathcal{I}_i$, referred to as information sets (infosets). For any infoset $I \in \mathcal{I}_i$, histories $h, h' \in I$ are indistinguishable to player $i$. Thus, $P(I) = P(h), A(I) = A(h), \forall h \in I$. The notation $\mathcal{I}$ denotes $\mathcal{I} = \{\mathcal{I}_i | i \in \mathcal{N}\}$. We also use $C_i(I, a)$ to denote the set of infosets that belongs to $i$ and will counter after executing $a \in A(I)$ at infoset $I \in \mathcal{I}_i$. The notations $A_{max}$ and $C_{max}$ denote $\max_{I \in \mathcal{I}} |A(I)|$ and $\max_{i \in \mathcal{N}, I \in \mathcal{I}_i, a \in A(I)} C_i(I, a)$, respectively. For each leaf node $z$, there is a pair $(u_0(z), u_1(z)) \in [-1, 1]$ which denotes the payoffs for the min player (player 0) and the max player (player 1), respectively. We define $H$ as the maximum number of actions taken by all players along any path from the root to a leaf node. In two-player zero-sum EFGs, $u_0(z) = -u_1(z), \forall z \in \mathcal{Z}$. To illustrate the components of an EFG, we provide an example in Appendix A.

**Sequence-form strategy.** A sequence is an infoset-action pair $(I, a)$, where $I \in \mathcal{I}$ is an infoset and $a$ is an action belonging to $A(I)$. Each sequence identifies a path from the root node to the infoset $I$, selecting the action $a$ along this path. The set of sequences for player $i$ is denoted by $\Sigma_i$. The last sequence encountered on the path from the root node $r$ to $I$ is denoted by $\rho_I$ ($\rho_I \in \Sigma_i$). In other words, $\forall i \in \mathcal{N}, I \in \mathcal{I}_i, I \in C_i(\rho_I)$. A sequence-form strategy for player $i$ is a non-negative vector $\boldsymbol{x}_i$ indexed over the set of sequences $\Sigma_i$. For each sequence $q = (I, a) \in \Sigma_i$, $\boldsymbol{x}_i(q)$ is the probability that player $i$ reaches the sequence $q$ when following the strategy $\boldsymbol{x}_i$. We formulate the sequence-form strategy space as a treeplex [Hoda et al., 2010]. Let $\boldsymbol{\mathcal{X}}_i$ denote the set of sequence-form strategies for player $i$. We use $\boldsymbol{x}_i(I) = [\boldsymbol{x}_i(I, a) | a \in A(I)]$ to denote the slice of a given strategy $\boldsymbol{x}_i$ corresponding to sequences belonging to infoset $I$, where $\boldsymbol{x}_i(I, a)$ is value of $\boldsymbol{x}_i$ at the sequence $(I, a)$. For each EFG, there always exists a $D$ such that $\forall i \in \mathcal{N}$ and $\boldsymbol{x}_i \in \boldsymbol{\mathcal{X}}_i, \|\boldsymbol{x}_i\|_1 \leq D$.

**Nash equilibrium (NE).** NE describes a rational behavior where no player can benefit by unilaterally deviating from the equilibrium. For any player, her strategy is the best response to the strategies of others. From the sequence-form strategy framework, learning an NE of EFGs is represented by

$$\min_{\boldsymbol{x}_0 \in \boldsymbol{\mathcal{X}}_0} \max_{\boldsymbol{x}_1 \in \boldsymbol{\mathcal{X}}_1} \boldsymbol{x}_0^{\mathrm{T}} \boldsymbol{A} \boldsymbol{x}_1, \tag{1}$$

where $\boldsymbol{A}$ is the payoff matrix. We use $\boldsymbol{\mathcal{X}}$ and $\boldsymbol{\mathcal{X}}^*$ to denote $\times_{i \in \mathcal{N}} \boldsymbol{\mathcal{X}}_i$ and the set of NE, respectively.

**Behavioral strategy.** This strategy $\sigma_i$ is defined on each infoset. For any infoset $I \in \mathcal{I}_i$, the probability for the action $a \in A(I)$ is denoted by $\sigma_i(I,a)$. We use $\sigma_i(I) = [\sigma_i(I,a)|a \in A(I)] \in \Delta^{|A(I)|}$ to denote the strategy at infoset $I$, where $\Delta^{|A(I)|}$ is a $(|A(I)|-1)$-dimension simplex. If all players follow the strategy profile $\sigma = \{\sigma_0, \sigma_1\}$ and reaches infoset $I$, the reaching probability is denoted by $\pi^\sigma(I)$. The probability contribution from player $i$ is represented by $\pi_i^\sigma(I)$, while the contribution from the other players is represented by $\pi_{-i}^\sigma(I)$, where $-i$ refers to all players except player $i$. Notably, $\forall i \in \mathcal{N}, I \in \mathcal{I}_i, a \in A(I), \boldsymbol{x}_i \in \boldsymbol{\mathcal{X}}_i, \boldsymbol{x}_i(I,a) = \pi_i^\sigma(I)\sigma_i(I,a)$, where $\sigma_i$ is the corresponding behavioral strategy of $\boldsymbol{x}_i$.

**Perturbed extensive-form games (Perturbed EFGs).** This game is a variant of the original EFG. Specifically, the strategy space of each infoset $I \in \mathcal{I}$ in a $\gamma$-perturbed EFG is a $\gamma$-perturbed simplex $\Delta_\gamma^{|A(I)|}$, a subset of $\Delta^{|A(I)|}$, rather than the standard simplex $\Delta^{|A(I)|}$ used in the original EFG, where $\gamma > 0$ is a constant. Formally, for any $\hat{\sigma}_i(I) \in \Delta_\gamma^{|A(I)|}$ and $a \in A(I)$, the constraint $\gamma \leq \hat{\sigma}_i(I,a) \leq 1$ holds, where $i = P(I)$. For convenience, we denote the set of sequence-form strategies for player $i$ in the $\gamma$-perturbed EFGs as $\boldsymbol{\mathcal{X}}_i^\gamma$. In $\gamma$-perturbed EFGs with $\gamma > 0$, any behavioral strategy $\hat{\sigma}_i$, with $\hat{\sigma}_i(I) \in \Delta_\gamma^{|A(I)|}$ for all $i \in \mathcal{N}$ and $I \in \mathcal{I}_i$, can be uniquely mapped to a sequence-form strategy $\hat{\boldsymbol{x}}_i \in \boldsymbol{\mathcal{X}}_i^\gamma$, and vice versa. Specifically, $\forall i \in \mathcal{N}, I \in \mathcal{I}_i, \hat{\sigma}_i(I) = \hat{\boldsymbol{x}}_i(I)/\hat{\boldsymbol{x}}_i(\rho_I) \geq \gamma$. Notably, $\forall i \in \mathcal{N}, \boldsymbol{\mathcal{X}}_i^\gamma$ is a subset of $\boldsymbol{\mathcal{X}}_i$. Similarly, we use the notation $\boldsymbol{\mathcal{X}}^\gamma$ and $\boldsymbol{\mathcal{X}}^{*,\gamma}$ to denote the joint strategy space $\times_{i \in \mathcal{N}} \boldsymbol{\mathcal{X}}_i^\gamma$ and the set of NEs of $\gamma$-perturbed EFGs, respectively.

**Learning an NE via regret minimization algorithms.** For any sequence of strategies $\boldsymbol{x}_i^1, \cdots, \boldsymbol{x}_i^T$ of of player $i$, player $i$'s regret is $R_i^T = \max_{\boldsymbol{x}_i \in \boldsymbol{\mathcal{X}}_i} \sum_{t=1}^T \langle \boldsymbol{\ell}_i^t, \boldsymbol{x}_i^t - \boldsymbol{x}_i \rangle$, where $\boldsymbol{\ell}_i^t$ is the loss for player $i$ at iteration $t$. Regret minimization algorithms are algorithms ensuring $R_i^T$ grows sublinearly. To learn an NE of EFGs via regret minimization algorithms, we set $\boldsymbol{\ell}_i^t = \boldsymbol{\ell}_i^{\boldsymbol{x}^t}$ with $\boldsymbol{\ell}_0^{\boldsymbol{x}} = \boldsymbol{A}\boldsymbol{x}_1$ and $\boldsymbol{\ell}_1^{\boldsymbol{x}} = -\boldsymbol{A}^{\mathrm{T}}\boldsymbol{x}_0$. If all players follow regret minimization algorithms, then the average strategy converges to the set of NEs in two-player zero-sum EFGs. In EFGs, there always exists $L$ and $P$ such that, $\forall \boldsymbol{x}, \boldsymbol{x}' \in \boldsymbol{\mathcal{X}}, \|\boldsymbol{\ell}^{\boldsymbol{x}} - \boldsymbol{\ell}^{\boldsymbol{x}'}\|_1 \leq L\|\boldsymbol{x} - \boldsymbol{x}'\|_1$ and $\|\boldsymbol{\ell}^{\boldsymbol{x}}\|_1 \leq P$, where $\boldsymbol{\ell}^{\boldsymbol{x}} = [\boldsymbol{\ell}_i^{\boldsymbol{x}}|i \in \mathcal{N}]$, as well as $L > 0$ and $P > 0$ are game-dependent constants.

**Counterfactual regret minimization (CFR) framework.** This framework [Zinkevich et al., 2007, Farina et al., 2019] is designed to solve EFGs by decomposing the global regret $R_i^T$ into local regrets at each infoset, allowing for independent minimization within each infoset, rather than directly minimizing global regret. This approach has led to the development of several superhuman Game AIs [Bowling et al., 2015, Moravčík et al., 2017, Brown and Sandholm, 2018, 2019b, Pérolat et al., 2022]. Formally, for player $i$, given the observed loss when all players follow $\boldsymbol{x} \in \boldsymbol{\mathcal{X}}$ is $\boldsymbol{\ell}_i^{\boldsymbol{x}}$, the CFR framework computes the counterfactual values at each infoset $I \in \mathcal{I}_i$ according to

$$\boldsymbol{v}_i^{\boldsymbol{x}}(I,a) = -\boldsymbol{\ell}_i^{\boldsymbol{x}}(I,a) + \sum_{I' \in C_i(I,a)} \langle \boldsymbol{v}_i^{\boldsymbol{x}}(I'), \sigma_i(I') \rangle$$

where $\boldsymbol{\ell}_i^{\boldsymbol{x}}(I,a)$ is the value of $\boldsymbol{\ell}_i^{\boldsymbol{x}}$ at the sequence $(I,a)$, $\boldsymbol{v}_i^{\boldsymbol{x}}(I') = [\boldsymbol{v}_i^{\boldsymbol{x}}(I',a')|a' \in A(I')]$, and $\sigma_i$ represents the behavioral strategy of player $i$ corresponds to $\boldsymbol{x}_i$. Farina et al. [2019] demonstrate that

$$R_i^T = \max_{\boldsymbol{x}_i \in \boldsymbol{\mathcal{X}}_i} \sum_{t=1}^T \langle \boldsymbol{\ell}_i^t, \boldsymbol{x}_i^t - \boldsymbol{x}_i \rangle \leq \sum_{I \in \mathcal{I}_i} \max_{\sigma_i(I)} \sum_{t=1}^T \langle \boldsymbol{v}_i^t(I), \sigma_i(I) - \sigma_i^t(I) \rangle,$$

where $\boldsymbol{v}_i^t(I) = \boldsymbol{v}_i^{\boldsymbol{x}^t}(I) = [\boldsymbol{v}_i^{\boldsymbol{x}^t}(I,a)|a \in A(I)]$ and $\sigma_i^t$ is the behavioral strategy of player $i$ corresponds to $\boldsymbol{x}_i^t$. It indicates that minimizing the local regret $\max_{\sigma_i(I)} \sum_{t=1}^T \langle \boldsymbol{v}_i^t(I), \sigma_i(I) - \sigma_i^t(I) \rangle$ at $I \in \mathcal{I}_i$ contributes to minimizing the global regret $R_i^T$.

**Blackwell approachability framework.** RM algorithms are come from this framework whose core insight lies in reframing the problem of regret minimization within the orignial strategy space $\boldsymbol{\mathcal{Z}}$ as

regret minimization within $\text{cone}(\boldsymbol{\mathcal{Z}}) = \{\lambda \boldsymbol{z} \mid \boldsymbol{z} \in \boldsymbol{\mathcal{Z}}, \lambda \geq 0\}$ [Blackwell, 1956, Abernethy et al., 2011, Farina et al., 2021]. Specifically, a regret minimization algorithm is instantiated in $\text{cone}(\boldsymbol{\mathcal{Z}})$, where its output at iteration $t$ is $\boldsymbol{\theta}^t$. This corresponds to the strategy $\boldsymbol{z}^t = \boldsymbol{\theta}^t / \langle \boldsymbol{\theta}^t, \mathbf{1} \rangle$ within $\boldsymbol{\mathcal{Z}}$. Given the loss $\boldsymbol{\ell}^t$ at iteration $t$, the algorithm observes the transformed loss $-\boldsymbol{m}^t = -\langle \boldsymbol{\ell}^t, \boldsymbol{z}^t \rangle \mathbf{1} + \boldsymbol{\ell}^t$ and subsequently generates $\boldsymbol{\theta}^{t+1}$. The main advantage of this framework is its capacity to develop parameter-free algorithms. More details are provided below.

**Regret Matching$^+$ (RM$^+$).** To minimize local regret within each infoset, CFR algorithms commonly employ local regret minimizers based on RM [Hart and Mas-Colell, 2000, Gordon, 2006, Bowling et al., 2015, Farina et al., 2021, 2023, Xu et al., 2022, 2024b, Cai et al., 2025], which show strong empirical convergence rate and are typically parameter-free. In this paper, we focus on RM$^+$ [Tammelin, 2014], a variant of RM that typically exhibits a faster empirical convergence rate than vanilla RM. RM$^+$ is a traditional algorithm grounded in Blackwell approachability framework. It corresponds to an OMD instantiated in the cone of the simplex [Farina et al., 2021]. Formally, at each iteration $t$ and infoset $I \in \mathcal{I}_i$, RM$^+$ updates the strategy via

$$\boldsymbol{\theta}_I^{t+1} \in \underset{\boldsymbol{\theta}_I \in \mathbb{R}_{\geq 0}^{|A(I)|}}{\arg\min} \left\{ \langle -\boldsymbol{m}_i^t(I), \boldsymbol{\theta}_I \rangle + \frac{1}{\eta} D_\psi(\boldsymbol{\theta}_I, \boldsymbol{\theta}_I^t) \right\}, \quad \sigma_i^{t+1}(I) = \frac{\boldsymbol{\theta}_I^{t+1}}{\langle \boldsymbol{\theta}_I^{t+1}, \mathbf{1} \rangle},$$

where $i = P(I)$, $\eta > 0$ is the step size, $\boldsymbol{m}_i^t(I) = -\langle \boldsymbol{v}_i^t(I), \sigma_i^t(I) \rangle \mathbf{1} + \boldsymbol{v}_i^t(I)$ represents the instantaneous counterfactual regret, and $D_\psi(\boldsymbol{u}, \boldsymbol{v}) = \psi(\boldsymbol{u}) - \psi(\boldsymbol{v}) - \langle \nabla \psi(\boldsymbol{v}), \boldsymbol{u} - \boldsymbol{v} \rangle$ is the Bregman divergence associated with the quadratic regularizer $\psi(\cdot) = \| \cdot \|_2^2 / 2$. If $\boldsymbol{\theta}_I^1 = 0$, for all the step size $\eta > 0$, the output sequence $\{\sigma_i^1(I), \sigma_i^2(I), \ldots, \sigma_i^t(I), \ldots\}$ remains unchanged [Farina et al., 2021]. Combining RM$^+$ with the CFR framework yields CFR$^+$ [Tammelin, 2014], which is a parameter-free CFR algorithm and has been used to build superhuman poker AI [Bowling et al., 2015].

## 3  Problem Statement

To demonstrate the last-iterate convergence of CFR algorithms, Pérolat et al. [2021, 2022], Liu et al. [2023] employ the RT framework. This framework reformulates the objective of learning an NE for the original EFG into finding NEs for a series of (perturbed) regularized EFGs, and ensures that the sequence of NEs of the regularized EFGs converges to the set of NEs of the original EFG. Therefore, establishing last-iterate convergence in learning an NE of the original EFG reduces to establishing last-iterate convergence in learning an NE of (perturbed) regularized EFGs. Inspired by Pérolat et al. [2021], Liu et al. [2023], Abe et al. [2024], we consider the following perturbed regularized EFG:

$$\min_{\hat{\boldsymbol{x}}_0 \in \boldsymbol{\mathcal{X}}_0^\gamma} \max_{\hat{\boldsymbol{x}}_1 \in \boldsymbol{\mathcal{X}}_1^\gamma} \hat{\boldsymbol{x}}_0^{\mathrm{T}} \boldsymbol{A} \hat{\boldsymbol{x}}_1 + \mu D_\psi(\hat{\boldsymbol{x}}_0, \boldsymbol{r}_0) - \mu D_\psi(\hat{\boldsymbol{x}}_1, \boldsymbol{r}_1), \tag{2}$$

where $\gamma > 0$ and $\mu > 0$ are constants, $\psi(\cdot)$ is the quadratic regularizer, and $\boldsymbol{r} = [\boldsymbol{r}_0; \boldsymbol{r}_1] \in \boldsymbol{\mathcal{X}}$ is the reference strategy profile. The NE of this perturbed regularized EFG is unique and denoted by $\hat{\boldsymbol{x}}^{*,\gamma,\mu,\boldsymbol{r}}$ or $\hat{\sigma}^{*,\gamma,\mu,\boldsymbol{r}}$. To ensure the sequence of the NEs of the perturbed regularized EFGs converges to the set of NEs of the original EFG, a valid approach is to continuously decreasing the value of $\gamma$ and updating $\boldsymbol{r}$ to $\hat{\boldsymbol{x}}^{*,\gamma,\mu,\boldsymbol{r}}$, according to the studies in Abe et al. [2024], Bernasconi et al. [2024]. Another approach involves simultaneously reducing the values of $\gamma$ and $\mu$ [Liu et al., 2023, Bernasconi et al., 2024]. Notably, in the approach where simultaneously reducing the values of $\gamma$ and $\mu$, updating $\boldsymbol{r}$ to $\hat{\boldsymbol{x}}^{*,\gamma,\mu,\boldsymbol{r}}$ is optional. Consequently, achieving the last-iterate convergence for solving Eq. (2) implies achieving the last-iterate convergence for solving Eq. (1). This paper refrains from investigating the RT framework and its convergence as these have been thoroughly investigated in other studies [Pérolat et al., 2021, Liu et al., 2023, Abe et al., 2024, Bernasconi et al., 2024, Wang et al., 2025].

The introduction of perturbation and regularization ensures the smoothness of counterfactual values and the strong monotonicity, respectively. The smoothness is $\|\boldsymbol{v}_i^{\hat{\sigma}}(I) - \boldsymbol{v}_i^{\hat{\sigma}'}(I)\|_1 \leq O(\|\hat{\boldsymbol{x}} - \hat{\boldsymbol{x}}'\|_1)$, $\forall \hat{\boldsymbol{x}}, \hat{\boldsymbol{x}}' \in \boldsymbol{\mathcal{X}}^\gamma$, where $\hat{\sigma}$ and $\hat{\sigma}'$ are the behavioral strategy profiles associated with $\hat{\boldsymbol{x}}$ and $\hat{\boldsymbol{x}}'$, respectively. The strong monotonicity indicates that $O(\langle \boldsymbol{\ell}^{\hat{\boldsymbol{x}}} - \boldsymbol{\ell}^{\hat{\boldsymbol{x}}'}, \hat{\boldsymbol{x}} - \hat{\boldsymbol{x}}' \rangle) \geq \|\hat{\boldsymbol{x}} - \hat{\boldsymbol{x}}'\|_2^2$, $\forall \hat{\boldsymbol{x}}, \hat{\boldsymbol{x}}' \in \boldsymbol{\mathcal{X}}^\gamma$.

Although some works have investigated the last-iterate convergence of CFR algorithms for solving perturbed regularized EFGs [Liu et al., 2023], their algorithms do not use RM-based algorithms as the local regret minimizer. The absence of RM-based algorithms leads to significantly weaker empirical last-iterate convergence performance than traditional RM-based average-iterate convergence CFR algorithms, as shown in our experiments. In addition, as solving multiple perturbed regularized EFGs

is required, fine-tuning across all perturbed regularized EFGs is infeasible. Consequently, parameter-free algorithms, implying no parameters need to be tuned [Grand-Clément and Kroer, 2021], are desirable. Based on these observations, we propose Reward Transformation CFR$^+$ (RTCFR$^+$), utilizing CFR+ [Tammelin, 2014], a classical parameter-free RM-based CFR algorithm, to solve perturbed regularized EFGs defined in Eq. (2) (details of RTCFR$^+$ are in Section 4). Unfortunately, it remains unknown whether CFR$^+$ achieves the parameter-free (i.e., holds for any step sizes) last-iterate convergence in solving Eq. (2). Thus, our objective is to establish the parameter-free last-iterate convergence for CFR$^+$ in solving Eq. (2). More discussions about the related works are in Appendix B.

# 4 Last-Iterate Convergence of CFR$^+$ in Solving Perturbed Regularized EFGs

Now, we show that CFR$^+$ exhibits last-iterate convergence for solving the perturbed regularized EFGs defined in Eq. (2). Before introducing the last-iterate convergence of CFR$^+$, we first extend CFR$^+$ to perturbed EFGs as the original CFR$^+$ algorithm is only designed for the case where $\gamma = 0$. Specifically, we (i) first update the accumulated counterfactual regrets within the original simplex's cone while ensuring strategy outputs lie within the perturbed simplex by mixing the non-perturbed strategy formed by the accumulated counterfactual regrets with the uniform vector, then (ii) compute the instantaneous counterfactual regrets using the non-perturbed strategy and the counterfactual values observed through following the output perturbed strategy. This enables the use of the strong monotonicity to establish last-iterate convergence in learning an NE of the perturbed regularized EFGs in Eq. (2), as shown in Eq. (6). Formally, the update rule of CFR$^+$ for learning an NE of the perturbed regularized EFGs in Eq. (2) at iteration $t$ and infoset $I \in \mathcal{I}_i$ is

$$
\begin{aligned}
&\boldsymbol{\theta}_I^{t+1} \in \underset{\boldsymbol{\theta}_I \in \mathbb{R}_{\geq 0}^{|A(I)|}}{\arg\min} \left\{ \langle -\hat{\boldsymbol{m}}_i^t(I), \boldsymbol{\theta}_I \rangle + \frac{1}{\eta} D_\psi(\boldsymbol{\theta}_I, \boldsymbol{\theta}_I^t) \right\}, \ \sigma_i^{t+1}(I) = \frac{\boldsymbol{\theta}_I^{t+1}}{\langle \boldsymbol{\theta}_I^{t+1}, \mathbf{1} \rangle}, \\
&\hat{\sigma}_i^{t+1}(I) = (1 - \alpha_I)\sigma_i^{t+1}(I) + \gamma\mathbf{1}, \ \alpha_I = \gamma|A(I)|, \\
&\hat{\boldsymbol{m}}_i^t(I) = \hat{\boldsymbol{v}}_i^t(I) - \langle \hat{\boldsymbol{v}}_i^t(I), \sigma_i^t(I) \rangle\mathbf{1}, \\
&\hat{\boldsymbol{v}}_i^t(I, a) = -\hat{\boldsymbol{\ell}}_i^t(I, a) + \sum_{I' \in C_i(I,a)} \langle \hat{\boldsymbol{v}}_i^t(I'), \hat{\sigma}_i^t(I') \rangle, \\
&\hat{\boldsymbol{\ell}}_0^t = \boldsymbol{A}\hat{\boldsymbol{x}}_1^t + \mu\nabla\psi(\hat{\boldsymbol{x}}_0^t) - \mu\nabla\psi(\boldsymbol{r}_0), \ \hat{\boldsymbol{\ell}}_1^t = -\boldsymbol{A}^{\mathrm{T}}\hat{\boldsymbol{x}}_0^t + \mu\nabla\psi(\hat{\boldsymbol{x}}_1^t) - \mu\nabla\psi(\boldsymbol{r}_1),
\end{aligned}
\tag{3}
$$

where $\eta > 0$ is the step size and $\hat{\boldsymbol{x}}_i^t(I) = \pi_i^{\hat{\sigma}^t}(I)\hat{\sigma}_i^t(I)$. The second line in Eq. (3) mixes the non-perturbed strategy $\sigma$ with the uniform vector $\mathbf{1}$, while the third line constructs the instantaneous counterfactual regrets $\hat{\boldsymbol{m}}_I^t$ using the non-perturbed strategy $\sigma_i^t$ derived from accumulated counterfactual regrets $\boldsymbol{\theta}_I^t$ and counterfactual values $\hat{\boldsymbol{v}}_i^t$ obtained from the perturbed strategy $\hat{\sigma}_i^t$.

**Theorem 4.1** (Proof is in Appendix D). *Assuming all players follow the update rule of CFR$^+$ with any $\boldsymbol{\theta}_I^1 \in \mathbb{R}_{\geq 0}^{|A(I)|}$ and $\eta > 0$, the strategy profile $\hat{\boldsymbol{x}}^t$ converges to the set of NEs of the perturbed regularized EFGs defined in Eq. (2) with any $\gamma > 0$ and $\mu > 0$.*

**Proof sketch of Theorem 4.1.** Our proof consists of two steps. Firstly, we establish the non-parameter-free last-iterate convergence; that is, for all $\boldsymbol{\theta}_I^1 \in \mathbb{R}_{\geq 0}^{|A(I)|}$, the last-iterate convergence of CFR$^+$ in solving Eq. (2) holds when $\eta$ exceeds a certain constant. The principal challenge is that the smoothness of the instantaneous counterfactual regrets cannot be used since RM algorithms update within the cone of the strategy space, $\mathrm{cone}(\Delta^{A(I)})$, whereas the final output lies in the strategy space, $\Delta^{A(I)}$. We address this challenge by leveraging the fact that an NE is a best response to other strategies at each infoset in perturbed EFGs, as shown in the text around Eq. (5) and (6), as well as Lemma 4.4. Secondly, we derive the parameter-free convergence result, namely, that the last-iterate convergence of CFR$^+$ holds for all $\boldsymbol{\theta}_I^1 \in \mathbb{R}_{\geq 0}^{|A(I)|}$ and $\eta > 0$. The main challenge here is that the property used in previous proofs of the parameter-free property—that the strategy sequence produced by CFR$^+$ is invariant w.r.t. different step sizes $\eta > 0$—holds only when $\boldsymbol{\theta}_I^1 = \mathbf{0}$. We overcome this by exploiting the linearity of the projection in CFR$^+$ and the fact that our non-parameter-free last-iterate convergence of CFR$^+$ holds for all $\boldsymbol{\theta}_I^1 \in \mathbb{R}_{\geq 0}^{|A(I)|}$, as presented in the second paragraph following Lemma 4.4. The details of our proof sketch is shown in the following.

**Lemma 4.2** (Adapted from the proof of Lemma 4 in Farina et al. [2021]). *Assuming all players follow the update rule of CFR$^+$, then for any $\boldsymbol{\theta}_I \in \mathbb{R}^{|A(I)|}_{\geq 0}$, we have*

$$D_\psi(\boldsymbol{\theta}_I, \boldsymbol{\theta}_I^{t+1}) - D_\psi(\boldsymbol{\theta}_I, \boldsymbol{\theta}_I^t) \leq \eta\langle \hat{\boldsymbol{m}}_i^t(I), \boldsymbol{\theta}_I^{t+1} - \boldsymbol{\theta}_I\rangle - D_\psi(\boldsymbol{\theta}_I^{t+1}, \boldsymbol{\theta}_I^t).$$

By applying Lemma 4.2 with $\boldsymbol{\theta}_I = \sigma_i^{*,\mu,\gamma,\boldsymbol{r}}(I) = (\hat{\sigma}_i^{*,\mu,\gamma,\boldsymbol{r}}(I) - \gamma\mathbf{1})/(1 - \alpha_I) \in \Delta^{|A(I)|}$, we get

$$\eta\langle \hat{\boldsymbol{m}}_i^t(I), \sigma_i^{*,\mu,\gamma,\boldsymbol{r}}(I) - \boldsymbol{\theta}_I^{t+1}\rangle \leq D_\psi(\sigma_i^{*,\mu,\gamma,\boldsymbol{r}}(I), \boldsymbol{\theta}_I^t) - D_\psi(\sigma_i^{*,\mu,\gamma,\boldsymbol{r}}(I), \boldsymbol{\theta}_I^{t+1}) - D_\psi(\boldsymbol{\theta}_I^{t+1}, \boldsymbol{\theta}_I^t). \quad (4)$$

Also, we define

$$\hat{\boldsymbol{m}}_i^{*,\mu,\gamma,\boldsymbol{r}}(I) = \hat{\boldsymbol{v}}_i^{*,\mu,\gamma,\boldsymbol{r}}(I) - \langle \hat{\boldsymbol{v}}_i^{*,\mu,\gamma,\boldsymbol{r}}(I), \sigma_i^{*,\mu,\gamma,\boldsymbol{r}}(I)\rangle\mathbf{1},$$

$$\hat{\boldsymbol{v}}_i^{*,\mu,\gamma,\boldsymbol{r}}(I) = -\hat{\boldsymbol{\ell}}_i^{*,\mu,\gamma,\boldsymbol{r}}(I,a) + \sum_{I' \in C_i(I,a)} \langle \hat{\boldsymbol{v}}_i^{*,\mu,\gamma,\boldsymbol{r}}(I'), \hat{\sigma}_i^{*,\mu,\gamma,\boldsymbol{r}}(I')\rangle,$$

$$\hat{\boldsymbol{\ell}}_0^{*,\mu,\gamma,\boldsymbol{r}} = \boldsymbol{A}\hat{\boldsymbol{x}}_1^{*,\mu,\gamma,\boldsymbol{r}} + \mu\nabla\psi(\hat{\boldsymbol{x}}_0^{*,\mu,\gamma,\boldsymbol{r}}) - \mu\nabla\psi(\boldsymbol{r}_0), \hat{\boldsymbol{\ell}}_1^{*,\mu,\gamma,\boldsymbol{r}} = -\boldsymbol{A}^{\mathrm{T}}\hat{\boldsymbol{x}}_0^{*,\mu,\gamma,\boldsymbol{r}} + \mu\nabla\psi(\hat{\boldsymbol{x}}_1^{*,\mu,\gamma,\boldsymbol{r}}) - \mu\nabla\psi(\boldsymbol{r}_1).$$

Then, adding $\eta\langle -\hat{\boldsymbol{m}}_i^{*,\mu,\gamma,\boldsymbol{r}}(I), \boldsymbol{\theta}_I^{t+1} - \boldsymbol{\theta}_I^t\rangle$ to each hand side of Eq. (4), we can get

$$\eta\langle \hat{\boldsymbol{m}}_i^t(I), \sigma_i^{*,\mu,\gamma,\boldsymbol{r}}(I) - \boldsymbol{\theta}_I^t\rangle - \eta^2\frac{\|\hat{\boldsymbol{m}}_i^t(I) - \hat{\boldsymbol{m}}_i^{*,\mu,\gamma,\boldsymbol{r}}(I)\|_2^2}{2}$$
$$\leq D_\psi(\sigma_i^{*,\mu,\gamma,\boldsymbol{r}}(I), \boldsymbol{\theta}_I^t) + \eta\langle -\hat{\boldsymbol{m}}_i^{*,\mu,\gamma,\boldsymbol{r}}(I), \boldsymbol{\theta}_I^t\rangle - D_\psi(\sigma_i^{*,\mu,\gamma,\boldsymbol{r}}(I), \boldsymbol{\theta}_I^{t+1}) - \eta\langle -\hat{\boldsymbol{m}}_i^{*,\mu,\gamma,\boldsymbol{r}}(I), \boldsymbol{\theta}_I^{t+1}\rangle. \quad (5)$$

In OMD algorithms [Sokota et al., 2023], the addition of the term $\eta\langle -\hat{\boldsymbol{m}}_i^{*,\mu,\gamma,\boldsymbol{r}}(I), \boldsymbol{\theta}_I^{t+1} - \boldsymbol{\theta}_I^t\rangle$ is not required to exploit the smoothness of the instantaneous counterfactual regrets. However, this term is necessary to prove the last-iterate convergence of CFR$^+$. This step is crucial in our proof, and to the best of our knowledge, no prior work has proposed a similar approach.

**Lemma 4.3** (Proof is in Appendix E.1). *For any $\boldsymbol{x}, \boldsymbol{x}' \in \mathcal{X}$, $\boldsymbol{\ell} \in \mathbb{R}^{|\mathcal{X}|}$, $i \in \mathcal{N}$, $\mu \geq 0$, and $\gamma \geq 0$,*

$$\langle \boldsymbol{\ell}_i, \boldsymbol{x}_i - \boldsymbol{x}_i'\rangle = \sum_{I \in \mathcal{I}_i} \pi_i^{\sigma'}(I)\langle -\boldsymbol{v}_i^\sigma(I), \sigma_i(I) - \sigma_i'(I)\rangle,$$

*where $\boldsymbol{v}_i^\sigma(I) = [\boldsymbol{v}_i^\sigma(I,a)|a \in A(I)]$ with $\boldsymbol{v}_i^\sigma(I,a) = -\boldsymbol{\ell}_i(I,a) + \sum_{I' \in C_i(I,a)} \langle \boldsymbol{v}_i^\sigma(I'), \sigma_i(I')\rangle$, as well as $\sigma$ and $\sigma'$ are the behavioral strategy profiles associated with $\boldsymbol{x}$ and $\boldsymbol{x}'$, respectively.*

Combining Eq. (5) with Lemma 4.3, and setting $\zeta_I = (1 - \alpha_I)\beta_I$ with $\beta_I = \pi_i^{\hat{\sigma}^{*,\mu,\gamma,\boldsymbol{r}}}(I)$, we have

$$\eta\sum_{t=1}^T\sum_{i \in \mathcal{N}} \langle \hat{\boldsymbol{\ell}}_i^t, \hat{\boldsymbol{x}}_i^t - \hat{\boldsymbol{x}}_i^{*,\mu,\gamma,\boldsymbol{r}}\rangle - \sum_{t=1}^T\sum_{i \in \mathcal{N}}\sum_{I \in \mathcal{I}_i} \eta^2\frac{\|\hat{\boldsymbol{m}}_i^t(I) - \hat{\boldsymbol{m}}_i^{*,\mu,\gamma,\boldsymbol{r}}(I)\|_2^2}{2}$$
$$\leq \sum_{i \in \mathcal{N}}\sum_{I \in \mathcal{I}_i} \zeta_I\Big( D_\psi(\sigma_i^{*,\mu,\gamma,\boldsymbol{r}}(I), \boldsymbol{\theta}_I^1) + \eta\langle -\hat{\boldsymbol{m}}_i^{*,\mu,\gamma,\boldsymbol{r}}(I), \boldsymbol{\theta}_I^1\rangle - D_\psi(\sigma_i^{*,\mu,\gamma,\boldsymbol{r}}(I), \boldsymbol{\theta}_I^{T+1}) - \eta\langle -\hat{\boldsymbol{m}}_i^{*,\mu,\gamma,\boldsymbol{r}}(I), \boldsymbol{\theta}_I^{T+1}\rangle\Big).$$

By using the strong monotonicity ($O(\sum_{t=1}^T\sum_{i \in \mathcal{N}} \langle \hat{\boldsymbol{\ell}}_i^t, \hat{\boldsymbol{x}}_i^t - \hat{\boldsymbol{x}}_i^{*,\mu,\gamma,\boldsymbol{r}}\rangle) \geq \|\hat{\boldsymbol{x}}^t - \hat{\boldsymbol{x}}^{*,\mu,\gamma,\boldsymbol{r}}\|_2^2$, as shown in Lemma D.1) and the smoothness of instantaneous counterfactual regrets ($\|\hat{\boldsymbol{m}}_i^t(I) - \hat{\boldsymbol{m}}_i^{*,\mu,\gamma,\boldsymbol{r}}(I)\|_2^2 \leq O(\|\hat{\boldsymbol{x}}^t - \hat{\boldsymbol{x}}^{*,\mu,\gamma,\boldsymbol{r}}\|_2^2)$) (see details in Appendix D), we get

$$\mu\eta\sum_{t=1}^T \|\hat{\boldsymbol{x}}^t - \hat{\boldsymbol{x}}^{*,\mu,\gamma,\boldsymbol{r}}\|_2^2 - \sum_{t=1}^T \eta^2 C_0\|\hat{\boldsymbol{x}}^t - \hat{\boldsymbol{x}}^{*,\mu,\gamma,\boldsymbol{r}}\|_2^2 \leq \sum_{i \in \mathcal{N}}\sum_{I \in \mathcal{I}_i} \zeta_I\bigg( D_\psi(\sigma_i^{*,\mu,\gamma,\boldsymbol{r}}(I), \boldsymbol{\theta}_I^1)$$
$$+ \eta\langle -\hat{\boldsymbol{m}}_i^{*,\mu,\gamma,\boldsymbol{r}}(I), \boldsymbol{\theta}_I^1\rangle - D_\psi(\sigma_i^{*,\mu,\gamma,\boldsymbol{r}}(I), \boldsymbol{\theta}_I^{T+1}) - \eta\langle -\hat{\boldsymbol{m}}_i^{*,\mu,\gamma,\boldsymbol{r}}(I), \boldsymbol{\theta}_I^{T+1}\rangle\bigg), \quad (6)$$

where $C_0 = |\mathcal{I}|A_{max}^2\left(6(L+\mu)^2 + 8(P + 2\mu D)^2(A_{max}C_{max} + 1)^2/\gamma^{2H}\right)$. Note that the form of smoothness we adopt differs from that commonly used in OMD algorithms [Sokota et al., 2023], where smoothness typically takes the form $\|\hat{\boldsymbol{m}}_i^t(I) - \hat{\boldsymbol{m}}_i^{t+1}(I)\|_2^2 \leq O(\|\hat{\boldsymbol{x}}^t - \hat{\boldsymbol{x}}^{t+1}\|)$ rather than $\|\hat{\boldsymbol{m}}_i^t(I) - \hat{\boldsymbol{m}}_i^{*,\mu,\gamma,\boldsymbol{r}}(I)\|_2^2 \leq O(\|\hat{\boldsymbol{x}}^t - \hat{\boldsymbol{x}}^{*,\mu,\gamma,\boldsymbol{r}}\|_2^2)$. This difference also highlights that our proof approach diverges from the approach used by OMD algorithms. Then, if $0 < \eta \leq \mu/(2C_0)$, we get

$$\frac{\mu\eta}{2}\sum_{t=1}^T \|\hat{\boldsymbol{x}}^t - \hat{\boldsymbol{x}}^{*,\mu,\gamma,\boldsymbol{r}}\|_2^2 \leq \sum_{i \in \mathcal{N}}\sum_{I \in \mathcal{I}_i} \zeta_I\bigg( D_\psi(\sigma_i^{*,\mu,\gamma,\boldsymbol{r}}(I), \boldsymbol{\theta}_I^1)$$
$$+ \eta\langle -\hat{\boldsymbol{m}}_i^{*,\mu,\gamma,\boldsymbol{r}}(I), \boldsymbol{\theta}_I^1\rangle - D_\psi(\sigma_i^{*,\mu,\gamma,\boldsymbol{r}}(I), \boldsymbol{\theta}_I^{T+1}) - \eta\langle -\hat{\boldsymbol{m}}_i^{*,\mu,\gamma,\boldsymbol{r}}(I), \boldsymbol{\theta}_I^{T+1}\rangle\bigg).$$

**Lemma 4.4** (Proof is in Appendix E.2). $\forall i \in \mathcal{N}$, $I \in \mathcal{I}_i$, and $\boldsymbol{\theta}_I \in \mathbb{R}^{|A(I)|}_{\geq 0}$, $\langle -\hat{\boldsymbol{m}}_i^{*,\mu,\gamma,\boldsymbol{r}}(I), \boldsymbol{\theta}_I \rangle \geq 0$.

Lemma 4.4 is from that an NE is a best response to others at each infoset in perturbed EFGs, i.e., $\forall \sigma_i$, $\langle \hat{\boldsymbol{v}}_i^{*,\mu,\gamma,\boldsymbol{r}}(I), \hat{\sigma}_i^{*,\mu,\gamma,\boldsymbol{r}}(I) - \hat{\sigma}_i(I) \rangle \geq 0$, where $\hat{\sigma}_i(I) = (1-\alpha_I)\sigma_i(I) + \gamma \mathbf{1}$ (details are in Appendix E.2). By using Lemma 4.4, we get $\forall T \geq 1, \sum_{t=1}^T \|\hat{\boldsymbol{x}}^t - \hat{\boldsymbol{x}}^{*,\mu,\gamma,\boldsymbol{r}}\|_2^2 \leq O(1)$, implying that $\hat{\boldsymbol{x}}^t$ converges to $\hat{\boldsymbol{x}}^{*,\mu,\gamma,\boldsymbol{r}}$ with $0 < \eta \leq \mu/(2C_0)$.

Farina et al. [2021] show that when $\boldsymbol{\theta}_I^1 = \mathbf{0}$, for any $\eta > 0$, the sequence $\{\hat{\boldsymbol{x}}^1, \hat{\boldsymbol{x}}^2, \cdots, \hat{\boldsymbol{x}}^t, \cdots\}$ remains the same. This implies that $\hat{\boldsymbol{x}}^t$ converges to $\hat{\boldsymbol{x}}^{*,\mu,\gamma,\boldsymbol{r}}$ for any $\eta > 0$, showing the parameter-free property. In this paper, we further show that for any initial $\boldsymbol{\theta}_I^1 \in \mathbb{R}^{|A(I)|}_{\geq 0}$ and $\eta > 0$, $\hat{\boldsymbol{x}}^t$ converges to $\hat{\boldsymbol{x}}^{*,\mu,\gamma,\boldsymbol{r}}$ (see advantages in discussions). This proof is simple yet novel, with the key insights being the linearity of the projection in CFR$^+$ and that $\sum_{t=1}^T \|\hat{\boldsymbol{x}}^t - \hat{\boldsymbol{x}}^{*,\mu,\gamma,\boldsymbol{r}}\|_2^2 \leq O(1)$ holds independently of the value of $\boldsymbol{\theta}_I^1$. Specifically, from the linearity of the projection in CFR$^+$, for any accumulated counterfactual regret sequence $\{\boldsymbol{\theta}_I^1, \boldsymbol{\theta}_I^2, \ldots, \boldsymbol{\theta}_I^t, \ldots\}$ generated by any $\boldsymbol{\theta}_I^1 \in \mathbb{R}^{|A(I)|}_{\geq 0}$ and $\eta > 0$, there exists a corresponding accumulated counterfactual regret sequence $\{\boldsymbol{\theta}_I^{1'}, \boldsymbol{\theta}_I^{2'}, \ldots, \boldsymbol{\theta}_I^{t'}, \ldots\}$ generated by $\boldsymbol{\theta}_I^{1'}$ and $\eta' = \mu/(2C_0)$, such that the resulting strategy profile sequence $\{\hat{\boldsymbol{x}}^1, \hat{\boldsymbol{x}}^2, \ldots, \hat{\boldsymbol{x}}^t, \ldots\}$ are identical. Additionally, as the condition $\sum_{t=1}^T \|\hat{\boldsymbol{x}}^t - \hat{\boldsymbol{x}}^{*,\mu,\gamma,\boldsymbol{r}}\|_2^2 \leq O(1)$ holds independently of the value of $\boldsymbol{\theta}_I^1$ ($\boldsymbol{\theta}_I^{1'}$). Based on this analysis, we conclude that for any accumulated counterfactual regret sequence $\{\boldsymbol{\theta}_I^1, \boldsymbol{\theta}_I^2, \ldots, \boldsymbol{\theta}_I^t, \ldots\}$ generated by any $\boldsymbol{\theta}_I^1$ and $\eta > 0$, the corresponding strategy profile sequence $\{\hat{\boldsymbol{x}}^1, \hat{\boldsymbol{x}}^2, \ldots, \hat{\boldsymbol{x}}^t, \ldots\}$ converges to $\hat{\boldsymbol{x}}^{*,\mu,\gamma,\boldsymbol{r}}$, which indicates the parameter-free property.

**Reward Transformation CFR$^+$ (RTCFR$^+$).** RTCFR$^+$ is the RT algorithm that applies CFR$^+$ to solve perturbed regularized EFGs, whose pseudocode is in Algorithm 1. As analyzed by Abe et al. [2024], Bernasconi et al. [2024], continuously decreasing $\gamma$ and updating $\boldsymbol{r}$ to $\hat{\boldsymbol{x}}^{*,\gamma,\mu,\boldsymbol{r}}$ allows the sequence of the NEs of the perturbed regularized EFGs to converge to the set of NEs of the original EFG. Specifically, as shown in Algorithm 1, after $T_u$ iterations, RTCFR$^+$ updates $\gamma$ and $\boldsymbol{r}$, with $N * T_u$ representing the total number of iterations. The implementation of RTCFR$^+$ is in Appendix H.

For RTCFR$^+$, we do not examine the convergence of the sequence of the NEs of the perturbed regularized EFGs to the set of NEs of the original EFG when the exact $\hat{\boldsymbol{x}}^{*,\gamma,\mu,\boldsymbol{r}}$ is not learned but only an approximate $\hat{\boldsymbol{x}}^{*,\gamma,\mu,\boldsymbol{r}}$ is obtained, as this problem can be solved by simultaneously decreasing the values of $\mu$ and $\gamma$, as mentioned in Section 3. Formally, line 8 of Algorithm 1 can be modified as: $\mu \leftarrow \mu \times (1-\varsigma), \gamma \leftarrow \gamma \times 0.5$, and $\boldsymbol{r} \leftarrow \hat{\boldsymbol{x}}^{T_u+1}$, where $0 < \varsigma < 1$. When $\varsigma$ is close to 0, e.g., $1e{-}16$, its effect on the empirical convergence rate of RTCFR$^+$ is minimal (Figure 3). Nonetheless, it ensures that the sequence of NEs for the perturbed regularized EFGs converges to the set of NEs of the original EFG, even the exact $\hat{\boldsymbol{x}}^{*,\gamma,\mu,\boldsymbol{r}}$ is not learned.

**Discussions.** Firstly, to the best of our knowledge, we provide the first parameter-free last-iterate convergence for RM-based

---

**Algorithm 1** RTCFR$^+$

1: **Input:** $N, T_u, \mu, \gamma, \boldsymbol{r}$
2: $\boldsymbol{\theta}_I^1 \leftarrow \mathbf{0}, \eta \leftarrow 1, \forall I \in \mathcal{I}$
3: **for** each $n \in [1, 2, \cdots, N]$ **do**
4:     Build the perturbed regularized EFGs in Eq. (2) via $\mu, \gamma$, and $\boldsymbol{r}$
5:     **for** each $t \in [1, 2, \cdots, T_u]$ **do**
6:         Obtain $\hat{\boldsymbol{x}}^{t+1}$ and $\boldsymbol{\theta}_I^{t+1}$ via the update rule in Eq. (3)
7:     **end for**
8:     $\gamma \leftarrow \gamma * 0.5, \boldsymbol{r} \leftarrow \hat{\boldsymbol{x}}^{T_u+1}$
9:     $\boldsymbol{\theta}_I^1 \leftarrow \boldsymbol{\theta}_I^{T_u+1}, \forall I \in \mathcal{I}$
10: **end for**
11: **Return** $\hat{\boldsymbol{x}}^{T_u+1}$

---

CFR algorithms in learning an NE of perturbed regularized EFGs. When considering NFGs, the last-iterate convergence result of CFR$^+$ (RM$^+$) holds even when $\gamma = 0$, due to that the smoothness of counterfactual values and Lemma 4.4 hold in NFGs with any $\gamma \geq 0$. Secondly, we extend the parameter-free results of CFR$^+$ from Farina et al. [2021], demonstrating that CFR$^+$ converges with the parameter-free property for any $\boldsymbol{\theta}_I^1 \in \mathbb{R}^{|A(I)|}_{\geq 0}$, not just when $\boldsymbol{\theta}_I^1 = \mathbf{0}$ in Farina et al. [2021]. This new parameter-free result is significant. Specifically, it indicates that after updating $\gamma$ and $\boldsymbol{r}$ (line 8 of Algorithm 1), there is no need to reset $\boldsymbol{\theta}_I^1$ to $\mathbf{0}$ to get the parameter-free property (line 9 of Algorithm 1). This improves the stability of CFR$^+$, i.e., rapid fluctuations in the strategy profiles across iterations, since such stability improves as the lower bound of the 1-norm of $\boldsymbol{\theta}_I^t$ increases [Farina et al., 2023] (for CFR$^+$, from the proof of Lemma C.2 of Liu et al. [2022], we get that $\|\boldsymbol{\theta}_I^t\|_2 \leq \|\boldsymbol{\theta}_I^{t+1}\|_2$, and the 1-norm lower bound is related to the 2-norm lower bound). Notably, as shown in Appendix G, resetting $\boldsymbol{\theta}_I^1$ to $\mathbf{0}$ after updating $\gamma$ and $\boldsymbol{r}$ (line 9 of Algorithm 1 becomes $\boldsymbol{\theta}_I^1 \leftarrow \mathbf{0}, \forall I \in \mathcal{I}$) causes RTCFR$^+$ to never converge (Figure 3)! Lastly, our proof approach for the parameter-free property can be used to show that CFR$^+$'s average-iterate convergence holds for all $\boldsymbol{\theta}_I^1 \in \mathbb{R}^{|A(I)|}_{\geq 0}$ and $\eta > 0$. As our primary focus is on last-iterate convergence, we discuss the parameter-free average-iterate convergence in Appendix F rather than the main text.

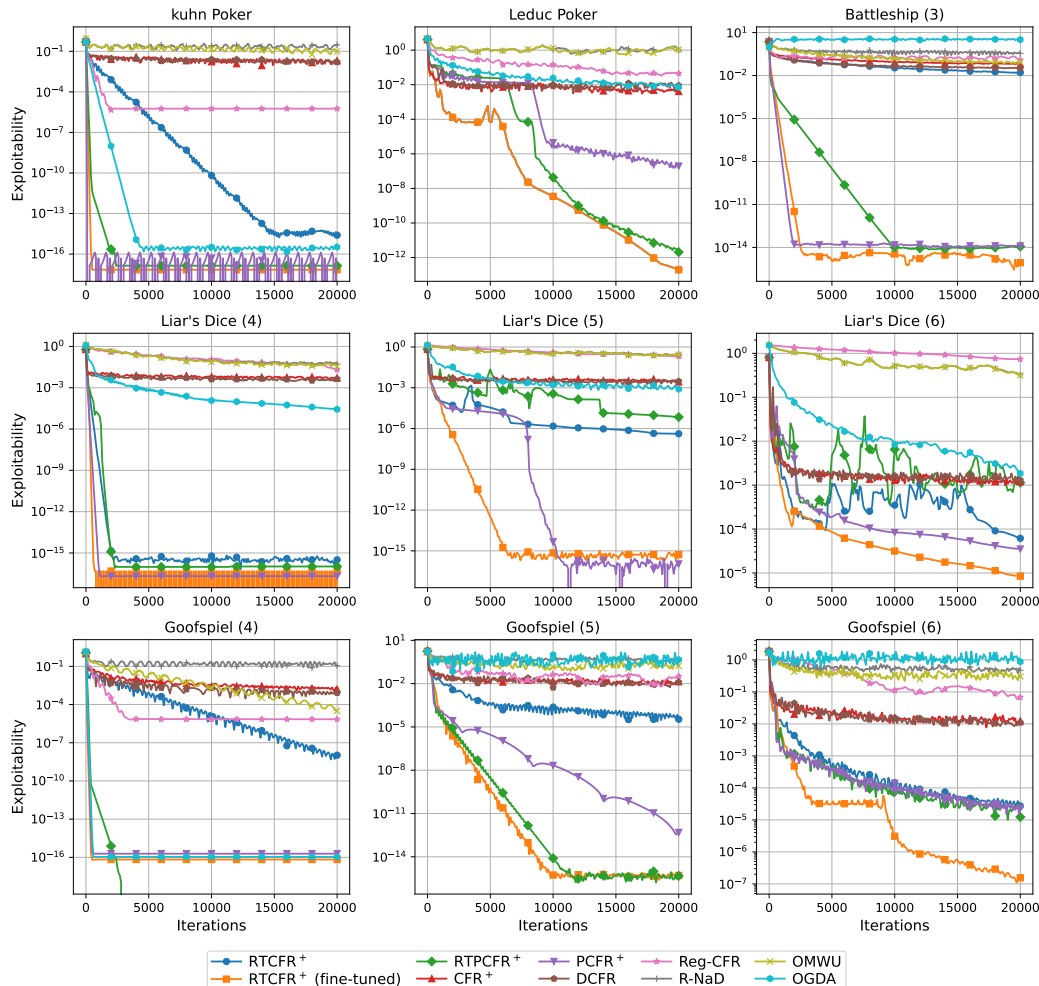

Figure 1: Last-iterate convergence rates of different algorithms. In all plots, the x-axis is the number of iteration, and the y-axis is exploitability, displayed on a logarithmic scale. Liar's Dice ($x$) represents that every player is given a die with $x$ sides. Goofspiel ($x$) denotes that each player is dealt $x$ cards. Battleship ($x$) implies the size of grids is $x$. The size of the tested games is in Appendix G (Table 2).

## 5 Experiments

**Configurations.** We now evaluate the empirical convergence rate of RTCFR$^+$ on five standard EFG benchmarks: Kuhn Poker, Leduc Poker, Goofspiel, Liar's Dice, and Battleship, all implemented using OpenSpiel [Lanctot et al., 2019]. We compare RTCFR$^+$ with classical CFR algorithms, such as CFR$^+$, PCFR$^+$ [Farina et al., 2021], and DCFR [Brown and Sandholm, 2019a], and those with theoretical guarantees for last-iterate convergence, including R-NaD [Pérolat et al., 2021, 2022] and Reg-CFR [Liu et al., 2023]. Additionally, we evaluate traditional last-iterate convergence algorithms, such as OMWU and OGDA [Wei et al., 2021, Lee et al., 2021]. The algorithm implementations are based on the open-source LiteEFG code [Liu et al., 2024], which offers a significant speedup—approximately 100 times faster than OpenSpiel's default implementation for the same number of iterations. For RTCFR$^+$, we set the initial values of $\eta$, $\gamma$, and $\mu$ to 1, 1e$-$10, and 1e$-$3, respectively. The number of iterations $T_u$ required to update $\gamma$ and $r$, is set to 100. For Reg-CFR, we use the parameters from the original paper. For R-NaD, we initialize $\mu = $ 1e$-$5 (R-NaD does not include the parameter $\gamma$), set $T_u = 1000$, and use a learning rate of $\eta = 0.1$. For OMWU and OGDA, we set $\eta$ to 0.5 and 0.1, respectively. All algorithms employ alternating updates to enhance empirical convergence rates. Each algorithm is run for 20,000 ($N = 20000/T_u$) iterations to analyze long-term

Table 1: Hyperparameters used in RTCFR$^+$ (fine-tuned).

| | Kuhn Poker | Leduc Poker | Battleship (3) | Liar's Dice (4) | Liar's Dice (5) |
|---|---|---|---|---|---|
| $\mu$ | 0.1 | 0.001 | 0.1 | 0.01 | 0.0005 |
| $T_u$ | 10 | 100 | 50 | 10 | 10 |
| | Liar's Dice (6) | Goofspiel (4) | Goofspiel (5) | Goofspiel (6) | |
| $\mu$ | 0.0001 | 0.1 | 0.05 | 0.005 | |
| $T_u$ | 500 | 10 | 100 | 50 | |

behavior. The experiments are conducted on a machine equipped with a Xeon(R) Gold 6444Y CPU and 256 GB of memory. More experimental results including (i) performance of RTCFR$^+$ under simultaneous decrease of $\mu$ and $\gamma$, (ii) performance of RTCFR$^+$ under reset accumulated regrets as **0**, (iii) comparison with average-iterate convergence CFR algorithms, (iv) performance of RTCFR$^+$ in HUNL Subgames, and (v) performance of RTCFR$^+$ under different hyperparameters, are in Appendix G.

**Results.** The experimental results are presented in Figure 1. RTCFR$^+$ demonstrates superior performance compared to all other tested algorithms except PCFR$^+$. Specifically, RTCFR$^+$ exhibits the fastest convergence rate across all games when compared to CFR$^+$. In comparison to existing theoretical last-iterate convergence CFR algorithms, such as Reg-CFR and R-NaD, RTCFR$^+$ is only surpassed by Reg-CFR during the initial stages in small-scale games like Kuhn Poker and Goofspiel (4). Similarly, when compared to traditional last-iterate convergence algorithms, RTCFR$^+$ is only outperformed by OGDA in small-scale games such as Kuhn Poker and Goofspiel (4). Inspired by our RTCFR$^+$ and the performance of PCFR$^+$, we propose RTPCFR$^+$, which employs PCFR$^+$ to solve the perturbed regularized EFG defined in Eq. (2) instead of CFR$^+$. For RTPCFR$^+$, we use the same parameters as RTCFR$^+$. Among RTCFR$^+$, RTPCFR$^+$, and PCFR$^+$, no single algorithm consistently outperforms the others across all EFGs, as their performance varies depending on the specific EFG. This variability may be attributed to the fact that RTCFR$^+$ and RTPCFR$^+$ have not been fine-tuned for individual EFGs. Therefore, we also include a comparison with the fine-tuned RTCFR$^+$, which is denoted as RTCFR$^+$ (fine-tuned) in Figure 1. Our findings demonstrate that fine-tuning enables RTCFR$^+$ to outperform all tested algorithms. The parameters used for the fine-tuned RTCFR$^+$ are presented in Table 1. However, the automatic adjustment of $\gamma$, $\mu$, and $T_u$ remains an open problem. One of our future research directions is to investigate the automotive adjustment of these parameters.

## 6 Conclusions

We explore the last-iterate convergence of parameter-free RM-based CFR algorithms. We establish that a classical parameter-free RM-based CFR algorithm, CFR$^+$, achieves last-iterate convergence in learning an NE of perturbed regularized EFGs. To our knowledge, this is the first parameter-free last-iterate convergence of RM-based CFR algorithms in perturbed regularized EFGs. Experimental results show that our proposed algorithm, RTCFR$^+$, exhibits a significantly faster empirical convergence rate than existing algorithms that achieve theoretical last-iterate convergence.

**Limitations.** The main limitation of RTCFR$^+$ is its dependency on parameter tuning. Specifically, RTCFR$^+$ requires careful fine-tuning of parameters $\mu$, $\gamma$, and $T_u$, which prevents it from being a parameter-free algorithm. Interestingly, when both $\mu$ and $\gamma$ are simultaneously reduced, RTCFR$^+$ achieves last-iterate convergence in learning an NE of the original EFGs, irrespective of the values of $\mu$, $\gamma$, and $T_u$. These parameters only impact the empirical convergence rate. Therefore, advancing automated methods to learn optimal values for $\mu$, $\gamma$, and $T_u$ represents a promising direction for future research.

## Acknowledgements

This work is supported in part by the National Natural Science Foundation of China under Grants 62192783 and 62506157, the Jiangsu Science and Technology Major Project BG2024031, the Fundamental Research Funds for the Central Universities (14380128), the Collaborative Innovation Center of Novel Software Technology and Industrialization, and the InnoHK funding.

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

## A    An Example of Extensive-Form Games

To illustrate the components of an EFG, we provide an example using the classic game of Matching Pennies, as depicted in its game tree representation in Figure 2. As shown in Section 2, an EFG is formally defined by the tuple $G = \{\mathcal{N}, \mathcal{H}, P, \mathcal{A}, \mathcal{I}, \{u_i\}\}$. In this example, the set of players is $\mathcal{N} = \{0, 1\}$. The game commences at the root of the tree, which corresponds to the empty history $\varnothing \in \mathcal{H}$. The player function $P(h)$ determines who moves at history $h$; here, $P(\varnothing) = 0$, so the player 0 makes the first move. The actions available to the player 0 at this initial decision node are given by $\mathcal{A}(\varnothing) = \{\text{heads, tails}\}$.

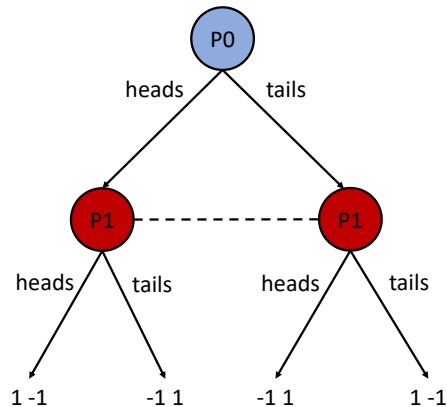

Figure 2: A classical EFG: Matching pennies games. "P0" and "P1" represents the player 0 and 1, respectively

Once the player 0 chooses an action, the game transitions to a new history. For instance, if the player 0 chooses "heads", the new history becomes (P0:heads). At this stage, the player function dictates that it is the player 1's turn to act, i.e., $P(\text{P0:heads}) = P(\text{P0:tails}) = 1$. A central concept in EFGs for modeling games with hidden information is the partition of each player's decision nodes into information sets $\mathcal{I}$. In Figure 2, the dashed line connecting the player 1's two decision nodes signifies that they belong to the same infoset. This means that when the player 1 makes a choice, they are unaware of the player 0's preceding move; the histories (P0:heads) and (P0:tails) are indistinguishable to the player 1. A formal requirement is that the set of available actions must be identical for all nodes within an information set, which holds true here as the available actions are $\mathcal{A}(\text{P0:heads}) = \mathcal{A}(\text{P0:tails}) = \{\text{heads, tails}\}$.

After the player 1 selects an action, the game concludes, reaching a terminal history, also known as a leaf node $z \in \mathcal{Z}$. Each leaf node is associated with a payoff vector that specifies the utility for each player, $(u_0(z), u_1(z))$. For example, if the sequence of actions is (heads, heads), the game terminates with the payoff vector $(1, -1)$, indicating a gain of 1 for the player 0 and a loss of 1 for the player 1. Conversely, if the coins do not match, as in the history (heads, tails), the payoff is $(-1, 1)$. Since for any terminal history $z$, the payoffs for the player 0 and the player 1 are structured such that $u_0(z) = -u_1(z)$, this particular EFG is classified as a two-player, zero-sum game. This single example effectively demonstrates how an EFG captures the sequential structure, information constraints, and outcomes of a strategic interaction.

## B    Related Work

**Counterfactual Regret Minimization (CFR) algorithms.** CFR algorithms are among the most widely used methods for solving real-world EFGs [Bowling et al., 2015, Moravčík et al., 2017, Brown and Sandholm, 2018, 2019b, Pérolat et al., 2022]. The core idea of CFR is to decompose the problem of regret minimization across the entire game into subproblems within each infoset, employing a regret minimization algorithm as a local regret minimizer. The vanilla CFR algorithm was introduced by Zinkevich et al. [2007], which utilize RM [Hart and Mas-Colell, 2000] as the local regret minimizer. To enhance the performance of CFR, a common approach is to design more effective local regret minimizers, as the choice of local regret minimizer largely determines the overall CFR algorithm's efficiency. Advanced local regret minimizers are typically based on RM, including RM$^+$ [Tammelin, 2014], Discounted RM (DRM) [Brown and Sandholm, 2019a], and Predictive RM$^+$ (PRM$^+$) [Farina et al., 2021], which correspond to CFR$^+$ [Tammelin, 2014], Discounted CFR (DCFR) [Brown and Sandholm, 2019a], and Predictive CFR$^+$ (PCFR$^+$) [Farina et al., 2021], respectively. However, CFR algorithms typically achieve theoretical convergence to the set of NEs of EFGs only through the average of iterates, also be called as average-iterate convergence.

**Last-iterate convergence results of CFR algorithms.** Pérolat et al. [2021] provide the first last-iterate convergence result for CFR algorithms in learning an NE of EFGs by transforming the task of learning an NE of the original EFG into finding the NEs of a sequence of regularized EFGs and

ensuring the sequence of the NEs of these regularized EFGs converges to the set of NEs of the original EFG. However, their analysis assumes continuous-time feedback, a condition rarely satisfied in practical scenarios. Subsequently, Liu et al. [2023] presents the first last-iterate convergence result for CFR under the discrete-time feedback by transforming the task of learning an NE of the original EFG into finding the NEs of a sequence of perturbed regularized EFGs rather than only regularized EFGs, since the addition of perturbation introduces the smoothness of counterfactual values. Nevertheless, both algorithms do not leverage RM algorithms as the local regret minimizer, leading to a suboptimal empirical last-iterate convergence rate compared to traditional RM-based CFR algorithms that only achieve average-iterate convergence, as demonstrated in our experiments.

**Last-iterate convergence results of RM algorithms.** Except this paper, Cai et al. [2025], Meng et al. [2025] also investigate the last-iterate convergence of RM algorithms. However, their results mainly focus on non-parameter-free RM algorithms, whereas we considers parameter-free RM algorithms. Specifically, Cai et al. [2025], Meng et al. [2025] mainly investigate smooth $RM^+$ variants [Farina et al., 2023]. The lack of the parameter-free property in the results of Cai et al. [2025], Meng et al. [2025] makes them less applicable when solving real-world games. Although Cai et al. [2025] investigate $RM^+$ ($CFR^+$ uses $RM^+$ as the local regret minimizer), a parameter-free RM algorithm, their proof techniques related to $RM^+$ primarily follow our proof techniques. Furthermore, the results in Cai et al. [2025], Meng et al. [2025] are confined to NFGs, whereas we focus on EFGs.

We establish the first parameter-free last-iterate convergence for RM-based CFR algorithms in learning an NE of perturbed regularized EFGs. Notably, our parameter-free property holds for any initial accumulated counterfactual regrets not only the zero initialization in previous works [Farina et al., 2021]. While $CFR^+$'s parameter-free property in its first theoretical convergence result [Tammelin et al., 2015] holds for any initial accumulated counterfactual regrets, this result is exclusively limited to average-iterate convergence. In contrast, our proof technique simultaneously establishes both parameter-free last-iterate (Theorem 4.1) and average-iterate convergence (Theorem F.1) for $CFR^+$ under any initial accumulated counterfactual regrets[2]. Notably, the proof techniques employed by Tammelin et al. [2015] differ fundamentally from those utilized in ours and most recent works on RM-based CFR algorithms [Farina et al., 2023, Xu et al., 2022, 2024a,b, Zhang et al., 2024]. These works, including ours, adopt the Blackwell approachability framework (as introduced in Section 2) in Farina et al. [2021] to prove the convergence of RM-based CFR algorithms, while Tammelin et al. [2015] use the potential function [Zhang et al., 2022a]. Unfortunately, as previously mentioned, the parameter-free property in Farina et al. [2021] (even including Farina et al. [2023], Xu et al. [2022, 2024a,b], Zhang et al. [2024]) holds only under the condition where the initial accumulated counterfactual regrets are zero. Lastly, experiments show that our algorithm, $RTCFR^+$, substantially outperform existing algorithms that achieve theoretical last-iterate convergence.

In this paper, we only focus on the last-iterate convergence and do not consider the best-iterate convergence because it offers limited utility in real-world games [Anagnostides et al., 2024, Wang et al., 2023]. With the best-iterate convergence, computing the exploitability of each iteration's strategy profile is necessary to select an optimal strategy, but this task is typically challenging due to the vast size of real-world games, such as HUNL, which reaches a size of $10^{170}$. In contrast, the last-iterate convergence circumvents the need to compute exploitability for every iteration; it simply requires the selection of the strategy from the final iteration.

## C   Discussion on the Application of $RTCFR^+$ in Large-Scale Games and Its Integration with Other Technologies

Firstly, $RTCFR^+$ can be directly applied to large-scale games without any modifications. In fact, the modifications introduced by $RTCFR^+$ over CFR+ are minimal. As demonstrated in our implementation provided in Appendix F, $RTCFR^+$ requires fewer than 30 additional lines compared to $CFR^+$ (specifically, lines 33, 40–41, 47–49, 51-55, and 62–66 of the $RTCFR^+$ implementation in Appendix H). The main limitation of applying $RTCFR^+$ to large-scale games lies in the need to tune the hyperparameters $\mu$, $\gamma$, and $T_u$, which can vary significantly across different games. Addressing the dependency on tuning $\mu$, $\gamma$, and $T_u$ remains a central direction for future work. It is important to clarify, however, that this requirement originates from the RT framework itself; all existing algorithms based on the RT framework require tuning of these parameters.

---

[2]Farina et al. [2021] also only establish parameter-free average-iterate convergence.

Secondly, integrating RTCFR$^+$ with the other technologies requires case-by-case analysis. (i) For algorithms that solely modify the game tree, such as depth-limited solving [Brown et al., 2018, 2020], impact-recall abstraction [Ganzfried and Sandholm, 2014], action abstraction [Li et al., 2024], and Vector CFR [Johanson et al., 2012], RTCFR$^+$ can be directly applied since RTCFR$^+$ only requires execution on the new game tree. This process is straightforward and presents no significant challenges. (ii) Regarding warm-start [Brown and Sandholm, 2016], while its concept of setting initial accumulated counterfactual regrets using an efficient initial strategy is insightful, current integration with RTCFR$^+$ is not feasible. Specifically, the warm-start approach in Brown and Sandholm [2016] is an enhancement tailored for the original CFR. Formally, the analysis presented on the bottom left of page four in Brown and Sandholm [2016] demonstrates that the substitute regret is given by $R'^T(I, a) = T(v'^\sigma(I, a) - v'^\sigma(I))$. This formulation implies that $R'^T(I, a)$ can be negative, a property that does not hold in CFR$^+$ and RTCFR$^+$. (iii) As for sparsification [Farina and Sandholm, 2022], which optimizes the computation of loss gradients ($\ell_i^t$, the last line of Eq. (3)), RTCFR$^+$ can seamlessly integrate. This compatibility arises because RTCFR$^+$ solely requires the input of loss gradients, which then facilitates strategy updates through the update rules defined in the first four lines of Eq. (3). (iv) The pruning approach in Li and Huang [2025] can be directly integrated with RTCFR$^+$. Since this pruning approach modifies the game tree before the algorithm execution (e.g., "permanently and correctly eliminating sub-optimal branches before the CFR begins"), it aligns with our earlier statement on game-tree modification approaches. Hence, RTCFR$^+$ can be directly applied.

## D  Proof of Theorem 4.1

*Proof.* To prove the last-iterate convergence of CFR$^+$ in learning an NE of perturbed regularized EFGs defined in Eq. (2), we introduce the following lemmas.

**Lemma D.1** (Adapted from Lemma D.4 in Sokota et al. [2023]). *For any $\boldsymbol{x} \in \mathcal{X}$, $\mu \geq 0$, and $\gamma \geq 0$,*

$$\sum_{i \in \mathcal{N}} \langle \boldsymbol{\ell}_i^{\boldsymbol{x}}, \boldsymbol{x}_i - \boldsymbol{x}_i^{*,\mu,\gamma,\boldsymbol{r}} \rangle \geq \sum_{i \in \mathcal{N}} \langle \boldsymbol{\ell}_i^{\boldsymbol{x}} - \boldsymbol{\ell}_i^{\boldsymbol{x}^{*,\mu,\gamma,\boldsymbol{r}}}, \boldsymbol{x}_i - \boldsymbol{x}_i^{*,\mu,\gamma,\boldsymbol{r}} \rangle \geq \mu \|\boldsymbol{x} - \boldsymbol{x}^{*,\mu,\gamma,\boldsymbol{r}}\|_2^2,$$

*where $\boldsymbol{\ell}_0^{\boldsymbol{x}} = \boldsymbol{A}\boldsymbol{x}_1 + \mu\nabla\psi(\boldsymbol{x}_0) - \mu\nabla\psi(\boldsymbol{r}_0)$ and $\boldsymbol{\ell}_1^{\boldsymbol{x}} = -\boldsymbol{A}^T\boldsymbol{x}_0 + \mu\nabla\psi(\boldsymbol{x}_1) - \mu\nabla\psi(\boldsymbol{r}_1)$.*

**Lemma D.2** (Proof is in Appendix E.3). *For any $\boldsymbol{x} \in \mathcal{X}$, $i \in \mathcal{N}$, $I \in \mathcal{I}_i$, $\mu \geq 0$, and $\gamma \geq 0$,*

$$\|\hat{\boldsymbol{v}}_i^\sigma(I)\|_2 \leq \|\hat{\boldsymbol{v}}_i^\sigma(I)\|_1 \leq P + 2\mu D$$

*where $\hat{\boldsymbol{v}}_i^\sigma(I) = [\hat{v}_i^\sigma(I, a)|a \in A(I)]$, $\hat{v}_i^\sigma(I, a) = -\hat{\ell}_i^{\boldsymbol{x}} + \sum_{I' \in C_i(I,a)} \langle \hat{\boldsymbol{v}}_i^\sigma(I'), \sigma_i(I') \rangle$ with $\hat{\ell}_0^{\boldsymbol{x}} = \boldsymbol{A}\boldsymbol{x}_1 + \mu\nabla\psi(\boldsymbol{x}_0) - \mu\nabla\psi(\boldsymbol{r}_0)$ and $\hat{\ell}_1^{\boldsymbol{x}} = -\boldsymbol{A}^T\boldsymbol{x}_0 + \mu\nabla\psi(\boldsymbol{x}_1) - \mu\nabla\psi(\boldsymbol{r}_1)$, as well as $\sigma$ is the behavioral strategy profile associated with $\boldsymbol{x}$.*

**Lemma D.3** (Proof is in Appendix E.4). *For any $\boldsymbol{x}, \boldsymbol{x}' \in \mathcal{X}$, $i \in \mathcal{N}$, $I \in \mathcal{I}_i$, $\mu \geq 0$, and $\gamma \geq 0$,*

$$\|\hat{\boldsymbol{v}}_i^\sigma(I) - \hat{\boldsymbol{v}}_i^{\sigma'}(I)\|_2 \leq 2(L+\mu)^2\|\boldsymbol{x} - \boldsymbol{x}'\|_1^2 + 2(P + 2\mu D)^2\|\sigma_i - \sigma_i'\|_1^2,$$

*where $\hat{\boldsymbol{v}}_i^\sigma(I) = [\hat{v}_i^\sigma(I, a)|a \in A(I)]$, $\hat{v}_i^\sigma(I, a) = -\hat{\ell}_i^{\boldsymbol{x}} + \sum_{I' \in C_i(I,a)} \langle \hat{\boldsymbol{v}}_i^\sigma(I'), \sigma_i(I') \rangle$ with $\hat{\ell}_0^{\boldsymbol{x}} = \boldsymbol{A}\boldsymbol{x}_1 + \mu\nabla\psi(\boldsymbol{x}_0) - \mu\nabla\psi(\boldsymbol{r}_0)$ and $\hat{\ell}_1^{\boldsymbol{x}} = -\boldsymbol{A}^T\boldsymbol{x}_0 + \mu\nabla\psi(\boldsymbol{x}_1) - \mu\nabla\psi(\boldsymbol{r}_1)$, as well as $\sigma$ and $\sigma'$ are the behavioral strategy profiles associated with $\boldsymbol{x}$ and $\boldsymbol{x}'$, respectively.*

**Lemma D.4** (Proof is in Appendix E.5). *For any $\hat{\boldsymbol{x}}, \hat{\boldsymbol{x}}' \in \mathcal{X}^\gamma$ with $\gamma > 0$, $i \in \mathcal{N}$, $I \in \mathcal{I}_i$, and $\mu \geq 0$,*

$$\|\hat{\sigma}_i - \hat{\sigma}_i'\|_1 \leq \frac{A_{max}C_{max} + 1}{\gamma^H}\|\hat{\boldsymbol{x}}_i - \hat{\boldsymbol{x}}_i'\|_1,$$

*where $\hat{\sigma}$ and $\hat{\sigma}'$ are the behavioral strategy profiles associated with $\hat{\boldsymbol{x}}$ and $\hat{\boldsymbol{x}}'$, respectively.*

By substituting $\boldsymbol{\theta}_I = \sigma_i^{*,\mu,\gamma,\boldsymbol{r}}(I) = \frac{\hat{\sigma}_i^{*,\mu,\gamma,\boldsymbol{r}}(I) - \gamma\boldsymbol{1}}{1 - \alpha_I}$ into Lemma 4.2, we get

$$\eta\langle \hat{\boldsymbol{m}}_i^t(I), \sigma_i^{*,\mu,\gamma,\boldsymbol{r}}(I) - \boldsymbol{\theta}_I^{t+1} \rangle \leq D_\psi(\sigma_i^{*,\mu,\gamma,\boldsymbol{r}}(I), \boldsymbol{\theta}_I^t) - D_\psi(\sigma_i^{*,\mu,\gamma,\boldsymbol{r}}(I), \boldsymbol{\theta}_I^{t+1}) - D_\psi(\boldsymbol{\theta}_I^{t+1}, \boldsymbol{\theta}_I^t).$$
(7)

Adding $\eta\langle -\hat{\boldsymbol{m}}_i^{*,\mu,\gamma,\boldsymbol{r}}(I), \boldsymbol{\theta}_I^{t+1} - \boldsymbol{\theta}_I^t \rangle$ to each hand side of Eq. (7), we have

$$\eta\langle \hat{\boldsymbol{m}}_i^t(I), \sigma_i^{*,\mu,\gamma,\boldsymbol{r}}(I) - \boldsymbol{\theta}_I^{t+1} \rangle + \eta\langle -\hat{\boldsymbol{m}}_i^{*,\mu,\gamma,\boldsymbol{r}}(I), \boldsymbol{\theta}_I^{t+1} - \boldsymbol{\theta}_I^t \rangle$$
$$\leq D_\psi(\sigma_i^{*,\mu,\gamma,\boldsymbol{r}}(I), \boldsymbol{\theta}_I^t) - D_\psi(\sigma_i^{*,\mu,\gamma,\boldsymbol{r}}(I), \boldsymbol{\theta}_I^{t+1}) + \eta\langle -\hat{\boldsymbol{m}}_i^{*,\mu,\gamma,\boldsymbol{r}}(I), \boldsymbol{\theta}_I^{t+1} - \boldsymbol{\theta}_I^t \rangle - D_\psi(\boldsymbol{\theta}_I^{t+1}, \boldsymbol{\theta}_I^t),$$

which implies

$$\eta\langle \hat{\boldsymbol{m}}_i^t(I), \sigma_i^{*,\mu,\gamma,\boldsymbol{r}}(I) - \boldsymbol{\theta}_I^t\rangle$$

$$\leq D_\psi(\sigma_i^{*,\mu,\gamma,\boldsymbol{r}}(I),\boldsymbol{\theta}_I^t) + \eta\langle -\hat{\boldsymbol{m}}_i^{*,\mu,\gamma,\boldsymbol{r}}(I), \boldsymbol{\theta}_I^t\rangle - D_\psi(\sigma_i^{*,\mu,\gamma,\boldsymbol{r}}(I),\boldsymbol{\theta}_I^{t+1}) - \eta\langle -\hat{\boldsymbol{m}}_i^{*,\mu,\gamma,\boldsymbol{r}}(I), \boldsymbol{\theta}_I^{t+1}\rangle$$

$$+ \eta\langle \hat{\boldsymbol{m}}_i^t(I) - \hat{\boldsymbol{m}}_i^{*,\mu,\gamma,\boldsymbol{r}}(I), \boldsymbol{\theta}_I^{t+1} - \boldsymbol{\theta}_I^t\rangle - D_\psi(\boldsymbol{\theta}_I^{t+1}, \boldsymbol{\theta}_I^t)$$

$$\leq D_\psi(\sigma_i^{*,\mu,\gamma,\boldsymbol{r}}(I),\boldsymbol{\theta}_I^t) + \eta\langle -\hat{\boldsymbol{m}}_i^{*,\mu,\gamma,\boldsymbol{r}}(I), \boldsymbol{\theta}_I^t\rangle - D_\psi(\sigma_i^{*,\mu,\gamma,\boldsymbol{r}}(I),\boldsymbol{\theta}_I^{t+1}) - \eta\langle -\hat{\boldsymbol{m}}_i^{*,\mu,\gamma,\boldsymbol{r}}(I), \boldsymbol{\theta}_I^{t+1}\rangle$$

$$+ \eta^2 \frac{\|\hat{\boldsymbol{m}}_i^t(I) - \hat{\boldsymbol{m}}_i^{*,\mu,\gamma,\boldsymbol{r}}(I)\|_2^2}{2} + \frac{\|\boldsymbol{\theta}_I^{t+1} - \boldsymbol{\theta}_I^t\|_2^2}{2} - D_\psi(\boldsymbol{\theta}_I^{t+1}, \boldsymbol{\theta}_I^t)$$

$$\leq D_\psi(\sigma_i^{*,\mu,\gamma,\boldsymbol{r}}(I),\boldsymbol{\theta}_I^t) + \eta\langle -\hat{\boldsymbol{m}}_i^{*,\mu,\gamma,\boldsymbol{r}}(I), \boldsymbol{\theta}_I^t\rangle - D_\psi(\sigma_i^{*,\mu,\gamma,\boldsymbol{r}}(I),\boldsymbol{\theta}_I^{t+1}) - \eta\langle -\hat{\boldsymbol{m}}_i^{*,\mu,\gamma,\boldsymbol{r}}(I), \boldsymbol{\theta}_I^{t+1}\rangle$$

$$+ \eta^2 \frac{\|\hat{\boldsymbol{m}}_i^t(I) - \hat{\boldsymbol{m}}_i^{*,\mu,\gamma,\boldsymbol{r}}(I)\|_2^2}{2}, \tag{8}$$

where the second inequality comes from that $\forall \boldsymbol{a}, \boldsymbol{b} \in \mathbb{R}^d, \rho > 0, \langle \boldsymbol{a}, \boldsymbol{b}\rangle \leq \rho\|\boldsymbol{a}\|_2^2/2 + \|\boldsymbol{b}\|_2^2/(2\rho)$ (in this case, $\boldsymbol{a} = \hat{\boldsymbol{m}}_i^t(I) - \hat{\boldsymbol{m}}_i^{*,\mu,\gamma,\boldsymbol{r}}(I)$, $\boldsymbol{b} = \boldsymbol{\theta}_I^{t+1} - \boldsymbol{\theta}_I^t$, and $\rho = \eta$), and the last inequality is from that $\forall \boldsymbol{a}, \boldsymbol{b} \in \mathbb{R}^d, \|\boldsymbol{a} - \boldsymbol{b}\|_2^2/2 = \|\boldsymbol{b} - \boldsymbol{a}\|_2^2/2 = D_\psi(\boldsymbol{a}, \boldsymbol{b})$ (in this case, $\boldsymbol{a} = \boldsymbol{\theta}_I^{t+1}$, and $\boldsymbol{b} = \boldsymbol{\theta}_I^t$).

Arranging the terms in Eq. (8), we get

$$\eta\langle \hat{\boldsymbol{m}}_i^t(I), \sigma_i^{*,\mu,\gamma,\boldsymbol{r}}(I) - \boldsymbol{\theta}_I^t\rangle - \eta^2 \frac{\|\hat{\boldsymbol{m}}_i^t(I) - \hat{\boldsymbol{m}}_i^{*,\mu,\gamma,\boldsymbol{r}}(I)\|_2^2}{2}$$

$$\leq D_\psi(\sigma_i^{*,\mu,\gamma,\boldsymbol{r}}(I),\boldsymbol{\theta}_I^t) + \eta\langle -\hat{\boldsymbol{m}}_i^{*,\mu,\gamma,\boldsymbol{r}}(I), \boldsymbol{\theta}_I^t\rangle - D_\psi(\sigma_i^{*,\mu,\gamma,\boldsymbol{r}}(I),\boldsymbol{\theta}_I^{t+1}) - \eta\langle -\hat{\boldsymbol{m}}_i^{*,\mu,\gamma,\boldsymbol{r}}(I), \boldsymbol{\theta}_I^{t+1}\rangle.$$

According to the definition of $\hat{\boldsymbol{m}}_i^t(I)$, we have

$$\langle \hat{\boldsymbol{m}}_i^t(I), \sigma_i^{*,\mu,\gamma,\boldsymbol{r}}(I) - \boldsymbol{\theta}_I^t\rangle = \langle \hat{\boldsymbol{v}}_i^t(I) - \langle \hat{\boldsymbol{v}}_i^t(I), \sigma_i^t(I)\rangle \mathbf{1}, \sigma_i^{*,\mu,\gamma,\boldsymbol{r}}(I) - \boldsymbol{\theta}_I^t\rangle$$

$$= \langle -\hat{\boldsymbol{v}}_i^t(I), \sigma_i^t(I) - \sigma_i^{*,\mu,\gamma,\boldsymbol{r}}(I)\rangle,$$

where the second equality comes from that

$$\langle \hat{\boldsymbol{v}}_i^t(I) - \langle \hat{\boldsymbol{v}}_i^t(I), \sigma_i^t(I)\rangle \mathbf{1}, \boldsymbol{\theta}_I^t\rangle = \langle \hat{\boldsymbol{v}}_i^t(I) - \langle \hat{\boldsymbol{v}}_i^t(I), \frac{\boldsymbol{\theta}_I^t}{\langle \boldsymbol{\theta}_I^t, \mathbf{1}\rangle}\rangle \mathbf{1}, \boldsymbol{\theta}_I^t\rangle = 0,$$

$$\langle \langle \hat{\boldsymbol{v}}_i^t(I), \sigma_i^t(I)\rangle \mathbf{1}, \sigma_i^{*,\mu,\gamma,\boldsymbol{r}}(I)\rangle = \langle \hat{\boldsymbol{v}}_i^t(I), \sigma_i^t(I)\rangle.$$

Therefore, we have

$$\eta\langle -\hat{\boldsymbol{v}}_i^t(I), \sigma_i^t(I) - \sigma_i^{*,\mu,\gamma,\boldsymbol{r}}(I)\rangle - \eta^2 \frac{\|\hat{\boldsymbol{m}}_i^t(I) - \hat{\boldsymbol{m}}_i^{*,\mu,\gamma,\boldsymbol{r}}(I)\|_2^2}{2}$$

$$\leq D_\psi(\sigma_i^{*,\mu,\gamma,\boldsymbol{r}}(I),\boldsymbol{\theta}_I^t) + \eta\langle -\hat{\boldsymbol{m}}_i^{*,\mu,\gamma,\boldsymbol{r}}(I), \boldsymbol{\theta}_I^t\rangle - D_\psi(\sigma_i^{*,\mu,\gamma,\boldsymbol{r}}(I),\boldsymbol{\theta}_I^{t+1}) - \eta\langle -\hat{\boldsymbol{m}}_i^{*,\mu,\gamma,\boldsymbol{r}}(I), \boldsymbol{\theta}_I^{t+1}\rangle. \tag{9}$$

Let $\beta_I = \pi_i^{\hat{\sigma}^{*,\mu,\gamma,\boldsymbol{r}}}(I)$. Continuing from Eq. (9), we get

$$\eta\beta_I\langle -\hat{\boldsymbol{v}}_i^t(I), (1-\alpha_I)\sigma_i^t(I) - (1-\alpha_I)\sigma_i^{*,\mu,\gamma,\boldsymbol{r}}(I)\rangle - \eta^2(1-\alpha_I)\beta_I \frac{\|\hat{\boldsymbol{m}}_i^t(I) - \hat{\boldsymbol{m}}_i^{*,\mu,\gamma,\boldsymbol{r}}(I)\|_2^2}{2}$$

$$\leq (1-\alpha_I)\beta_I\left( D_\psi(\sigma_i^{*,\mu,\gamma,\boldsymbol{r}}(I),\boldsymbol{\theta}_I^t) + \eta\langle -\hat{\boldsymbol{m}}_i^{*,\mu,\gamma,\boldsymbol{r}}(I), \boldsymbol{\theta}_I^t\rangle - D_\psi(\sigma_i^{*,\mu,\gamma,\boldsymbol{r}}(I),\boldsymbol{\theta}_I^{t+1}) - \eta\langle -\hat{\boldsymbol{m}}_i^{*,\mu,\gamma,\boldsymbol{r}}(I), \boldsymbol{\theta}_I^{t+1}\rangle\right)$$

$$\Rightarrow \eta\beta_I\langle -\hat{\boldsymbol{v}}_i^t(I), (1-\alpha_I)\sigma_i^t(I) + \gamma\mathbf{1} - (1-\alpha_I)\sigma_i^{*,\mu,\gamma,\boldsymbol{r}}(I) - \gamma\mathbf{1}\rangle - \eta^2(1-\alpha_I)\beta_I \frac{\|\hat{\boldsymbol{m}}_i^t(I) - \hat{\boldsymbol{m}}_i^{*,\mu,\gamma,\boldsymbol{r}}(I)\|_2^2}{2}$$

$$\leq (1-\alpha_I)\beta_I\left( D_\psi(\sigma_i^{*,\mu,\gamma,\boldsymbol{r}}(I),\boldsymbol{\theta}_I^t) + \eta\langle -\hat{\boldsymbol{m}}_i^{*,\mu,\gamma,\boldsymbol{r}}(I), \boldsymbol{\theta}_I^t\rangle - D_\psi(\sigma_i^{*,\mu,\gamma,\boldsymbol{r}}(I),\boldsymbol{\theta}_I^{t+1}) - \eta\langle -\hat{\boldsymbol{m}}_i^{*,\mu,\gamma,\boldsymbol{r}}(I), \boldsymbol{\theta}_I^{t+1}\rangle\right)$$

$$\Rightarrow \eta\beta_I\langle -\hat{\boldsymbol{v}}_i^t(I), \hat{\sigma}_i^t(I) - \hat{\sigma}_i^{*,\mu,\gamma,\boldsymbol{r}}(I)\rangle - \eta^2(1-\alpha_I)\beta_I \frac{\|\hat{\boldsymbol{m}}_i^t(I) - \hat{\boldsymbol{m}}_i^{*,\mu,\gamma,\boldsymbol{r}}(I)\|_2^2}{2}$$

$$\leq (1-\alpha_I)\beta_I\left( D_\psi(\sigma_i^{*,\mu,\gamma,\boldsymbol{r}}(I),\boldsymbol{\theta}_I^t) + \eta\langle -\hat{\boldsymbol{m}}_i^{*,\mu,\gamma,\boldsymbol{r}}(I), \boldsymbol{\theta}_I^t\rangle - D_\psi(\sigma_i^{*,\mu,\gamma,\boldsymbol{r}}(I),\boldsymbol{\theta}_I^{t+1}) - \eta\langle -\hat{\boldsymbol{m}}_i^{*,\mu,\gamma,\boldsymbol{r}}(I), \boldsymbol{\theta}_I^{t+1}\rangle\right).$$

By applying Lemma 4.3, we have

$$\eta\sum_{t=1}^T\sum_{i\in\mathcal{N}}\langle \hat{\boldsymbol{\ell}}_i^t, \hat{\boldsymbol{x}}_i^t - \hat{\boldsymbol{x}}_i^{*,\mu,\gamma,\boldsymbol{r}}\rangle - \sum_{t=1}^T\sum_{i\in\mathcal{N}}\sum_{I\in\mathcal{I}_i}\eta^2\zeta_I \frac{\|\hat{\boldsymbol{m}}_i^t(I) - \hat{\boldsymbol{m}}_i^{*,\mu,\gamma,\boldsymbol{r}}(I)\|_2^2}{2}$$

$$\leq \sum_{i\in\mathcal{N}}\sum_{I\in\mathcal{I}_i}\zeta_I\left( D_\psi(\sigma_i^{*,\mu,\gamma,\boldsymbol{r}}(I),\boldsymbol{\theta}_I^1) + \eta\langle -\hat{\boldsymbol{m}}_i^{*,\mu,\gamma,\boldsymbol{r}}(I), \boldsymbol{\theta}_I^1\rangle - D_\psi(\sigma_i^{*,\mu,\gamma,\boldsymbol{r}}(I),\boldsymbol{\theta}_I^{T+1}) - \eta\langle -\hat{\boldsymbol{m}}_i^{*,\mu,\gamma,\boldsymbol{r}}(I), \boldsymbol{\theta}_I^{T+1}\rangle\right),$$

where $\zeta_I = (1 - \alpha_I)\beta_I$. Since $0 \le \zeta_I \le 1$ (as $0 \le \beta_I \le 1$ and $0 \le \alpha_I \le 1$), we get

$$\eta\sum_{t=1}^{T}\sum_{i\in\mathcal{N}}\langle\hat{\boldsymbol{\ell}}_i^t,\hat{\boldsymbol{x}}_i^t-\hat{\boldsymbol{x}}_i^{*,\mu,\gamma,\boldsymbol{r}}\rangle-\sum_{t=1}^{T}\sum_{i\in\mathcal{N}}\sum_{I\in\mathcal{I}_i}\eta^2\frac{\|\hat{\boldsymbol{m}}_i^t(I)-\hat{\boldsymbol{m}}_i^{*,\mu,\gamma,\boldsymbol{r}}(I)\|_2^2}{2}$$

$$\le\sum_{i\in\mathcal{N}}\sum_{I\in\mathcal{I}_i}\zeta_I\big(D_\psi(\sigma_i^{*,\mu,\gamma,\boldsymbol{r}}(I),\boldsymbol{\theta}_I^1)+\eta\langle-\hat{\boldsymbol{m}}_i^{*,\mu,\gamma,\boldsymbol{r}}(I),\boldsymbol{\theta}_I^1\rangle-D_\psi(\sigma_i^{*,\mu,\gamma,\boldsymbol{r}}(I),\boldsymbol{\theta}_I^{T+1})-\eta\langle-\hat{\boldsymbol{m}}_i^{*,\mu,\gamma,\boldsymbol{r}}(I),\boldsymbol{\theta}_I^{T+1}\rangle\big).$$

By applying Lemma D.1, we obtain

$$\sum_{t=1}^{T}\mu\eta\|\hat{\boldsymbol{x}}^t-\hat{\boldsymbol{x}}^{*,\mu,\gamma,\boldsymbol{r}}\|_2^2-\sum_{t=1}^{T}\sum_{i\in\mathcal{N}}\sum_{I\in\mathcal{I}_i}\eta^2\frac{\|\hat{\boldsymbol{m}}_i^t(I)-\hat{\boldsymbol{m}}_i^{*,\mu,\gamma,\boldsymbol{r}}(I)\|_2^2}{2}$$

$$\le\sum_{i\in\mathcal{N}}\sum_{I\in\mathcal{I}_i}\zeta_I\Big(D_\psi(\sigma_i^{*,\mu,\gamma,\boldsymbol{r}}(I),\boldsymbol{\theta}_I^1)+\eta\langle-\hat{\boldsymbol{m}}_i^{*,\mu,\gamma,\boldsymbol{r}}(I),\boldsymbol{\theta}_I^1\rangle-D_\psi(\sigma_i^{*,\mu,\gamma,\boldsymbol{r}}(I),\boldsymbol{\theta}_I^{T+1})-\eta\langle-\hat{\boldsymbol{m}}_i^{*,\mu,\gamma,\boldsymbol{r}}(I),\boldsymbol{\theta}_I^{T+1}\rangle\Big).$$

$$(10)$$

Now, we use the smoothness of the instantaneous counterfactual regrets to transform $\|\hat{\boldsymbol{m}}_i^t(I) - \hat{\boldsymbol{m}}_i^{*,\mu,\gamma,\boldsymbol{r}}(I)\|_2^2$ into a term only related to $\|\hat{\boldsymbol{x}}^t - \hat{\boldsymbol{x}}^{*,\mu,\gamma,\boldsymbol{r}}\|_2^2$. Formally, for the term $\|\hat{\boldsymbol{m}}_i^t(I) - \hat{\boldsymbol{m}}_i^{*,\mu,\gamma,\boldsymbol{r}}(I)\|_2^2$, from the definition of $\hat{\boldsymbol{m}}_i^t(I)$ and $\hat{\boldsymbol{m}}_i^{*,\mu,\gamma,\boldsymbol{r}}(I)$, we have

$$\|\hat{\boldsymbol{m}}_i^t(I)-\hat{\boldsymbol{m}}_i^{*,\mu,\gamma,\boldsymbol{r}}(I)\|_2^2$$

$$=\|\hat{\boldsymbol{v}}_i^t(I)-\langle\hat{\boldsymbol{v}}_i^t(I),\sigma_i^t(I)\rangle\mathbf{1}-\hat{\boldsymbol{v}}_i^{*,\mu,\gamma,\boldsymbol{r}}(I)+\langle\hat{\boldsymbol{v}}_i^{*,\mu,\gamma,\boldsymbol{r}}(I),\sigma_i^{*,\mu,\gamma,\boldsymbol{r}}(I)\rangle\mathbf{1}\|_2^2$$

$$=\|\hat{\boldsymbol{v}}_i^t(I)-\hat{\boldsymbol{v}}_i^{*,\mu,\gamma,\boldsymbol{r}}(I)-\langle\hat{\boldsymbol{v}}_i^t(I),\sigma_i^t(I)\rangle\mathbf{1}+\langle\hat{\boldsymbol{v}}_i^{*,\mu,\gamma,\boldsymbol{r}}(I),\sigma_i^{*,\mu,\gamma,\boldsymbol{r}}(I)\rangle\mathbf{1}\|_2^2$$

$$\le2\|\hat{\boldsymbol{v}}_i^t(I)-\hat{\boldsymbol{v}}_i^{*,\mu,\gamma,\boldsymbol{r}}(I)\|_2^2+2|A(I)|^2\|\langle\hat{\boldsymbol{v}}_i^t(I),\sigma_i^t(I)\rangle-\langle\hat{\boldsymbol{v}}_i^{*,\mu,\gamma,\boldsymbol{r}}(I),\sigma_i^{*,\mu,\gamma,\boldsymbol{r}}(I)\rangle\|_2^2$$

$$\le2\|\hat{\boldsymbol{v}}_i^t(I)-\hat{\boldsymbol{v}}_i^{*,\mu,\gamma,\boldsymbol{r}}(I)\|_2^2$$

$$+2|A(I)|^2\|\langle\hat{\boldsymbol{v}}_i^t(I),\sigma_i^t(I)\rangle-\langle\hat{\boldsymbol{v}}_i^t(I),\sigma_i^{*,\mu,\gamma,\boldsymbol{r}}(I)\rangle+\langle\hat{\boldsymbol{v}}_i^t(I),\sigma_i^{*,\mu,\gamma,\boldsymbol{r}}(I)\rangle-\langle\hat{\boldsymbol{v}}_i^{*,\mu,\gamma,\boldsymbol{r}}(I),\sigma_i^{*,\mu,\gamma,\boldsymbol{r}}(I)\rangle\|_2^2$$

$$\le2\|\hat{\boldsymbol{v}}_i^t(I)-\hat{\boldsymbol{v}}_i^{*,\mu,\gamma,\boldsymbol{r}}(I)\|_2^2+4|A(I)|^2\|\langle\hat{\boldsymbol{v}}_i^t(I),\sigma_i^t(I)\rangle-\langle\hat{\boldsymbol{v}}_i^t(I),\sigma_i^{*,\mu,\gamma,\boldsymbol{r}}(I)\rangle\|_2^2$$

$$+4|A(I)|^2\|\langle\hat{\boldsymbol{v}}_i^t(I),\sigma_i^{*,\mu,\gamma,\boldsymbol{r}}(I)\rangle-\langle\hat{\boldsymbol{v}}_i^{*,\mu,\gamma,\boldsymbol{r}}(I),\sigma_i^{*,\mu,\gamma,\boldsymbol{r}}(I)\rangle\|_2^2,$$

where $\sigma_i^{*,\mu,\gamma,\boldsymbol{r}}(I) = \frac{\hat{\sigma}_i^{*,\mu,\gamma,\boldsymbol{r}}(I)-\gamma\mathbf{1}}{1-\alpha_I}$. By using $A_{max} = \max_{I\in\mathcal{I}}|A(I)|$, we have

$$\|\hat{\boldsymbol{m}}_i^t(I) - \hat{\boldsymbol{m}}_i^{*,\mu,\gamma,\boldsymbol{r}}(I)\|_2^2$$

$$\le2\|\hat{\boldsymbol{v}}_i^t(I) - \hat{\boldsymbol{v}}_i^{*,\mu,\gamma,\boldsymbol{r}}(I)\|_2^2 + 4A_{max}^2\|\langle\hat{\boldsymbol{v}}_i^t(I),\sigma_i^t(I)\rangle - \langle\hat{\boldsymbol{v}}_i^t(I),\sigma_i^{*,\mu,\gamma,\boldsymbol{r}}(I)\rangle\|_2^2 \qquad (11)$$

$$+ 4A_{max}^2\|\langle\hat{\boldsymbol{v}}_i^t(I),\sigma_i^{*,\mu,\gamma,\boldsymbol{r}}(I)\rangle - \langle\hat{\boldsymbol{v}}_i^{*,\mu,\gamma,\boldsymbol{r}}(I),\sigma_i^{*,\mu,\gamma,\boldsymbol{r}}(I)\rangle\|_2^2.$$

For the term $\|\langle\hat{\boldsymbol{v}}_i^t(I),\sigma_i^t(I)\rangle - \langle\hat{\boldsymbol{v}}_i^t(I),\sigma_i^{*,\mu,\gamma,\boldsymbol{r}}(I)\rangle\|_2^2$ in Eq. (11), we have

$$\|\langle\hat{\boldsymbol{v}}_i^t(I),\sigma_i^t(I)\rangle - \langle\hat{\boldsymbol{v}}_i^t(I),\sigma_i^{*,\mu,\gamma,\boldsymbol{r}}(I)\rangle\|_2^2$$

$$=\|\langle\hat{\boldsymbol{v}}_i^t(I),\sigma_i^t(I) - \sigma_i^{*,\mu,\gamma,\boldsymbol{r}}(I)\rangle\|_2^2$$

$$\le\|\hat{\boldsymbol{v}}_i^t(I)\|_2^2\|\sigma_i^t(I) - \sigma_i^{*,\mu,\gamma,\boldsymbol{r}}(I)\|_2^2 \qquad (12)$$

$$\le(P + 2\mu D)^2\|\sigma_i^t(I) - \sigma_i^{*,\mu,\gamma,\boldsymbol{r}}(I)\|_2^2,$$

where the last line is from Lemma D.2. For the term $\|\langle\hat{\boldsymbol{v}}_i^t(I),\sigma_i^{*,\mu,\gamma,\boldsymbol{r}}(I)\rangle - \langle\hat{\boldsymbol{v}}_i^{*,\mu,\gamma,\boldsymbol{r}}(I),\sigma_i^{*,\mu,\gamma,\boldsymbol{r}}(I)\rangle\|_2^2$ in Eq. (11), we get

$$\|\langle\hat{\boldsymbol{v}}_i^t(I),\sigma_i^{*,\mu,\gamma,\boldsymbol{r}}(I)\rangle - \langle\hat{\boldsymbol{v}}_i^{*,\mu,\gamma,\boldsymbol{r}}(I),\sigma_i^{*,\mu,\gamma,\boldsymbol{r}}(I)\rangle\|_2^2$$

$$=\|\langle\hat{\boldsymbol{v}}_i^t(I) - \hat{\boldsymbol{v}}_i^{*,\mu,\gamma,\boldsymbol{r}}(I),\sigma_i^{*,\mu,\gamma,\boldsymbol{r}}(I)\rangle\|_2^2$$

$$\le\|\hat{\boldsymbol{v}}_i^t(I) - \hat{\boldsymbol{v}}_i^{*,\mu,\gamma,\boldsymbol{r}}(I)\|_2^2\|\sigma_i^{*,\mu,\gamma,\boldsymbol{r}}(I)\rangle\|_2^2 \qquad (13)$$

$$\le\|\hat{\boldsymbol{v}}_i^t(I) - \hat{\boldsymbol{v}}_i^{*,\mu,\gamma,\boldsymbol{r}}(I)\|_2^2,$$

where the last inequality comes from $\|\sigma_i^{*,\mu,\gamma,\boldsymbol{r}}(I)\|_2^2 \le 1$ as $\sigma_i^{*,\mu,\gamma,\boldsymbol{r}}(I)$ is in simplex. By substituting Eq. (12) and (13) into Eq. (11), as well as using $A_{max} \ge 1$, we obtain

$$\|\hat{\boldsymbol{m}}_i^t(I) - \hat{\boldsymbol{m}}_i^{*,\mu,\gamma,\boldsymbol{r}}(I)\|_2^2$$

$$\le2\|\hat{\boldsymbol{v}}_i^t(I) - \hat{\boldsymbol{v}}_i^{*,\mu,\gamma,\boldsymbol{r}}(I)\|_2^2 + 4A_{max}^2(P + 2\mu D)^2\|\hat{\boldsymbol{v}}_i^t(I) - \hat{\boldsymbol{v}}_i^{*,\mu,\gamma,\boldsymbol{r}}(I)\|_2^2$$

$$+ 4A_{max}^2\|\hat{\boldsymbol{v}}_i^t(I) - \hat{\boldsymbol{v}}_i^{*,\mu,\gamma,\boldsymbol{r}}(I)\|_2^2 \qquad (14)$$

$$\le6A_{max}^2\|\hat{\boldsymbol{v}}_i^t(I) - \hat{\boldsymbol{v}}_i^{*,\mu,\gamma,\boldsymbol{r}}(I)\|_2^2 + 4A_{max}^2(P + 2\mu D)^2\|\sigma_i^t(I) - \sigma_i^{*,\mu,\gamma,\boldsymbol{r}}(I)\|_2^2.$$

By applying Lemma D.3 into Eq. (14), we get

$$
\begin{aligned}
&\|\hat{\boldsymbol{m}}_i^t(I) - \hat{\boldsymbol{m}}_i^{*,\mu,\gamma,\boldsymbol{r}}(I)\|_2^2 \\
\leq & 12 A_{max}^2 (L+\mu)^2 \|\hat{\boldsymbol{x}}^t - \hat{\boldsymbol{x}}^{*,\mu,\gamma,\boldsymbol{r}}\|_1^2 + 12 A_{max}^2 (P+2\mu D)^2 \|\hat{\sigma}_i^t - \hat{\sigma}_i^{*,\mu,\gamma,\boldsymbol{r}}\|_1^2 \\
& + 4 A_{max}^2 (P+2\mu D)^2 \|\sigma_i^t(I) - \sigma_i^{*,\mu,\gamma,\boldsymbol{r}}(I)\|_2^2 \\
\leq & 12 A_{max}^2 (L+\mu)^2 \|\hat{\boldsymbol{x}}^t - \hat{\boldsymbol{x}}^{*,\mu,\gamma,\boldsymbol{r}}\|_1^2 + 16 A_{max}^2 (P+2\mu D)^2 \|\hat{\sigma}_i^t - \hat{\sigma}_i^{*,\mu,\gamma,\boldsymbol{r}}\|_1^2.
\end{aligned} \tag{15}
$$

By applying Lemma D.4 into Eq. (15), we get

$$
\begin{aligned}
&\|\hat{\boldsymbol{m}}_i^t(I) - \hat{\boldsymbol{m}}_i^{*,\mu,\gamma,\boldsymbol{r}}(I)\|_2^2 \\
\leq & 12 A_{max}^2 (L+\mu)^2 \|\hat{\boldsymbol{x}}^t - \hat{\boldsymbol{x}}^{*,\mu,\gamma,\boldsymbol{r}}\|_1^2 + 16 A_{max}^2 (P+2\mu D)^2 \frac{(A_{max} C_{max}+1)^2}{\gamma^{2H}} \|\hat{\boldsymbol{x}}^t - \hat{\boldsymbol{x}}^{*,\mu,\gamma,\boldsymbol{r}}\|_1^2.
\end{aligned} \tag{16}
$$

By substituting Eq. (16) into Eq. (10), we have

$$
\begin{aligned}
&\sum_{t=1}^T \mu\eta \|\hat{\boldsymbol{x}}^t - \hat{\boldsymbol{x}}^{*,\mu,\gamma,\boldsymbol{r}}\|_2^2 - \sum_{t=1}^T \sum_{i\in\mathcal{N}} \sum_{I\in\mathcal{I}_i} \eta^2 A_{max}^2 \left( 12(L+\mu)^2 + 16(P+2\mu D)^2 \frac{(A_{max}C_{max}+1)^2}{\gamma^{2H}} \right) \frac{\|\hat{\boldsymbol{x}}^t - \hat{\boldsymbol{x}}^{*,\mu,\gamma,\boldsymbol{r}}\|_2^2}{2} \\
\leq & \sum_{i\in\mathcal{N}} \sum_{I\in\mathcal{I}_i} \zeta_I \left( D_\psi(\sigma_i^{*,\mu,\gamma,\boldsymbol{r}}(I),\boldsymbol{\theta}_I^1) + \eta\langle -\hat{\boldsymbol{m}}_i^{*,\mu,\gamma,\boldsymbol{r}}(I),\boldsymbol{\theta}_I^1\rangle - D_\psi(\sigma_i^{*,\mu,\gamma,\boldsymbol{r}}(I),\boldsymbol{\theta}_I^{T+1}) - \eta\langle -\hat{\boldsymbol{m}}_i^{*,\mu,\gamma,\boldsymbol{r}}(I),\boldsymbol{\theta}_I^{T+1}\rangle \right),
\end{aligned}
$$

which implies

$$
\begin{aligned}
&\sum_{t=1}^T \mu\eta \|\hat{\boldsymbol{x}}^t - \hat{\boldsymbol{x}}^{*,\mu,\gamma,\boldsymbol{r}}\|_2^2 - \sum_{t=1}^T \eta^2 |\mathcal{I}| A_{max}^2 \left( 6(L+\mu)^2 + 8(P+2\mu D)^2 \frac{(A_{max}C_{max}+1)^2}{\gamma^{2H}} \right) \|\hat{\boldsymbol{x}}^t - \hat{\boldsymbol{x}}^{*,\mu,\gamma,\boldsymbol{r}}\|_2^2 \\
\leq & \sum_{i\in\mathcal{N}} \sum_{I\in\mathcal{I}_i} \zeta_I \left( D_\psi(\sigma_i^{*,\mu,\gamma,\boldsymbol{r}}(I),\boldsymbol{\theta}_I^1) + \eta\langle -\hat{\boldsymbol{m}}_i^{*,\mu,\gamma,\boldsymbol{r}}(I),\boldsymbol{\theta}_I^1\rangle - D_\psi(\sigma_i^{*,\mu,\gamma,\boldsymbol{r}}(I),\boldsymbol{\theta}_I^{T+1}) - \eta\langle -\hat{\boldsymbol{m}}_i^{*,\mu,\gamma,\boldsymbol{r}}(I),\boldsymbol{\theta}_I^{T+1}\rangle \right),
\end{aligned}
$$

Obviously, if

$$
\mu \geq 2\eta |\mathcal{I}| A_{max}^2 \left( 6(L+\mu)^2 + 8(P+2\mu D)^2 \frac{(A_{max}C_{max}+1)^2}{\gamma^{2H}} \right) > 0
$$

$$
\Leftrightarrow 0 < \eta \leq \frac{\mu\gamma^{2H}}{2|\mathcal{I}| A_{max}^2 (6\gamma^{2H}(L+\mu)^2 + 8(A_{max}C_{max}+1)^2(P+2\mu D)^2)},
$$

we have

$$
\begin{aligned}
&\sum_{t=1}^T \frac{\mu\eta}{2} \|\hat{\boldsymbol{x}}^t - \hat{\boldsymbol{x}}^{*,\mu,\gamma,\boldsymbol{r}}\|_2^2 \\
\leq & \sum_{i\in\mathcal{N}} \sum_{I\in\mathcal{I}_i} \zeta_I \left( D_\psi(\sigma_i^{*,\mu,\gamma,\boldsymbol{r}}(I),\boldsymbol{\theta}_I^1) + \eta\langle -\hat{\boldsymbol{m}}_i^{*,\mu,\gamma,\boldsymbol{r}}(I),\boldsymbol{\theta}_I^1\rangle - D_\psi(\sigma_i^{*,\mu,\gamma,\boldsymbol{r}}(I),\boldsymbol{\theta}_I^{T+1}) - \eta\langle -\hat{\boldsymbol{m}}_i^{*,\mu,\gamma,\boldsymbol{r}}(I),\boldsymbol{\theta}_I^{T+1}\rangle \right),
\end{aligned}
$$

By using Lemma 4.4, we have that $\eta\langle -\hat{\boldsymbol{m}}_i^{*,\mu,\gamma,\boldsymbol{r}}(I),\boldsymbol{\theta}_I^{T+1}\rangle \geq 0$. As a result, we get $-D_\psi(\sigma_i^{*,\mu,\gamma,\boldsymbol{r}}(I),\boldsymbol{\theta}_I^{T+1}) - \eta\langle -\hat{\boldsymbol{m}}_i^{*,\mu,\gamma,\boldsymbol{r}}(I),\boldsymbol{\theta}_I^{T+1}\rangle \leq 0$. Then, we conclude that $\forall T \geq 1$

$$
\begin{aligned}
&\sum_{t=1}^T \frac{\mu\eta}{2} \|\hat{\boldsymbol{x}}^t - \hat{\boldsymbol{x}}^{*,\mu,\gamma,\boldsymbol{r}}\|_2^2 \leq \sum_{i\in\mathcal{N}} \sum_{I\in\mathcal{I}_i} \zeta_I \left( D_\psi(\sigma_i^{*,\mu,\gamma,\boldsymbol{r}}(I),\boldsymbol{\theta}_I^1) + \eta\langle -\hat{\boldsymbol{m}}_i^{*,\mu,\gamma,\boldsymbol{r}}(I),\boldsymbol{\theta}_I^1\rangle \right) \\
&\Rightarrow \sum_{t=1}^T \|\hat{\boldsymbol{x}}^t - \hat{\boldsymbol{x}}^{*,\mu,\gamma,\boldsymbol{r}}\|_2^2 \leq O(1),
\end{aligned}
$$

which implies the asymptotic last-iterate convergence of the sequence $\{\hat{\boldsymbol{x}}^1, \hat{\boldsymbol{x}}^2, \cdots, \hat{\boldsymbol{x}}^t, \cdots\}$ to NE $\hat{\boldsymbol{x}}^{*,\mu,\gamma,\boldsymbol{r}}$ of the perturbed regularized EFG since $0 \leq \zeta_I \leq 1$ (as mentioned above).

As analyzed in Farina et al. [2021], if $\boldsymbol{\theta}_I^1 = \boldsymbol{0}$, for any $\eta > 0$, the generated sequence $\{\hat{\boldsymbol{x}}^1, \hat{\boldsymbol{x}}^2, \ldots, \hat{\boldsymbol{x}}^t, \ldots\}$ remains identical, achieving the parameter-free property. In this paper, we further establish that for any initial $\boldsymbol{\theta}_I^1 \in \mathbb{R}_{\geq 0}^{|A(I)|}$ and $\eta > 0$, the sequence $\hat{\boldsymbol{x}}^t$ converges to $\hat{\boldsymbol{x}}^{*,\mu,\gamma,\boldsymbol{r}}$.

We first prove that for the accumulated counterfactual regret sequence $\{\boldsymbol{\theta}_I^1, \boldsymbol{\theta}_I^2, \ldots, \boldsymbol{\theta}_I^t, \ldots\}$ generated by $\boldsymbol{\theta}_I^1 \in \mathbb{R}_{\geq 0}^{|A(I)|}$ and $\eta > 0$, there exists a corresponding sequence $\{\boldsymbol{\theta}_I^{1'}, \boldsymbol{\theta}_I^{2'}, \ldots, \boldsymbol{\theta}_I^{t'}, \ldots\}$ generated by $\boldsymbol{\theta}_I^{1'} \in \mathbb{R}_{\geq 0}^{|A(I)|}$ and $\eta' = \mu/(2C_0)$, such that the resulting strategy profile sequence $\{\hat{\boldsymbol{x}}^1, \hat{\boldsymbol{x}}^2, \ldots, \hat{\boldsymbol{x}}^t, \ldots\}$ is identical. By the update rule of $\text{CFR}^+$ defined in Eq. (3) and the analysis in Farina et al. [2021], $\boldsymbol{\theta}_I^{t+1} \in \arg\min_{\boldsymbol{\theta}_I \in \mathbb{R}_{\geq 0}^{|A(I)|}} \left\{ \langle -\hat{\boldsymbol{m}}_i^t(I), \boldsymbol{\theta}_I \rangle + \frac{1}{\eta} D_\psi(\boldsymbol{\theta}_I, \boldsymbol{\theta}_I^t) \right\}$ can be expressed as the projection $\boldsymbol{\theta}_I^{t+1} = [\boldsymbol{\theta}_I^t + \eta \hat{\boldsymbol{m}}_i^t(I)]^+$, where $[\cdot]^+ = \max(\cdot, \boldsymbol{0})$. Setting $\boldsymbol{\theta}_I^{t'} = \eta' \boldsymbol{\theta}_I^t / \eta$ for $t \geq 1$, it follows that $\boldsymbol{\theta}_I^{t+1'} = [\boldsymbol{\theta}_I^{t'} + \eta' \hat{\boldsymbol{m}}_i^t(I)]^+$ and $\sigma_i^t(I) = \boldsymbol{\theta}_I^t / \langle \boldsymbol{\theta}_I^t, \mathbf{1} \rangle = \boldsymbol{\theta}_I^{t'} / \langle \boldsymbol{\theta}_I^{t'}, \mathbf{1} \rangle$ hold [Chakrabarti et al., 2024]. Furthermore, it is evident that $\sum_{t=1}^T \|\hat{\boldsymbol{x}}^t - \hat{\boldsymbol{x}}^{*,\mu,\gamma,\boldsymbol{r}}\|_2^2 \leq O(1)$ holds independently of the value of the initial accumulated counterfactual regret.

Based on the above analysis, we conclude that (i) for any accumulated counterfactual regret sequence $\{\boldsymbol{\theta}_I^1, \boldsymbol{\theta}_I^2, \ldots, \boldsymbol{\theta}_I^t, \ldots\}$ generated by any $\boldsymbol{\theta}_I^1 \in \mathbb{R}_{\geq 0}^{|A(I)|}$ and $\eta > 0$, there exists a corresponding accumulated counterfactual regret sequence $\{\boldsymbol{\theta}_I^{1'}, \boldsymbol{\theta}_I^{2'}, \ldots, \boldsymbol{\theta}_I^{t'}, \ldots\}$ generated by $\boldsymbol{\theta}_I^{1'}$ and $\eta' = \mu/(2C_0)$, such that the resulting strategy profile sequence $\{\hat{\boldsymbol{x}}^1, \hat{\boldsymbol{x}}^2, \ldots, \hat{\boldsymbol{x}}^t, \ldots\}$ are identical, as well as (ii) the strategy profile sequence $\{\hat{\boldsymbol{x}}^1, \hat{\boldsymbol{x}}^2, \ldots, \hat{\boldsymbol{x}}^t, \ldots\}$ generated by the accumulated counterfactual regret sequence $\{\boldsymbol{\theta}_I^{1'}, \boldsymbol{\theta}_I^{2'}, \ldots, \boldsymbol{\theta}_I^{t'}, \ldots\}$ converges to $\hat{\boldsymbol{x}}^{*,\mu,\gamma,\boldsymbol{r}}$. Therefore, we have that for any $\boldsymbol{\theta}_I^1 \in \mathbb{R}_{\geq 0}^{|A(I)|}$ and $\eta > 0$, the generated strategy profile sequence $\{\hat{\boldsymbol{x}}^1, \hat{\boldsymbol{x}}^2, \ldots, \hat{\boldsymbol{x}}^t, \ldots\}$ converges to $\hat{\boldsymbol{x}}^{*,\mu,\gamma,\boldsymbol{r}}$, demonstrating the parameter-free property. We complete the proof. $\qquad\square$

# E  Proof of Useful Lemmas

## E.1  Proof of Lemma 4.3

*Proof.* From the definition of $\sum_{I \in \mathcal{I}_i} \pi_i^{\sigma'}(I) \langle -\boldsymbol{v}_i^\sigma(I), \sigma_i(I) - \sigma_i'(I) \rangle$, we get

$$
\begin{aligned}
&\sum_{I \in \mathcal{I}_i} \pi_i^{\sigma'}(I) \langle -\boldsymbol{v}_i^\sigma(I), \sigma_i(I) - \sigma_i'(I) \rangle \\
&= \sum_{I \in \mathcal{I}_i} \pi_i^{\sigma'}(I) \langle -\boldsymbol{v}_i^\sigma(I), \sigma_i(I) \rangle - \sum_{I \in \mathcal{I}_i} \pi_i^{\sigma'}(I) \langle -\boldsymbol{v}_i^\sigma(I), \sigma_i'(I) \rangle.
\end{aligned}
\tag{17}
$$

For the term $\sum_{I \in \mathcal{I}_i} \pi_i^{\sigma'}(I) \langle -\boldsymbol{v}_i^\sigma(I), \sigma_i'(I) \rangle$, we have

$$
\begin{aligned}
&\sum_{I \in \mathcal{I}_i} \pi_i^{\sigma'}(I) \langle -\boldsymbol{v}_i^\sigma(I), \sigma_i'(I) \rangle \\
&= \sum_{I \in \mathcal{I}_i} \pi_i^{\sigma'}(I) \sum_{a \in A(I)} \sigma_i'(I,a) \left( \boldsymbol{\ell}_i(I,a) + \sum_{I' \in C_i(I,a)} \langle -\boldsymbol{v}_i^\sigma(I'), \sigma_i(I') \rangle \right) \\
&= \sum_{I \in \mathcal{I}_i} \sum_{a \in A(I)} \pi_i^{\sigma'}(I) \sigma_i'(I,a) \boldsymbol{\ell}_i(I,a) + \sum_{I \in \mathcal{I}_i} \sum_{a \in A(I)} \pi_i^{\sigma'}(I) \sigma_i'(I,a) \sum_{I' \in C_i(I,a)} \langle -\boldsymbol{v}_i^\sigma(I'), \sigma_i(I') \rangle.
\end{aligned}
\tag{18}
$$

Then, by substituting Eq. (18) into Eq. (17), we have

$$
\begin{aligned}
&\sum_{I \in \mathcal{I}_i} \pi_i^{\sigma'}(I) \langle -\boldsymbol{v}_i^\sigma(I), \sigma_i(I) - \sigma_i'(I) \rangle \\
&= \sum_{I \in \mathcal{I}_i} \pi_i^{\sigma'}(I) \langle -\boldsymbol{v}_i^\sigma(I), \sigma_i(I) \rangle - \sum_{I \in \mathcal{I}_i} \sum_{a \in A(I)} \pi_i^{\sigma'}(I) \sigma_i'(I,a) \boldsymbol{\ell}_i(I,a) - \sum_{I \in \mathcal{I}_i} \sum_{a \in A(I)} \pi_i^{\sigma'}(I) \sigma_i'(I,a) \sum_{I' \in C_i(I,a)} \langle -\boldsymbol{v}_i^\sigma(I'), \sigma_i(I') \rangle \\
&= \sum_{I \in \mathcal{I}_i} \pi_i^{\sigma'}(I) \langle -\boldsymbol{v}_i^\sigma(I), \sigma_i(I) \rangle - \sum_{I \in \mathcal{I}_i} \sum_{a \in A(I)} \pi_i^{\sigma'}(I) \sigma_i'(I,a) \boldsymbol{\ell}_i(I,a) - \sum_{I \in \mathcal{I}_i} \sum_{a \in A(I)} \sum_{I' \in C_i(I,a)} \pi_i^{\sigma'}(I') \langle -\boldsymbol{v}_i^\sigma(I'), \sigma_i(I') \rangle.
\end{aligned}
$$
$$\tag{19}$$

We denote the initial infosets as $\mathcal{I}_i^{init}$, i.e., for any $I \in \mathcal{I}_i^{init}$, there does not exist $I'' \in \mathcal{I}_i$ such that $I \in C_i(I'', a'')$ holds for a $a'' \in A(I'')$. For the term $\sum_{I \in \mathcal{I}_i} \pi_i^{\sigma'}(I) \langle -\boldsymbol{v}_i^\sigma(I), \sigma_i(I) \rangle -$

$\sum_{I \in \mathcal{I}_i} \sum_{a \in A(I)} \sum_{I' \in C_i(I,a)} \pi_i^{\sigma'}(I') \langle -\boldsymbol{v}_i^\sigma(I'), \sigma_i(I') \rangle$ in Eq. (19), it follows that

$$
\sum_{I \in \mathcal{I}_i} \pi_i^{\sigma'}(I) \langle -\boldsymbol{v}_i^\sigma(I), \sigma_i(I) \rangle - \sum_{I \in \mathcal{I}_i} \sum_{a \in A(I)} \sum_{I' \in C_i(I,a)} \pi_i^{\sigma'}(I') \langle -\boldsymbol{v}_i^\sigma(I'), \sigma_i(I') \rangle
$$
$$
= \sum_{I \in \mathcal{I}_i^{init}} \pi_i^{\sigma'}(I) \langle -\boldsymbol{v}_i^\sigma(I), \sigma_i(I) \rangle. \tag{20}
$$

Since the probability of reaching any $I \in \mathcal{I}_i^{init}$ is always 1, regardless of the strategies $\sigma$ or $\sigma'$, we have that $\forall \sigma, \sigma'$, and $I \in \mathcal{I}_i^{init}$, $\pi_i^{\sigma'}(I) = \pi_i^\sigma(I)$. Substituting this into Eq. (20), we obtain

$$
\sum_{I \in \mathcal{I}_i^{init}} \pi_i^{\sigma'}(I) \langle -\boldsymbol{v}_i^\sigma(I), \sigma_i(I) \rangle
$$
$$
= \sum_{I \in \mathcal{I}_i^{init}} \pi_i^\sigma(I) \langle -\boldsymbol{v}_i^\sigma(I), \sigma_i(I) \rangle
$$
$$
= \sum_{I \in \mathcal{I}_i^{init}} \pi_i^\sigma(I) \sum_{a \in A(I)} \sigma_i(I,a) \left( \boldsymbol{\ell}_i(I,a) + \sum_{I' \in C_i(I,a)} \langle -\boldsymbol{v}_i^\sigma(I'), \sigma_i(I') \rangle \right) \tag{21}
$$
$$
= \sum_{I \in \mathcal{I}_i} \sum_{a \in A(I)} \pi_i^\sigma(I) \sigma_i(I,a) \boldsymbol{\ell}_i(I,a),
$$

where the last line follows from the recursion. Substituting Eq. (21) into Eq. (19), we obtain

$$
\sum_{I \in \mathcal{I}_i} \pi_i^{\sigma'}(I) \langle -\boldsymbol{v}_i^\sigma(I), \sigma_i(I) - \sigma_i'(I) \rangle
$$
$$
= \sum_{I \in \mathcal{I}_i} \sum_{a \in A(I)} \left[ \pi_i^\sigma(I) \sigma_i(I,a) \boldsymbol{\ell}_i(I,a) - \pi_i^{\sigma'}(I) \sigma_i'(I,a) \boldsymbol{\ell}_i(I,a) \right] = \langle \boldsymbol{\ell}_i, \boldsymbol{x}_i - \boldsymbol{x}_i' \rangle,
$$

as $\forall i \in \mathcal{N}, I \in \mathcal{I}_i, \pi_i^\sigma(I) \sigma_i(I,a) = \boldsymbol{x}_i(I,a)$ and $\pi_i^{\sigma'}(I) \sigma_i'(I,a) = \boldsymbol{x}_i'(I,a)$ via the definition of the sequence-form strategy. It finishes the proof. $\qquad \square$

### E.2  Proof of Lemma 4.4

*Proof.* First, when $\boldsymbol{\theta}_I = \boldsymbol{0}$, we have that $\forall I \in \mathcal{I}_i$, $\langle -\hat{\boldsymbol{m}}_i^{*,\mu,\gamma,\boldsymbol{r}}(I), \boldsymbol{\theta}_I \rangle = 0$.

Next, we prove by contradiction that when $\boldsymbol{\theta}_I > \boldsymbol{0}$, $\forall I \in \mathcal{I}_i$, it holds that $\langle -\hat{\boldsymbol{m}}_i^{*,\mu,\gamma,\boldsymbol{r}}(I), \boldsymbol{\theta}_I \rangle \geq 0$.

Suppose there exists one $I' \in \mathcal{I}_i$ and $\boldsymbol{\theta}_{I'}' > \boldsymbol{0}$ such that $\langle -\hat{\boldsymbol{m}}^{*,\mu,\gamma,\boldsymbol{r}}(I'), \boldsymbol{\theta}_{I'}' \rangle < 0$. We construct a new strategy $\sigma_i'$, which matches $\sigma_i^{*,\mu,\gamma,\boldsymbol{r}}$ (not $\hat{\sigma}_i^{*,\mu,\gamma,\boldsymbol{r}}$) except at the infoset $I'$, where it is defined as $\boldsymbol{\theta}_{I'}' / \langle \boldsymbol{\theta}_{I'}', \boldsymbol{1} \rangle$. For $\langle -\hat{\boldsymbol{m}}^{*,\mu,\gamma,\boldsymbol{r}}(I'), \boldsymbol{\theta}_{I'}' \rangle$, we have

$$
\langle -\hat{\boldsymbol{m}}^{*,\mu,\gamma,\boldsymbol{r}}(I'), \boldsymbol{\theta}_{I'}' \rangle = -\langle \hat{\boldsymbol{v}}_i^{*,\mu,\gamma,\boldsymbol{r}}(I') - \langle \hat{\boldsymbol{v}}_i^{*,\mu,\gamma,\boldsymbol{r}}(I'), \sigma_i^{*,\mu,\gamma,\boldsymbol{r}}(I') \rangle \boldsymbol{1}, \boldsymbol{\theta}_{I'}' \rangle
$$
$$
= -\|\boldsymbol{\theta}_{I'}'\|_1 \langle \hat{\boldsymbol{v}}_i^{*,\mu,\gamma,\boldsymbol{r}}(I'), \sigma_i'(I') - \sigma_i^{*,\mu,\gamma,\boldsymbol{r}}(I') \rangle
$$
$$
= -\|\boldsymbol{\theta}_{I'}'\|_1 \langle -\hat{\boldsymbol{v}}_i^{*,\mu,\gamma,\boldsymbol{r}}(I'), \sigma_i^{*,\mu,\gamma,\boldsymbol{r}}(I') - \sigma_i'(I') \rangle.
$$

Since $\langle -\hat{\boldsymbol{m}}^{*,\mu,\gamma,\boldsymbol{r}}(I'), \boldsymbol{\theta}_{I'}' \rangle < 0$ and $\|\boldsymbol{\theta}_{I'}'\|_1 > 0$, we have $\langle -\hat{\boldsymbol{v}}_i^{*,\mu,\gamma,\boldsymbol{r}}(I'), \sigma_i^{*,\mu,\gamma,\boldsymbol{r}}(I') - \sigma_i'(I') \rangle > 0$. We define $\hat{\sigma}_i'(I) = (1 - \alpha_I) \sigma_i'(I) + \gamma \boldsymbol{1}$ for all $I \in \mathcal{I}_i$. Additionally, we know that $\hat{\sigma}_i^{*,\mu,\gamma,\boldsymbol{r}}(I) = (1 - \alpha_I) \sigma_i^{*,\mu,\gamma,\boldsymbol{r}}(I) + \gamma \boldsymbol{1}$ for all $I \in \mathcal{I}_i$, and that $\langle -\hat{\boldsymbol{v}}_i^{*,\mu,\gamma,\boldsymbol{r}}(I'), \sigma_i^{*,\mu,\gamma,\boldsymbol{r}}(I') - \sigma_i'(I') \rangle > 0$. Hence, it follows that $\langle -\hat{\boldsymbol{v}}_i^{*,\mu,\gamma,\boldsymbol{r}}(I'), \hat{\sigma}_i^{*,\mu,\gamma,\boldsymbol{r}}(I') - \hat{\sigma}_i'(I') \rangle > 0$.

The correspond sequence-form strategy of $\hat{\sigma}_i'$ is represented by $\hat{\boldsymbol{x}}_i'$. According to Lemma 4.3 and the definition of NE, we get

$$
\langle \hat{\boldsymbol{\ell}}_i^{\boldsymbol{x}^{*,\mu,\gamma,\boldsymbol{r}}}, \hat{\boldsymbol{x}}_i^{*,\mu,\gamma,\boldsymbol{r}} - \hat{\boldsymbol{x}}_i' \rangle = \sum_{I \in \mathcal{I}_i} \pi_i^{\sigma'}(I) \langle -\hat{\boldsymbol{v}}_i^{*,\mu,\gamma,\boldsymbol{r}}(I), \hat{\sigma}_i^{*,\mu,\gamma,\boldsymbol{r}}(I) - \hat{\sigma}_i'(I) \rangle \leq 0. \tag{22}
$$

Since $\sigma_i'$ matches $\sigma_i^{*,\mu,\gamma,\boldsymbol{r}}$ except at the infoset $I'$, and given that $\hat{\sigma}_i'(I) = (1 - \alpha_I) \sigma_i'(I) + \gamma \boldsymbol{1}$ for all $I \in \mathcal{I}_i$, as well as $\hat{\sigma}_i^{*,\mu,\gamma,\boldsymbol{r}}(I) = (1 - \alpha_I) \sigma_i^{*,\mu,\gamma,\boldsymbol{r}}(I) + \gamma \boldsymbol{1}$ for all $I \in \mathcal{I}_i$, we obtain

$\hat{\sigma}_i^{*,\mu,\gamma,\boldsymbol{r}}(I) - \hat{\sigma}_i'(I) = \mathbf{0}$ holds for all $I \in \mathcal{I}_i$ except $I'$. Therefore, we get

$$\langle \hat{\boldsymbol{\ell}}_i^{\boldsymbol{x}^{*,\mu,\gamma,\boldsymbol{r}}}, \hat{\boldsymbol{x}}_i^{*,\mu,\gamma,\boldsymbol{r}} - \hat{\boldsymbol{x}}_i' \rangle = \sum_{I \in \mathcal{I}_i} \pi^{\sigma'}(I) \langle -\hat{\boldsymbol{v}}_i^{*,\mu,\gamma,\boldsymbol{r}}(I), \hat{\sigma}_i^{*,\mu,\gamma,\boldsymbol{r}}(I) - \hat{\sigma}_i'(I) \rangle \tag{23}$$
$$= \langle -\hat{\boldsymbol{v}}_i^{*,\mu,\gamma,\boldsymbol{r}}(I'), \hat{\sigma}_i^{*,\mu,\gamma,\boldsymbol{r}}(I') - \hat{\sigma}_i'(I') \rangle > 0,$$

where $\hat{\boldsymbol{x}}_i'$ is the sequence-form strategy profile associated with $\hat{\sigma}_i$. By the definition of $\hat{\boldsymbol{x}}_i'$, it follows that $\hat{\boldsymbol{x}}_i' \in \mathcal{X}_i^\gamma$. However, from the definition of NE, as shown in Eq. (22), $\langle \hat{\boldsymbol{\ell}}_i^{\boldsymbol{x}^{*,\mu,\gamma,\boldsymbol{r}}}, \hat{\boldsymbol{x}}_i^{*,\mu,\gamma,\boldsymbol{r}} - \hat{\boldsymbol{x}}_i' \rangle \leq 0$, which contradicts the result in Eq. (23). Therefore, there exists no $I' \in \mathcal{I}_i$ and $\boldsymbol{\theta}_{I'}' > \mathbf{0}$ such that $\langle -\hat{\boldsymbol{m}}^{*,\mu,\gamma,\boldsymbol{r}}(I'), \boldsymbol{\theta}_{I'}' \rangle < 0$. Consequently, when $\boldsymbol{\theta}_I > \mathbf{0}$ for all $I \in \mathcal{I}_i$, it holds that $\langle -\hat{\boldsymbol{m}}_i^{*,\mu,\gamma,\boldsymbol{r}}(I), \boldsymbol{\theta}_I \rangle \geq 0$.

Through the discussion of the above two situations, we complete the proof. $\qquad\square$

### E.3 Proof of Lemma D.2

*Proof.* From the definition of $\hat{\boldsymbol{v}}_i^\sigma(I)$, we get

$$
\begin{aligned}
\|\hat{\boldsymbol{v}}_i^\sigma(I)\|_2 \leq \|\hat{\boldsymbol{v}}_i^\sigma(I)\|_1 &= \sum_{a \in A(I)} \|\hat{\boldsymbol{v}}_i^\sigma(I,a)\|_1 \\
&= \sum_{a \in A(I)} \| -\hat{\boldsymbol{\ell}}_i^{\boldsymbol{x}}(I,a) + \sum_{I' \in C_i(I,a)} \langle \hat{\boldsymbol{v}}_i^\sigma(I'), \sigma_i(I') \rangle \|_1 \\
&\leq \sum_{a \in A(I)} \| -\hat{\boldsymbol{\ell}}_i^{\boldsymbol{x}}(I,a) \|_1 + \sum_{a \in A(I)} \sum_{I' \in C_i(I,a)} \| \langle \hat{\boldsymbol{v}}_i^\sigma(I'), \sigma_i(I') \rangle \|_1 \\
&\leq \sum_{a \in A(I)} \| -\hat{\boldsymbol{\ell}}_i^{\boldsymbol{x}}(I,a) \|_1 + \sum_{a \in A(I)} \sum_{I' \in C_i(I,a)} \| \hat{\boldsymbol{v}}_i^\sigma(I') \|_1 \| \sigma_i(I') \|_1 \\
&\leq \sum_{a \in A(I)} \| -\hat{\boldsymbol{\ell}}_i^{\boldsymbol{x}}(I,a) \|_1 + \sum_{a \in A(I)} \sum_{I' \in C_i(I,a)} \| \hat{\boldsymbol{v}}_i^\sigma(I') \|_1 \\
&\leq \sum_{I' \in \mathcal{I}_i} \sum_{a' \in A(I')} \| -\hat{\boldsymbol{\ell}}_i^{\boldsymbol{x}}(I'a') \|_1,
\end{aligned} \tag{24}
$$

where the last line is from recursion. Continuing from the above inequality, we get

$$
\begin{aligned}
&\sum_{I' \in \mathcal{I}_i} \sum_{a' \in A(I')} \| -\hat{\boldsymbol{\ell}}_i^{\boldsymbol{x}}(I'a') \|_1 \\
&= \| \boldsymbol{\ell}_i^{\boldsymbol{x}} + \mu \nabla \psi(\boldsymbol{x}_i) - \mu \nabla \psi(\boldsymbol{r}_i) \|_1 \\
&\leq \| \boldsymbol{\ell}_i^{\boldsymbol{x}}(I) \|_1 + \mu \| \nabla \psi(\boldsymbol{x}_i)(I) \|_1 + \mu \| \nabla \psi(\boldsymbol{r}_i)(I) \|_1 \leq P + 2\mu D,
\end{aligned} \tag{25}
$$

where $\boldsymbol{\ell}_0^{\boldsymbol{x}} = \boldsymbol{A}\boldsymbol{x}_1$ and $\boldsymbol{\ell}_1^{\boldsymbol{x}} = -\boldsymbol{A}^{\mathrm{T}}\boldsymbol{x}_0$. By substituting Eq. (25) into Eq. (24), we have

$$\|\hat{\boldsymbol{v}}_i^\sigma(I)\|_2 \leq \|\hat{\boldsymbol{v}}_i^\sigma(I)\|_1 \leq P + 2\mu D,$$

It completes the proof. $\qquad\square$

### E.4 Proof of Lemma D.3

*Proof.* From the definition of $\hat{\boldsymbol{v}}_i^\sigma(I)$ and $\hat{\boldsymbol{v}}_i^{\sigma'}(I)$, we have

$$
\begin{aligned}
&\|\hat{\boldsymbol{v}}_i^\sigma(I) - \hat{\boldsymbol{v}}_i^{\sigma'}(I)\|_2 \\
&\leq \|\hat{\boldsymbol{v}}_i^\sigma(I) - \hat{\boldsymbol{v}}_i^{\sigma'}(I)\|_1 \\
&= \sum_{a \in A(I)} \| -\hat{\boldsymbol{\ell}}_i^{\boldsymbol{x}}(I,a) + \sum_{I' \in C_i(I,a)} \langle \hat{\boldsymbol{v}}_i^\sigma(I'), \sigma_i(I') \rangle + \hat{\boldsymbol{\ell}}_i^{\boldsymbol{x}'}(I,a) - \sum_{I' \in C_i(I,a)} \langle \hat{\boldsymbol{v}}_i^{\sigma'}(I'), \sigma_i'(I') \rangle \|_1 \\
&\leq \sum_{a \in A(I)} \| -\hat{\boldsymbol{\ell}}_i^{\boldsymbol{x}}(I,a) + \hat{\boldsymbol{\ell}}_i^{\boldsymbol{x}'}(I,a) \|_1 + \sum_{a \in A(I)} \sum_{I' \in C_i(I,a)} \| \langle \hat{\boldsymbol{v}}_i^\sigma(I'), \sigma_i(I') \rangle - \langle \hat{\boldsymbol{v}}_i^{\sigma'}(I'), \sigma_i'(I') \rangle \|_1.
\end{aligned}
$$

Then, we have

$$
\begin{aligned}
&\|\hat{\boldsymbol{v}}_i^\sigma(I){-}\hat{\boldsymbol{v}}_i^{\sigma'}(I)\|_2 \\
\leq{}& \sum_{a\in A(I)} \|{-}\hat{\boldsymbol{\ell}}_i^{\boldsymbol{x}}(I,a){+}\hat{\boldsymbol{\ell}}_i^{\boldsymbol{x}'}(I,a)\|_1 \\
&+\sum_{a\in A(I)}\sum_{I'\in C_i(I,a)} \|\langle\hat{\boldsymbol{v}}_i^\sigma(I'),\sigma_i(I')\rangle{-}\langle\hat{\boldsymbol{v}}_i^\sigma(I'),\sigma_i'(I')\rangle{+}\langle\hat{\boldsymbol{v}}_i^\sigma(I'),\sigma_i'(I')\rangle{-}\langle\hat{\boldsymbol{v}}_i^{\sigma'}(I'),\sigma_i'(I')\rangle\|_1 \\
\leq{}& \sum_{a\in A(I)} \|{-}\hat{\boldsymbol{\ell}}_i^{\boldsymbol{x}}(I,a){+}\hat{\boldsymbol{\ell}}_i^{\boldsymbol{x}'}(I,a)\|_1 +\sum_{a\in A(I)}\sum_{I'\in C_i(I,a)} \|\langle\hat{\boldsymbol{v}}_i^\sigma(I'),\sigma_i(I')\rangle{-}\langle\hat{\boldsymbol{v}}_i^\sigma(I'),\sigma_i'(I')\rangle\|_1 \\
&+\sum_{a\in A(I)}\sum_{I'\in C_i(I,a)} \|\langle\hat{\boldsymbol{v}}_i^\sigma(I'),\sigma_i'(I')\rangle{-}\langle\hat{\boldsymbol{v}}_i^{\sigma'}(I'),\sigma_i'(I')\rangle\|_1.
\end{aligned}
\tag{26}
$$

For the term $\|\langle\hat{\boldsymbol{v}}_i^\sigma(I'),\sigma_i(I')\rangle - \langle\hat{\boldsymbol{v}}_i^\sigma(I'),\sigma_i'(I')\rangle\|_1$ in Eq. (26), we get

$$
\begin{aligned}
\|\langle\hat{\boldsymbol{v}}_i^\sigma(I'),\sigma_i(I')\rangle - \langle\hat{\boldsymbol{v}}_i^\sigma(I'),\sigma_i'(I')\rangle\|_1 ={}&\|\langle\hat{\boldsymbol{v}}_i^\sigma(I'),\sigma_i(I') - \sigma_i'(I')\rangle\|_1 \\
\leq{}&\|\hat{\boldsymbol{v}}_i^\sigma(I')\|_1\|\sigma_i(I') - \sigma_i'(I')\|_1 \\
\leq{}&(P + 2\mu D)\|\sigma_i(I') - \sigma_i'(I')\|_1,
\end{aligned}
\tag{27}
$$

where the last line comes from Lemma D.2. For the term $\|\langle\hat{\boldsymbol{v}}_i^\sigma(I'),\sigma_i'(I')\rangle - \langle\hat{\boldsymbol{v}}_i^{\sigma'}(I'),\sigma_i'(I')\rangle\|_1$ in Eq. (26), we get

$$
\begin{aligned}
\|\langle\hat{\boldsymbol{v}}_i^\sigma(I'),\sigma_i'(I')\rangle - \langle\hat{\boldsymbol{v}}_i^{\sigma'}(I'),\sigma_i'(I')\rangle\|_1 ={}&\|\langle\hat{\boldsymbol{v}}_i^\sigma(I') - \hat{\boldsymbol{v}}_i^{\sigma'}(I'),\sigma_i'(I')\rangle\|_1 \\
\leq{}&\|\hat{\boldsymbol{v}}_i^\sigma(I') - \hat{\boldsymbol{v}}_i^{\sigma'}(I')\|_1\|\sigma_i'(I')\rangle\|_1 \\
\leq{}&\|\hat{\boldsymbol{v}}_i^\sigma(I') - \hat{\boldsymbol{v}}_i^{\sigma'}(I')\|_1,
\end{aligned}
\tag{28}
$$

where the last line comes from $\|\sigma_i'(I')\rangle\|_1 \leq 1$. By substituting Eq. (27) and (28) into Eq. (26), we obtain

$$
\begin{aligned}
&\|\hat{\boldsymbol{v}}_i^\sigma(I) - \hat{\boldsymbol{v}}_i^{\sigma'}(I)\|_2 \\
\leq{}&\|\hat{\boldsymbol{v}}_i^\sigma(I) - \hat{\boldsymbol{v}}_i^{\sigma'}(I)\|_1 \\
\leq{}& \sum_{a\in A(I)} \|-\hat{\boldsymbol{\ell}}_i^{\boldsymbol{x}}(I,a) + \hat{\boldsymbol{\ell}}_i^{\boldsymbol{x}'}(I,a)\|_1 + \sum_{a\in A(I)}\sum_{I'\in C_i(I,a)} (P + 2\mu D)\|\sigma_i(I') - \sigma_i'(I')\|_1 \\
&+ \sum_{a\in A(I)}\sum_{I'\in C_i(I,a)} \|\hat{\boldsymbol{v}}_i^\sigma(I') - \hat{\boldsymbol{v}}_i^{\sigma'}(I')\|_1 \\
\leq{}&\|\hat{\boldsymbol{\ell}}_i^{\boldsymbol{x}} - \hat{\boldsymbol{\ell}}_i^{\boldsymbol{x}'}\|_1 + (P + 2\mu D)\|\sigma_i - \sigma_i'\|_1,
\end{aligned}
\tag{29}
$$

where the last line is from recursion. For the term $\|\hat{\boldsymbol{\ell}}_i^{\boldsymbol{x}} - \hat{\boldsymbol{\ell}}_i^{\boldsymbol{x}'}\|_1$ in Eq. (29), we get

$$
\begin{aligned}
\|\hat{\boldsymbol{\ell}}_i^{\boldsymbol{x}} - \hat{\boldsymbol{\ell}}_i^{\boldsymbol{x}'}\|_1 ={}&\|\boldsymbol{\ell}_i^{\boldsymbol{x}} + \mu\nabla\psi(\boldsymbol{x}_i) - \mu\nabla\psi(\boldsymbol{r}_i) - \boldsymbol{\ell}_i^{\boldsymbol{x}'} - \mu\nabla\psi(\boldsymbol{x}_i') + \mu\nabla\psi(\boldsymbol{r}_i)\|_1 \\
={}&\|\boldsymbol{\ell}_i^{\boldsymbol{x}} + \mu\boldsymbol{x}_i - \boldsymbol{\ell}_i^{\boldsymbol{x}'} - \mu\boldsymbol{x}_i'\|_1 \\
\leq{}&L\|\boldsymbol{x} - \boldsymbol{x}'\|_1 + \mu\|\boldsymbol{x} - \boldsymbol{x}'\|_1 \\
\leq{}&(L + \mu)\|\boldsymbol{x} - \boldsymbol{x}'\|_1,
\end{aligned}
\tag{30}
$$

where $\boldsymbol{\ell}_0^{\boldsymbol{x}} = \boldsymbol{A}\boldsymbol{x}_1$ and $\boldsymbol{\ell}_1^{\boldsymbol{x}} = -\boldsymbol{A}^{\mathrm{T}}\boldsymbol{x}_0$. By substituting Eq. (30) into Eq. (29), we get

$$
\begin{aligned}
&\|\hat{\boldsymbol{v}}_i^\sigma(I) - \hat{\boldsymbol{v}}_i^{\sigma'}(I)\|_2 \leq (L + \mu)\|\boldsymbol{x} - \boldsymbol{x}'\|_1 + (P + 2\mu D)\|\sigma_i - \sigma_i'\|_1 \\
\Rightarrow{}&\|\hat{\boldsymbol{v}}_i^\sigma(I) - \hat{\boldsymbol{v}}_i^{\sigma'}(I)\|_2^2 \leq 2(L + \mu)^2\|\boldsymbol{x} - \boldsymbol{x}'\|_1^2 + 2(P + 2\mu D)^2\|\sigma_i - \sigma_i'\|_1^2,
\end{aligned}
$$

where the second line is from $\forall b, c \in \mathbb{R}$, $(b + c)^2 \leq 2b^2 + 2c^2$ (in this case, $b = (L + \mu)\|\boldsymbol{x} - \boldsymbol{x}'\|_1$ and $c = (P + 2\mu D)\|\sigma_i - \sigma_i'\|_1$). It completes the proof. $\qquad\square$

## E.5 Proof of Lemma D.4

*Proof.* From the definition of $\|\hat{\sigma}_i - \hat{\sigma}_i'\|_1$, we get

$$\|\hat{\sigma}_i - \hat{\sigma}_i'\|_1$$

$$= \sum_{I \in \mathcal{I}_i} \sum_{a \in A(I)} \|\frac{\hat{\boldsymbol{x}}_i(I,a)}{\hat{\boldsymbol{x}}_i(\rho_I)} - \frac{\hat{\boldsymbol{x}}_i'(I,a)}{\hat{\boldsymbol{x}}_i'(\rho_I)}\|_1$$

$$= \sum_{I \in \mathcal{I}_i} \sum_{a \in A(I)} \|\frac{\hat{\boldsymbol{x}}_i(I,a)\hat{\boldsymbol{x}}_i'(\rho_I)}{\hat{\boldsymbol{x}}_i(\rho_I)\hat{\boldsymbol{x}}_i'(\rho_I)} - \frac{\hat{\boldsymbol{x}}_i'(I,a)\hat{\boldsymbol{x}}_i(\rho_I)}{\hat{\boldsymbol{x}}_i(\rho_I)\hat{\boldsymbol{x}}_i'(\rho_I)}\|_1$$

$$= \sum_{I \in \mathcal{I}_i} \sum_{a \in A(I)} \frac{1}{\hat{\boldsymbol{x}}_i(\rho_I)\hat{\boldsymbol{x}}_i'(\rho_I)} \|\hat{\boldsymbol{x}}_i(I,a)\hat{\boldsymbol{x}}_i'(\rho_I) - \hat{\boldsymbol{x}}_i'(I,a)\hat{\boldsymbol{x}}_i(\rho_I)\|_1$$

$$= \sum_{I \in \mathcal{I}_i} \sum_{a \in A(I)} \frac{1}{\hat{\boldsymbol{x}}_i(\rho_I)\hat{\boldsymbol{x}}_i'(\rho_I)} \|\hat{\boldsymbol{x}}_i(I,a)\hat{\boldsymbol{x}}_i'(\rho_I) - \hat{\boldsymbol{x}}_i(I,a)\hat{\boldsymbol{x}}_i(\rho_I) + \hat{\boldsymbol{x}}_i(I,a)\hat{\boldsymbol{x}}_i(\rho_I) - \hat{\boldsymbol{x}}_i'(I,a)\hat{\boldsymbol{x}}_i(\rho_I)\|_1$$

$$= \sum_{I \in \mathcal{I}_i} \sum_{a \in A(I)} \frac{1}{\hat{\boldsymbol{x}}_i(\rho_I)\hat{\boldsymbol{x}}_i'(\rho_I)} \left(\|\hat{\boldsymbol{x}}_i(I,a)\hat{\boldsymbol{x}}_i'(\rho_I) - \hat{\boldsymbol{x}}_i(I,a)\hat{\boldsymbol{x}}_i(\rho_I)\|_1 + \|\hat{\boldsymbol{x}}_i(I,a)\hat{\boldsymbol{x}}_i(\rho_I) - \hat{\boldsymbol{x}}_i'(I,a)\hat{\boldsymbol{x}}_i(\rho_I)\|_1\right).$$

(31)

For the term $\|\hat{\boldsymbol{x}}_i(I,a)\hat{\boldsymbol{x}}_i'(\rho_I) - \hat{\boldsymbol{x}}_i(I,a)\hat{\boldsymbol{x}}_i(\rho_I)\|_1$ in Eq. (31), we have

$$\|\hat{\boldsymbol{x}}_i(I,a)\hat{\boldsymbol{x}}_i'(\rho_I) - \hat{\boldsymbol{x}}_i(I,a)\hat{\boldsymbol{x}}_i(\rho_I)\|_1 = \hat{\boldsymbol{x}}_i(I,a)\|\hat{\boldsymbol{x}}_i'(\rho_I) - \hat{\boldsymbol{x}}_i(\rho_I)\|_1. \tag{32}$$

For the term $\|\hat{\boldsymbol{x}}_i(I,a)\hat{\boldsymbol{x}}_i'(\rho_I) - \hat{\boldsymbol{x}}_i(I,a)\hat{\boldsymbol{x}}_i(\rho_I)\|_1$ in Eq. (31), we have

$$\|\hat{\boldsymbol{x}}_i(I,a)\hat{\boldsymbol{x}}_i(\rho_I) - \hat{\boldsymbol{x}}_i'(I,a)\hat{\boldsymbol{x}}_i(\rho_I)\|_1 = \hat{\boldsymbol{x}}_i(\rho_I)\|\hat{\boldsymbol{x}}_i(I,a) - \hat{\boldsymbol{x}}_i'(I,a)\|_1. \tag{33}$$

By substituting Eq. (32) and (33) into Eq. (31), we have

$$\|\hat{\sigma}_i - \hat{\sigma}_i'\|_1 = \sum_{I \in \mathcal{I}_i} \sum_{a \in A(I)} \frac{1}{\hat{\boldsymbol{x}}_i(\rho_I)\hat{\boldsymbol{x}}_i'(\rho_I)} \left(\hat{\boldsymbol{x}}_i(I,a)\|\hat{\boldsymbol{x}}_i'(\rho_I) - \hat{\boldsymbol{x}}_i(\rho_I)\|_1 + \hat{\boldsymbol{x}}_i(\rho_I)\|\hat{\boldsymbol{x}}_i(I,a) - \hat{\boldsymbol{x}}_i'(I,a)\|_1\right)$$

$$= \sum_{I \in \mathcal{I}_i} \sum_{a \in A(I)} \left(\frac{\hat{\boldsymbol{x}}_i(I,a)}{\hat{\boldsymbol{x}}_i(\rho_I)\hat{\boldsymbol{x}}_i'(\rho_I)} \|\hat{\boldsymbol{x}}_i'(\rho_I) - \hat{\boldsymbol{x}}_i(\rho_I)\|_1 + \frac{\hat{\boldsymbol{x}}_i(\rho_I)}{\hat{\boldsymbol{x}}_i(\rho_I)\hat{\boldsymbol{x}}_i'(\rho_I)} \|\hat{\boldsymbol{x}}_i(I,a) - \hat{\boldsymbol{x}}_i'(I,a)\|_1\right).$$

Since $\hat{\boldsymbol{x}}_i(I,a)/\hat{\boldsymbol{x}}_i(\rho_I) = \hat{\sigma}_i(I,a) \leq 1$, we obtain

$$\|\hat{\sigma}_i - \hat{\sigma}_i'\|_1 \leq \sum_{I \in \mathcal{I}_i} \sum_{a \in A(I)} \left(\frac{1}{\hat{\boldsymbol{x}}_i'(\rho_I)} \|\hat{\boldsymbol{x}}_i'(\rho_I) - \hat{\boldsymbol{x}}_i(\rho_I)\|_1 + \frac{1}{\hat{\boldsymbol{x}}_i'(\rho_I)} \|\hat{\boldsymbol{x}}_i(I,a) - \hat{\boldsymbol{x}}_i'(I,a)\|_1\right)$$

$$\leq \sum_{I \in \mathcal{I}_i} \sum_{a \in A(I)} \frac{1}{\gamma^H} (\|\hat{\boldsymbol{x}}_i'(\rho_I) - \hat{\boldsymbol{x}}_i(\rho_I)\|_1 + \|\hat{\boldsymbol{x}}_i(I,a) - \hat{\boldsymbol{x}}_i'(I,a)\|_1),$$

where the last inequality comes from $\hat{\boldsymbol{x}}_i(I) \leq 1/\gamma^H$ for all $i \in \mathcal{N}, I \in \mathcal{I}_i$, and $\hat{\boldsymbol{x}}_i \in \mathcal{X}_i^\gamma$ (this follows from the facts that $H$ denotes the maximum number of actions taken by all players along any path from the root to a leaf node and the probability of selecting each action is guaranteed to be greater than $\gamma$ in perturbed EFGs). For the term $\sum_{I \in \mathcal{I}_i} \sum_{a \in A(I)} \frac{1}{\gamma^H} \|\hat{\boldsymbol{x}}_i'(\rho_I) - \hat{\boldsymbol{x}}_i(\rho_I)\|_1$, we get

$$\sum_{I \in \mathcal{I}_i} \sum_{a \in A(I)} \frac{1}{\gamma^H} \|\hat{\boldsymbol{x}}_i'(\rho_I) - \hat{\boldsymbol{x}}_i(\rho_I)\|_1$$

$$= \sum_{I \in \mathcal{I}_i} \sum_{a \in A(I)} \sum_{I' \in C_i(I,a)} \sum_{a' \in A(I')} \frac{1}{\gamma^H} \|\hat{\boldsymbol{x}}_i'(I,a) - \hat{\boldsymbol{x}}_i(I,a)\|_1$$

$$\leq \sum_{I \in \mathcal{I}_i} \sum_{a \in A(I)} \frac{A_{max}C_{max}}{\gamma^H} \|\hat{\boldsymbol{x}}_i'(I,a) - \hat{\boldsymbol{x}}_i(I,a)\|_1.$$

Therefore, we have

$$\|\hat{\sigma}_i - \hat{\sigma}_i'\|_1 \leq \sum_{I \in \mathcal{I}_i} \sum_{a \in A(I)} \frac{A_{max}C_{max} + 1}{\gamma^H} \|\hat{\boldsymbol{x}}_i(I,a) - \hat{\boldsymbol{x}}_i'(I,a)\|_1$$

$$= \frac{A_{max}C_{max} + 1}{\gamma^H} \|\hat{\boldsymbol{x}}_i - \hat{\boldsymbol{x}}_i'\|_1.$$

It finishes the proof. $\qquad\square$

# F  Our Parameter-Free Average-Iterate Convergence of CFR$^+$

Now, we extend the proof of CFR$^+$ in Farina et al. [2021] via our proof approach in Appendix D to demonstrate that for all $\eta > 0$, CFR$^+$'s average-iterate convergence holds for all $\boldsymbol{\theta}_I^1 \in \mathbb{R}_{\geq 0}^{|A(I)|}$ not only for $\boldsymbol{\theta}_I^1 = \mathbf{0}$. This result is significant because it implies that even when the strategies generated during the initial iterations are discarded, CFR$^+$ remains achieving average-iterate convergence. Specifically, since average-iterate convergence holds for all $\boldsymbol{\theta}_I^1 \in \mathbb{R}_{\geq 0}^{|A(I)|}$, $\boldsymbol{\theta}_I^t$ can be treated as a new $\boldsymbol{\theta}_I^1$, ensuring that CFR$^+$ enjoys average-iterate convergence for all $\eta > 0$ after iteration $t$. Indeed, discarding the initial phase strategies is a common technique to improve the empirical convergence rate of CFR$^+$ [Steinberger, 2019].

**Theorem F.1.** *Assuming all players follow the update rule of CFR$^+$ with any $\boldsymbol{\theta}_I^1 \in \mathbb{R}_{\geq 0}^{|A(I)|}$ and $\eta > 0$, the average strategy profile $\bar{\boldsymbol{x}}^T = \frac{\sum_{t=1}^T \boldsymbol{x}^t}{T}$ converges to the set of NEs of the perturbed regularized EFGs defined in Eq. (2) with any $\gamma \geq 0$ and $\mu \geq 0$ as $T \to \infty$.*

*Proof.* By substituting $\boldsymbol{\theta}_I = \sigma_i(I) = \frac{\hat{\sigma}_i(I) - \gamma \mathbf{1}}{1 - \alpha_I} \in \Delta_\gamma^{|A(I)|}$ with $\hat{\sigma}_i(I) \in \Delta_\gamma^{|A(I)|}$ into Lemma 4.2, we get

$$\eta \langle \hat{\boldsymbol{m}}_i^t(I), \sigma_i(I) - \boldsymbol{\theta}_I^{t+1} \rangle \leq D_\psi(\sigma_i(I), \boldsymbol{\theta}_I^t) - D_\psi(\sigma_i(I), \boldsymbol{\theta}_I^{t+1}) - D_\psi(\boldsymbol{\theta}_I^{t+1}, \boldsymbol{\theta}_I^t)$$
$$\Leftrightarrow \eta \langle \hat{\boldsymbol{m}}_i^t(I), \sigma_i(I) - \boldsymbol{\theta}_I^t \rangle \leq D_\psi(\sigma_i(I), \boldsymbol{\theta}_I^t) - D_\psi(\sigma_i(I), \boldsymbol{\theta}_I^{t+1}) - D_\psi(\boldsymbol{\theta}_I^{t+1}, \boldsymbol{\theta}_I^t) + \eta \langle \hat{\boldsymbol{m}}_i^t(I), \boldsymbol{\theta}_I^{t+1} - \boldsymbol{\theta}_I^t \rangle.$$

According to the definition of $\hat{\boldsymbol{m}}_i^t(I)$, we have

$$\langle \hat{\boldsymbol{m}}_i^t(I), \sigma_i(I) - \boldsymbol{\theta}_I^t \rangle = \langle \hat{\boldsymbol{v}}_i^t(I) - \langle \hat{\boldsymbol{v}}_i^t(I), \sigma_i^t(I) \rangle \mathbf{1}, \sigma_i(I) - \boldsymbol{\theta}_I^t \rangle$$
$$= \langle -\hat{\boldsymbol{v}}_i^t(I), \sigma_i^t(I) - \sigma_i(I) \rangle,$$

where the second equality comes from that

$$\langle \hat{\boldsymbol{v}}_i^t(I) - \langle \hat{\boldsymbol{v}}_i^t(I), \sigma_i^t(I) \rangle \mathbf{1}, \boldsymbol{\theta}_I^t \rangle = \langle \hat{\boldsymbol{v}}_i^t(I) - \langle \hat{\boldsymbol{v}}_i^t(I), \frac{\boldsymbol{\theta}_I^t}{\langle \boldsymbol{\theta}_I^t, \mathbf{1} \rangle} \rangle \mathbf{1}, \boldsymbol{\theta}_I^t \rangle = 0,$$
$$\langle \hat{\boldsymbol{v}}_i^t(I) - \langle \hat{\boldsymbol{v}}_i^t(I), \sigma_i^t(I) \rangle \mathbf{1}, \sigma_i(I) \rangle = \langle \hat{\boldsymbol{v}}_i^t(I), \sigma_i(I) - \sigma_i^t(I) \rangle.$$

Therefore, we have

$$\eta \langle -\hat{\boldsymbol{v}}_i^t(I), \sigma_i^t(I) - \sigma_i(I) \rangle$$
$$\leq D_\psi(\sigma_i(I), \boldsymbol{\theta}_I^t) - D_\psi(\sigma_i(I), \boldsymbol{\theta}_I^{t+1}) - D_\psi(\boldsymbol{\theta}_I^{t+1}, \boldsymbol{\theta}_I^t) + \eta \langle \hat{\boldsymbol{m}}_i^t(I), \boldsymbol{\theta}_I^{t+1} - \boldsymbol{\theta}_I^t \rangle. \tag{34}$$

Continuing from Eq. (34), we have

$$\eta \langle -\hat{\boldsymbol{v}}_i^t(I), (1 - \alpha_I)\sigma_i^t(I) - (1 - \alpha_I)\sigma_i(I) \rangle$$
$$\leq (1 - \alpha_I)\left( D_\psi(\sigma_i(I), \boldsymbol{\theta}_I^t) - D_\psi(\sigma_i(I), \boldsymbol{\theta}_I^{t+1}) + \eta \langle \hat{\boldsymbol{m}}_i^t(I), \boldsymbol{\theta}_I^{t+1} - \boldsymbol{\theta}_I^t \rangle \right),$$

which implies

$$\eta \langle -\hat{\boldsymbol{v}}_i^t(I), (1 - \alpha_I)\sigma_i^t(I) + \gamma \mathbf{1} - (1 - \alpha_I)\sigma_i(I) - \gamma \mathbf{1} \rangle$$
$$\leq (1 - \alpha_I)\left( D_\psi(\sigma_i(I), \boldsymbol{\theta}_I^t) - D_\psi(\sigma_i(I), \boldsymbol{\theta}_I^{t+1}) + \eta \langle \hat{\boldsymbol{m}}_i^t(I), \boldsymbol{\theta}_I^{t+1} - \boldsymbol{\theta}_I^t \rangle \right).$$

Therefore, we get

$$\eta \langle -\hat{\boldsymbol{v}}_i^t(I), \hat{\sigma}_i^t(I) - \hat{\sigma}_i(I) \rangle \leq (1 - \alpha_I)\left( D_\psi(\sigma_i(I), \boldsymbol{\theta}_I^t) - D_\psi(\sigma_i(I), \boldsymbol{\theta}_I^{t+1}) + \eta \langle \hat{\boldsymbol{m}}_i^t(I), \boldsymbol{\theta}_I^{t+1} - \boldsymbol{\theta}_I^t \rangle \right).$$

Continuing from Eq. (34), we have

$$\eta \pi_i^{\hat{\sigma}}(I) \langle -\hat{\boldsymbol{v}}_i^t(I), \hat{\sigma}_i^t(I) - \hat{\sigma}_i(I) \rangle$$
$$\leq (1 - \alpha_I)\pi_i^{\hat{\sigma}}(I)\left( D_\psi(\sigma_i(I), \boldsymbol{\theta}_I^t) - D_\psi(\sigma_i(I), \boldsymbol{\theta}_I^{t+1}) + \eta \langle \hat{\boldsymbol{m}}_i^t(I), \boldsymbol{\theta}_I^{t+1} - \boldsymbol{\theta}_I^t \rangle \right).$$

By applying Lemma 4.3, we get

$$\sum_{t=1}^T \langle \hat{\boldsymbol{\ell}}_i^t, \hat{\boldsymbol{x}}_i^t - \hat{\boldsymbol{x}}_i \rangle$$
$$\leq \sum_{t=1}^T \sum_{i \in \mathcal{N}} \sum_{I \in \mathcal{I}_i} (1 - \alpha_I)\pi_i^{\hat{\sigma}}(I)\left( \frac{D_\psi(\sigma_i(I), \boldsymbol{\theta}_I^t)}{\eta} - \frac{D_\psi(\sigma_i(I), \boldsymbol{\theta}_I^{t+1})}{\eta} + \langle \hat{\boldsymbol{m}}_i^t(I), \boldsymbol{\theta}_I^{t+1} - \boldsymbol{\theta}_I^t \rangle \right),$$

where $\hat{\boldsymbol{x}}_i$ is the sequence-form strategy corresponding to $\hat{\sigma}_i$. Using $\xi_I$ to denote $(1 - \alpha_I)\pi_i^{\hat{\sigma}}(I)$, we get

$$
\begin{aligned}
\sum_{t=1}^{T}\langle\hat{\boldsymbol{\ell}}_i^t,\hat{\boldsymbol{x}}_i^t-\hat{\boldsymbol{x}}_i\rangle &\leq \sum_{t=1}^{T}\sum_{i\in\mathcal{N}}\sum_{I\in\mathcal{I}_i}\xi_I\left(\frac{D_\psi(\sigma_i(I),\boldsymbol{\theta}_I^t)}{\eta}-\frac{D_\psi(\sigma_i(I),\boldsymbol{\theta}_I^{t+1})}{\eta}+\langle\hat{\boldsymbol{m}}_i^t(I),\boldsymbol{\theta}_I^{t+1}-\boldsymbol{\theta}_I^t\rangle\right)\\
&\leq \sum_{t=1}^{T}\sum_{i\in\mathcal{N}}\sum_{I\in\mathcal{I}_i}\xi_I\left(\frac{D_\psi(\sigma_i(I),\boldsymbol{\theta}_I^t)}{\eta}-\frac{D_\psi(\sigma_i(I),\boldsymbol{\theta}_I^{t+1})}{\eta}+\|\hat{\boldsymbol{m}}_i^t(I)\|_2\|\boldsymbol{\theta}_I^{t+1}-\boldsymbol{\theta}_I^t\rangle\|_2\right).
\end{aligned}
\tag{35}
$$

**Lemma F.2** (Adapted from Lemma 11 of Wei et al. [2021])**.** *. If the player $i$ follow the update rule of $CFR^+$, with $\eta > 0$ then for any $I \in \mathcal{I}_i$ and $t \geq 1$, we have*

$$
\|\boldsymbol{\theta}_I^{t+1} - \boldsymbol{\theta}_I^t\|_2 \leq \eta\|\hat{\boldsymbol{m}}_i^t(I)\|_2.
$$

By substituting Lemma F.2 into Eq. (35), we get

$$
\sum_{t=1}^{T}\langle\hat{\boldsymbol{\ell}}_i^t,\hat{\boldsymbol{x}}_i^t-\hat{\boldsymbol{x}}_i\rangle \leq \sum_{t=1}^{T}\sum_{i\in\mathcal{N}}\sum_{I\in\mathcal{I}_i}\xi_I\left(\frac{D_\psi(\sigma_i(I),\boldsymbol{\theta}_I^t)}{\eta}-\frac{D_\psi(\sigma_i(I),\boldsymbol{\theta}_I^{t+1})}{\eta}+\eta\|\hat{\boldsymbol{m}}_i^t(I)\|_2^2\right).
$$

Assuming $\|\hat{\boldsymbol{m}}_i^t(I)\|_2^2 \leq M$, we have

$$
\begin{aligned}
\sum_{t=1}^{T}\langle\hat{\boldsymbol{\ell}}_i^t,\hat{\boldsymbol{x}}_i^t-\hat{\boldsymbol{x}}_i\rangle &\leq \sum_{t=1}^{T}\sum_{i\in\mathcal{N}}\sum_{I\in\mathcal{I}_i}\xi_I\left(\frac{D_\psi(\sigma_i(I),\boldsymbol{\theta}_I^t)}{\eta}-\frac{D_\psi(\sigma_i(I),\boldsymbol{\theta}_I^{t+1})}{\eta}+\eta M\right)\\
&\leq \sum_{i\in\mathcal{N}}\sum_{I\in\mathcal{I}_i}\xi_I\left(\frac{D_\psi(\sigma_i(I),\boldsymbol{\theta}_I^1)}{\eta}+\sum_{t=1}^{T}\eta M\right)
\end{aligned}
\tag{36}
$$

According to the analysis in Appendix D, we have that for any accumulated counterfactual regret sequence $\{\boldsymbol{\theta}_I^1,\boldsymbol{\theta}_I^2,\ldots,\boldsymbol{\theta}_I^t,\ldots\}$ generated by any $\boldsymbol{\theta}_I^1 \in \mathbb{R}_{\geq 0}^{|A(I)|}$ and $\eta > 0$, there exists a corresponding accumulated counterfactual regret sequence $\{\boldsymbol{\theta}_I^{1\prime},\boldsymbol{\theta}_I^{2\prime},\ldots,\boldsymbol{\theta}_I^{t\prime},\ldots\}$ generated by $\boldsymbol{\theta}_I^{1\prime} \in \mathbb{R}_{\geq 0}^{|A(I)|}$ and $\eta' > 0$, such that the resulting strategy profile sequence $\{\hat{\boldsymbol{x}}^1,\hat{\boldsymbol{x}}^2,\ldots,\hat{\boldsymbol{x}}^t,\ldots\}$ are identical, where $\boldsymbol{\theta}_I^{t\prime} = \eta'\boldsymbol{\theta}_I^t/\eta$. To analysis the convergence rate of the accumulated counterfactual regret sequence $\{\boldsymbol{\theta}_I^{1\prime},\boldsymbol{\theta}_I^{2\prime},\ldots,\boldsymbol{\theta}_I^{t\prime},\ldots\}$, from Eq. (36), we have

$$
\sum_{t=1}^{T}\langle\hat{\boldsymbol{\ell}}_i^t,\hat{\boldsymbol{x}}_i^t-\hat{\boldsymbol{x}}_i\rangle \leq \sum_{i\in\mathcal{N}}\sum_{I\in\mathcal{I}_i}\xi_I\left(\frac{D_\psi(\sigma_i(I),\boldsymbol{\theta}_I^{1\prime})}{\eta'}+\eta'TM\right).
\tag{37}
$$

By substituting $\boldsymbol{\theta}_I^{t\prime} = \eta'\boldsymbol{\theta}_I^t/\eta$ into Eq. (37), we get

$$
\sum_{t=1}^{T}\langle\hat{\boldsymbol{\ell}}_i^t,\hat{\boldsymbol{x}}_i^t-\hat{\boldsymbol{x}}_i\rangle \leq \sum_{i\in\mathcal{N}}\sum_{I\in\mathcal{I}_i}\xi_I\left(\frac{D_\psi(\sigma_i(I),\frac{\eta'\boldsymbol{\theta}_I^t}{\eta})}{\eta'}+\eta'TM\right).
$$

From the fact that $\forall \boldsymbol{a},\boldsymbol{b}\in\mathbb{R}^d$, $\|\boldsymbol{a}-\boldsymbol{b}\|_2^2/2 = \|\boldsymbol{b}-\boldsymbol{a}\|_2^2/2 = D_\psi(\boldsymbol{a},\boldsymbol{b})$, by using $\boldsymbol{a} = \sigma_i(I)$ and $\boldsymbol{b} = \frac{\eta'\boldsymbol{\theta}_I^1}{\eta}$, we get

$$
\sum_{t=1}^{T}\langle\hat{\boldsymbol{\ell}}_i^t,\hat{\boldsymbol{x}}_i^t-\hat{\boldsymbol{x}}_i\rangle \leq \sum_{i\in\mathcal{N}}\sum_{I\in\mathcal{I}_i}\xi_I\left(\frac{\|\sigma_i(I)\|_2^2}{2\eta'}+\frac{(\eta'\boldsymbol{\theta}_I^1)^2}{2\eta'\eta^2}+\eta'TM\right).
\tag{38}
$$

As $\sigma_i(I) \in \Delta^{|A(I)|}$, we have $\|\sigma_i(I)\|_2^2 \leq 1$. In addition, $\|\eta'\boldsymbol{\theta}_I^1\|_2^2/(2\eta'\eta^2) = \eta'\|\boldsymbol{\theta}_I^1\|_2^2/(2\eta^2)$. Continuing from Eq. (38), we get

$$
\sum_{t=1}^{T}\langle\hat{\boldsymbol{\ell}}_i^t,\hat{\boldsymbol{x}}_i^t-\hat{\boldsymbol{x}}_i\rangle \leq \sum_{i\in\mathcal{N}}\sum_{I\in\mathcal{I}_i}\xi_I\left(\frac{1}{2\eta'}+\eta'\frac{\|\boldsymbol{\theta}_I^1\|_2^2}{2\eta^2}+\eta'TM\right).
$$

Table 2: Sizes of the games.

| Game | #Histories | #Infosets | #Terminal histories | #Depth | #Max size of infosets |
|---|---|---|---|---|---|
| Kuhn Poker | 58 | 12 | 30 | 6 | 2 |
| Leduc Poker | 9,457 | 936 | 5,520 | 12 | 5 |
| Battleship (3) | 732,607 | 81,027 | 552,132 | 9 | 7 |
| Liar's Dice (4) | 8,181 | 1,024 | 4,080 | 12 | 4 |
| Liar's Dice (5) | 51,181 | 5,120 | 25,575 | 14 | 5 |
| Liar's Dice (6) | 294,883 | 24,576 | 147,420 | 16 | 6 |
| Goofspiel (4) | 1,077 | 162 | 576 | 7 | 14 |
| Goofspiel (5) | 26,931 | 2,124 | 14,400 | 9 | 46 |
| Goofspiel (6) | 969,523 | 34,482 | 518,400 | 11 | 230 |
| Subgame 3 | 398,112,843 | 69,184 | 261,126,360 | 10 | 1,980 |
| Subgame 4 | 244,005,483 | 43,240 | 158,388,120 | 8 | 1,980 |

We use $M_\eta^{\boldsymbol{\theta}}$ to denote $\max(\|\boldsymbol{\theta}_I^1\|_2^2/(2\eta^2), M)$. In addition, as $0 \le (1 - \alpha_I) \le 1$ and $0 \le \pi_i^{\hat{\sigma}}(I) \le 1$, we have $0 \le \xi_I \le 1$. Therefore, we get

$$\sum_{t=1}^T \langle \hat{\boldsymbol{\ell}}_i^t, \hat{\boldsymbol{x}}_i^t - \hat{\boldsymbol{x}}_i \rangle \le \sum_{i \in \mathcal{N}} \sum_{I \in \mathcal{I}_i} \left( \frac{1}{2\eta'} + \eta'(T+1)M_\eta^{\boldsymbol{\theta}} \right).$$

By setting $\eta' = 1/\sqrt{2(T+1)M_\eta^{\boldsymbol{\theta}}}$, we have $\sum_{t=1}^T \langle \hat{\boldsymbol{\ell}}_i^t, \hat{\boldsymbol{x}}_i^t - \hat{\boldsymbol{x}}_i \rangle \le \sqrt{2(T+1)M_\eta^{\boldsymbol{\theta}}}|\mathcal{I}| \le \sqrt{4TM_\eta^{\boldsymbol{\theta}}}|\mathcal{I}|$ with any $\boldsymbol{\theta}_I^1 \in \mathbb{R}_{\ge 0}^{|A(I)|}$ and $\eta > 0$. It completes the proof. $\qquad\square$

# G  Additional Experiments

**Sizes of the Games.** Before introducing our additional experiments, we present the sizes of the games used in our study, as detailed in Table 2. In this table, #Histories denotes the total number of histories within the game tree, whereas #Infosets represents the count of information sets. The term #Terminal histories indicates the number of leaf nodes, and #Depth refers to the game's tree depth, defined as the maximum sequence of actions in any single history. Finally, #Max size of infosets signifies the largest number of histories contained within a single infoset.

**Performance of RTCFR$^+$ under simultaneous decrease of $\mu$ and $\gamma$.** we present the results for RTCFR$^+$ with modifications in line 8 where $\mu \times (1 - \varsigma), \gamma \leftarrow \gamma \times 0.5$, and $\boldsymbol{r} \leftarrow \hat{\boldsymbol{x}}^{T_u+1}$, with $\varsigma = 1\mathrm{e} - 16$, as shown in Figure 3. We denote this variant as "RTCFR$^+$ V2". Our findings reveal that the empirical convergence performance of RTCFR$^+$ and RTCFR$^+$ V2 is similar.

**Performance of RTCFR$^+$ under reset accumulated regrets as 0.** we examine the performance of RTCFR$^+$ that resets $\boldsymbol{\theta}_I^1$ to $\mathbf{0}$, which is denoted as "Unstable RTCFR$^+$" in Figure 3. The parameters of Unstable RTCFR$^+$ are same as RTCFR$^+$ in Section 5. We observe that Unstable RTCFR$^+$ never converges across all tested games.

**Comparison with average-iterate convergence CFR algorithms.** We compare the last-iterate convergence performance of RTCFR$^+$ with the average-iterate performance of CFR$^+$, PCFR$^+$, and DCFR. The experimental results are shown in Figure 4. With fine-tuning, RTCFR$^+$ outperforms the average-iterate performance of CFR+, PCFR+, and DCFR in nearly all tested games, except for Liar's Dice (6). Even without fine-tuning, RTCFR$^+$ achieves superior performance to the average-iterate of CFR+$^+$, PCFR+, and DCFR in 5 out of the evaluated 9 games (Kuhn Pker, Leduc Poker, Liar's Dice (4), Liar's Dice (5), and Goofspiel (4)). In addition, as shown in Figure 4, even when considering only CFR$^+$, PCFR$^+$, and DCFR, no single algorithm consistently outperforms the other two across all games.

**Convergence rates in the initial phase.** We now present the results of our algorithms RTCFR$^+$ and RTPCFR$^+$, alongside CFR$^+$, R-NaD, Reg-CFR, OMWU, OGDA, PCFR$^+$, and DCFR, over the first 1000 iterations. The results are shown in Figure 5. Consistent with the results in Figures 1, RTCFR$^+$, RTPCFR$^+$, and PCFR$^+$ demonstrate superior performance compared to the other algorithms. However, no single algorithm without fine-tuning outperforms all others across all games.

**Performance of RTCFR$^+$ in HUNL Subgames.** We now present the convergence rate of RTCFR$^+$ within HUNL Subgames, particularly the ones open-sourced by Libratus [Brown and Sandholm,

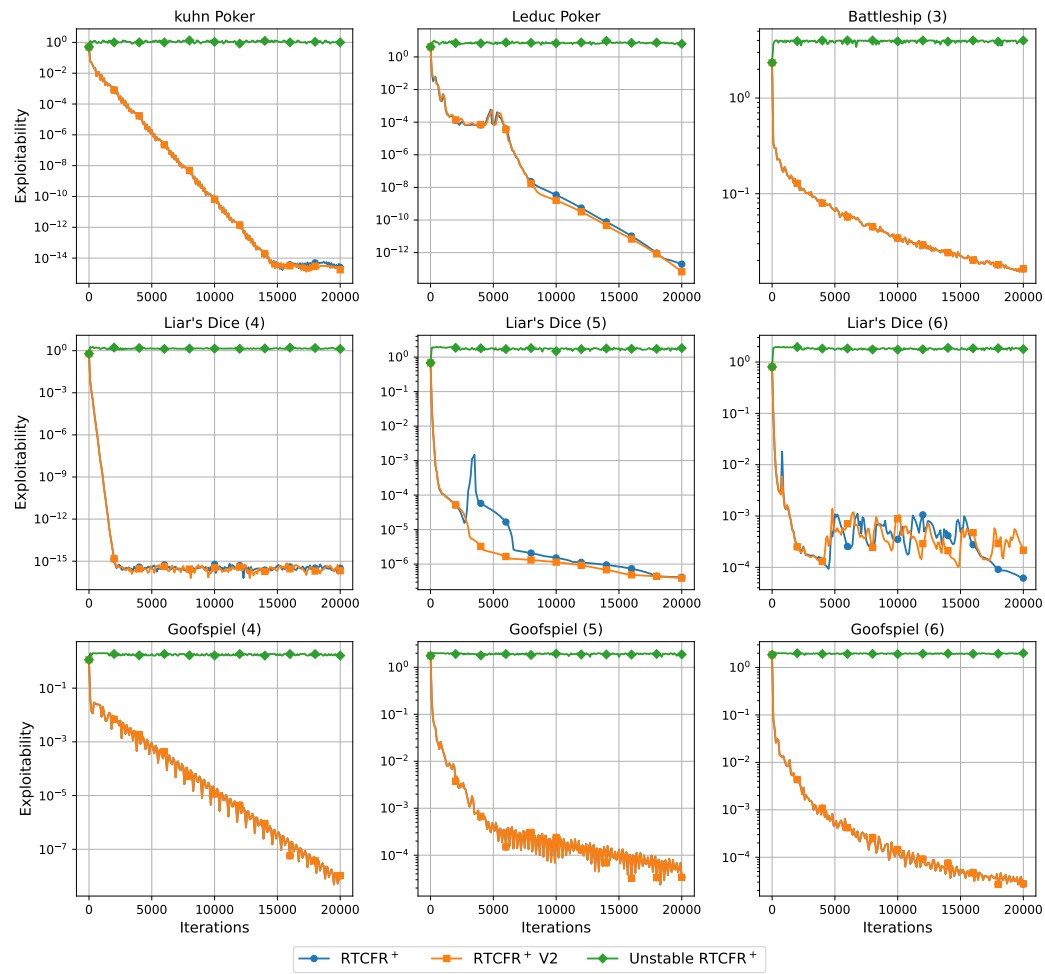

Figure 3: Last-iterate convergence rates of RTCFR$^+$, RTCFR$^+$ V2, and Unstable RTCFR$^+$.

2018]. We compare RTCFR$^+$ with CFR$^+$, PCFR$^+$, and DCFR$^+$. Given the immense size of HUNL Subgames, we implement the tested algorithm using vector CFR. We employ the open-source code from Poker RL [Steinberger, 2019, Xu et al., 2024b], which supports vector CFR and Subgames from Libratus, specifically Subgame 3 and Subgame 4. The comparison of RTCFR$^+$ and the last-iterate convergence performance of CFR$^+$, PCFR$^+$, and DCFR$^+$ is illustrated in Figure 6, while the comparison of RTCFR$^+$ and the average-iterate convergence performance of CFR$^+$, PCFR$^+$, and DCFR$^+$ is depicted in Figure 7. RTCFR$^+$ exceeds the last-iterate convergence performance of CFR$^+$ and PCFR$^+$ across both HUNL Subgames. Additionally, in Subgame3, RTCFR$^+$ also surpasses the average-iterate convergence performance of CFR$^+$ and PCFR$^+$. It is worth noting that CFR$^+$ and PCFR$^+$ do not provide a last-iterate convergence guarantee. For DCFR, RTCFR$^+$, as well as CFR$^+$ and PCFR$^+$, underperform in both last-iterate and average-iterate convergence performance. We speculate this is because DCFR is fine-tuned specifically for the tested HUNL Subgames, unlike the other evaluated algorithms.

**Performance of RTCFR$^+$ under different hyperparameters.** We investigate the convergence rates of RTCFR$^+$ under various hyperparameter settings. Specifically, we focus on the impact of $\mu$ and $T_u$ on the convergence rates, as we observe that $\gamma$ only needs to be set to a sufficiently small value. The tested ranges for $\mu$ and $T_u$ are $[1e-4,\ 5e-4,\ 1e-3,\ 5e-3,\ 1e-2,\ 5e-2,\ 1e-1,\ 5e-1]$ and $[10,\ 50,\ 100,\ 500,\ 1000]$, respectively. Experimental results reveal that the performance of RTCFR$^+$ is primarily contingent upon the value of $\mu$. To elucidate this dependency, we discuss the performance implications of varying $\mu$ values. Specifically, for small $\mu$ values, CFR$^+$ encounters difficulties in

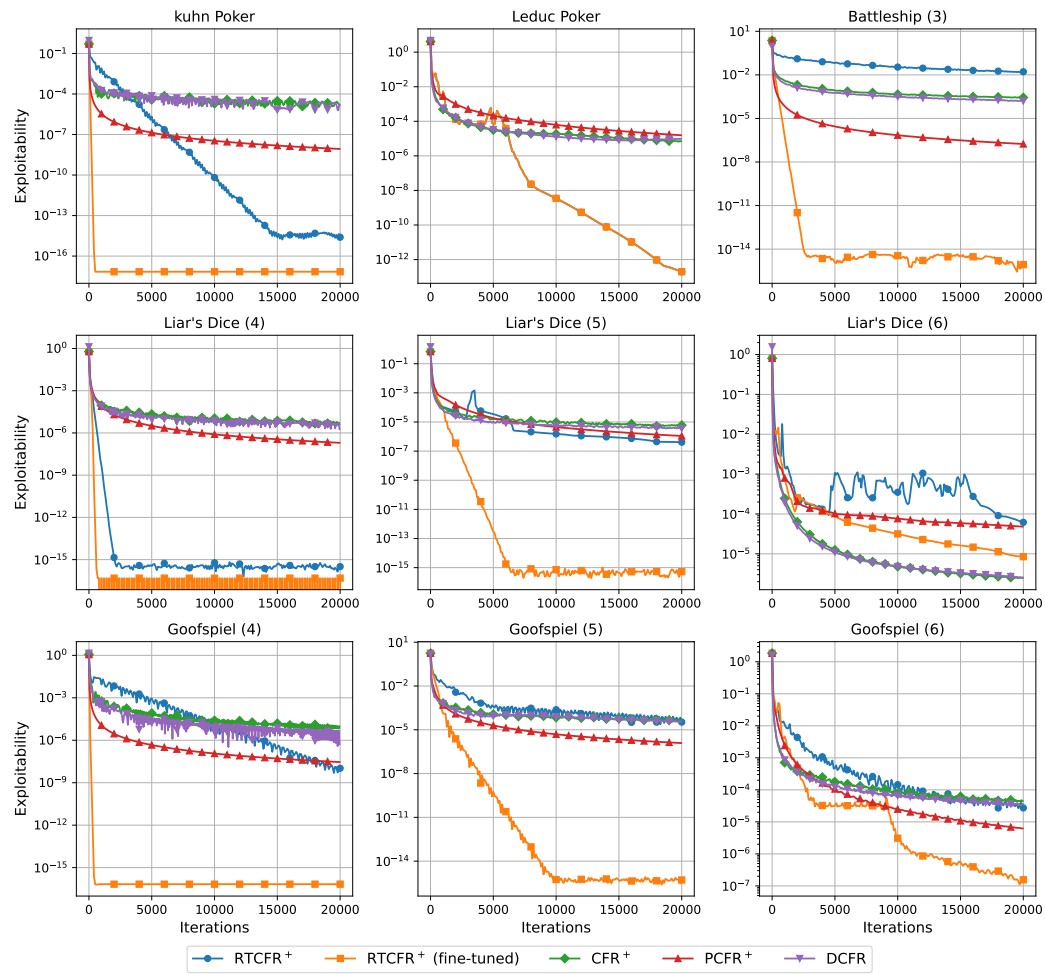

Figure 4: Comparison with classical average-iterate convergence CFR algorithms.

accurately learning an NE of perturbed regularized EFGs. Consequently, this challenge persists irrespective of the value of $T_u$, enabling that learning an NE of perturbed regularized EFGs becomes impossible. As a result, attaining an NE of the original game becomes impracticable for any $T_u$ value, which is also consistent with the experimental results. Conversely, when $\mu$ is optimal, neither too small nor too large, this condition enables $CFR^+$ to learn sufficiently accurate approximate an NE of perturbed regularized EFGs. These allow $RTCFR^+$ to achieve commendable performance. However, for large $\mu$ values, although $CFR^+$ are capable of learning the exact NE of perturbed regularized EFGs, the requisite number of reference strategy updates becomes excessively large. Hence, we observe that with large $\mu$ values, a smaller $T_u$ yields better performance. Based on these analyses, we advocate for the prioritization of determining $\mu$'s value, followed by the value of $T_u$, when practically applying our algorithm.

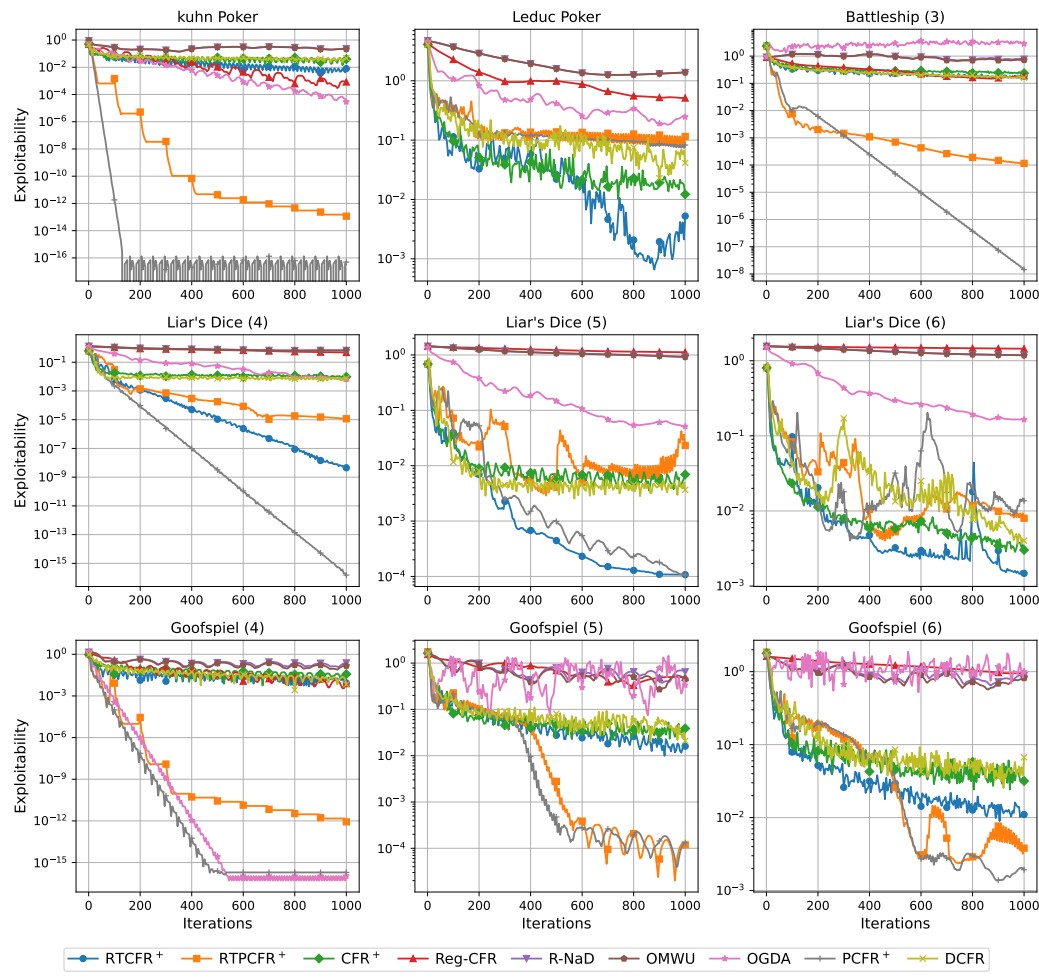

Figure 5: Last-iterate convergence rates over the first 1000 iterations.

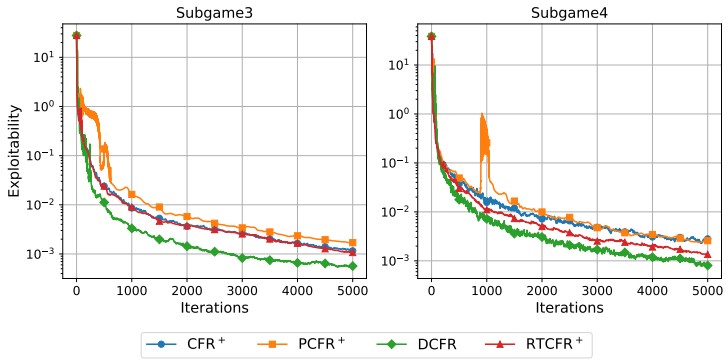

Figure 6: Comparison with the last-iterate convergence performance of CFR$^+$, PCFR$^+$, and DCFR in HUNL Subgames.

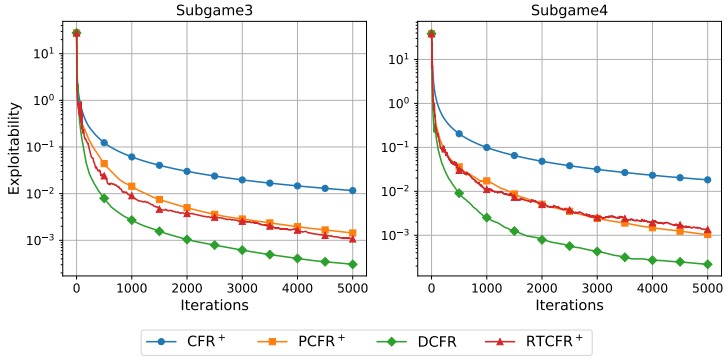

Figure 7: Comparison with the average-iterate convergence performance of CFR$^+$, PCFR$^+$, and DCFR in HUNL Subgames.

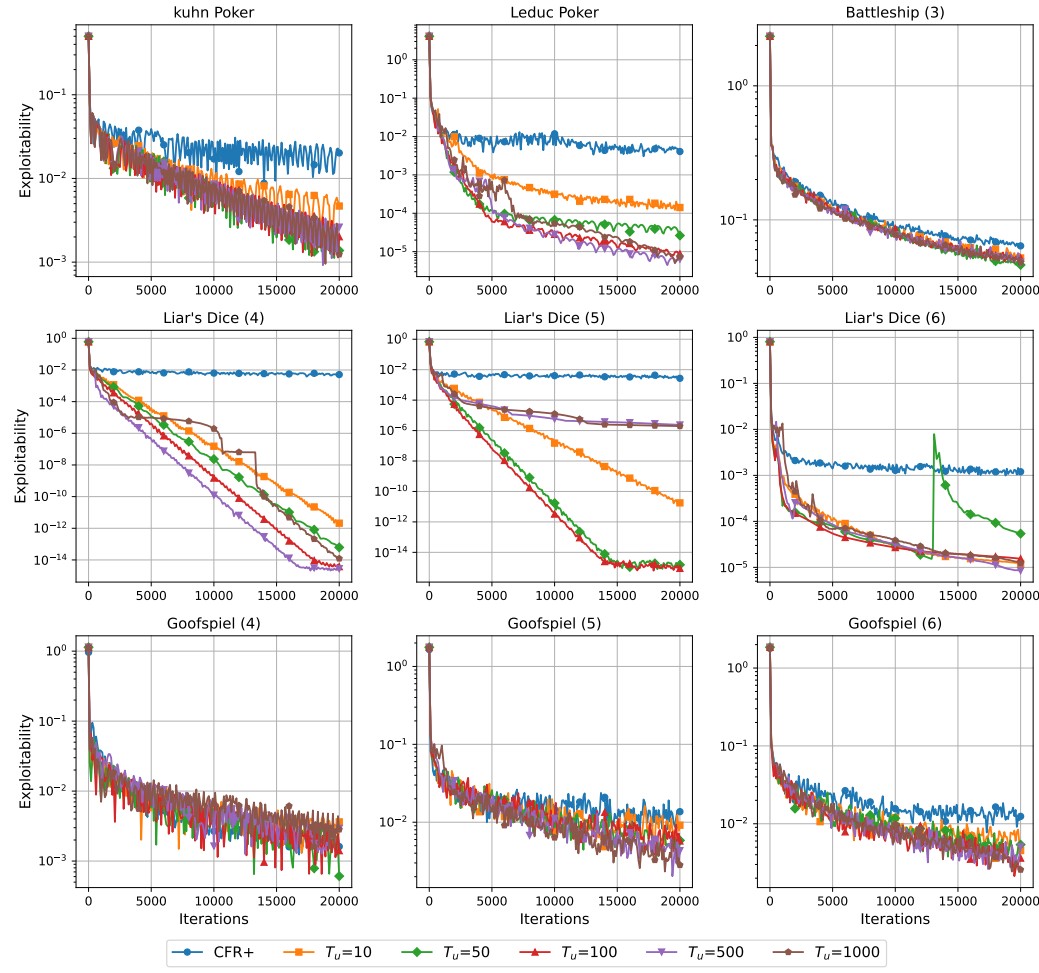

Figure 8: Last-iterate convergence rates of RTCFR$^+$ with $\mu = 0.0001$.

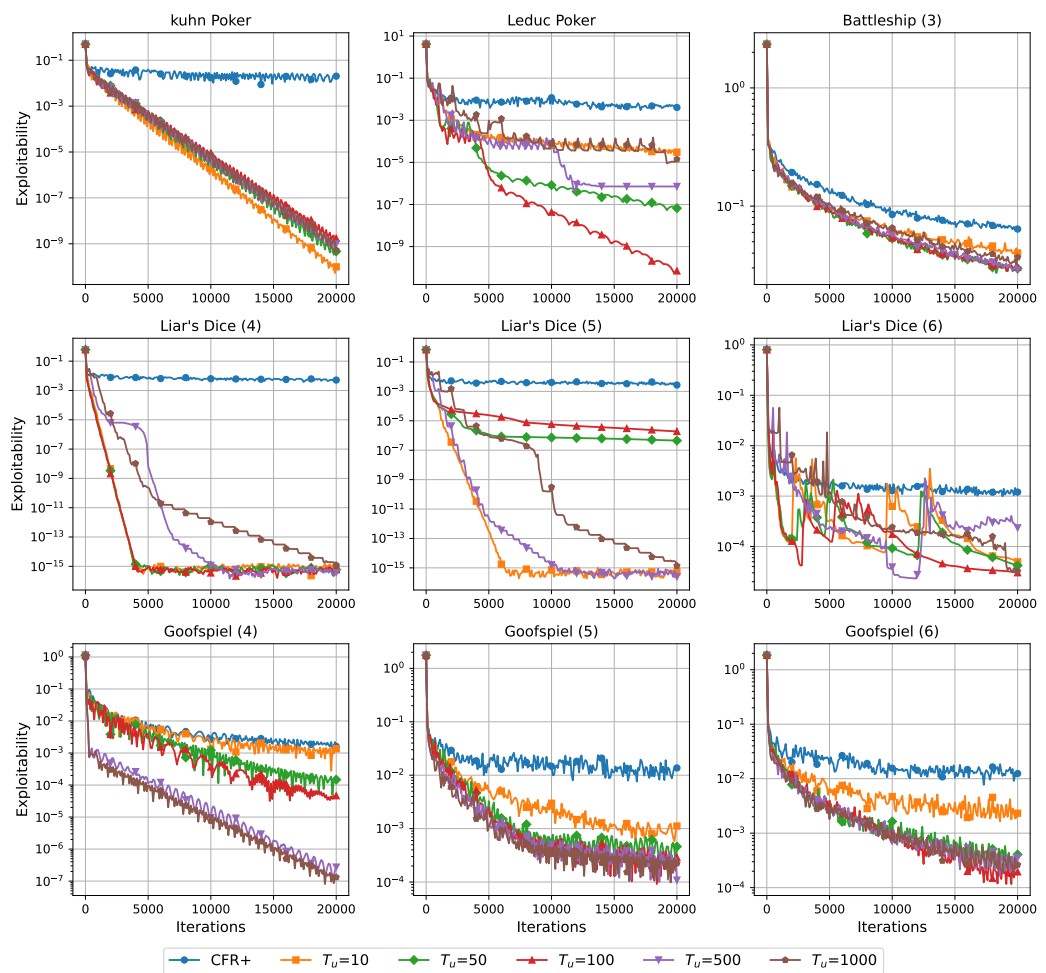

Figure 9: Last-iterate convergence rates of RTCFR$^+$ with $\mu = 0.0005$.

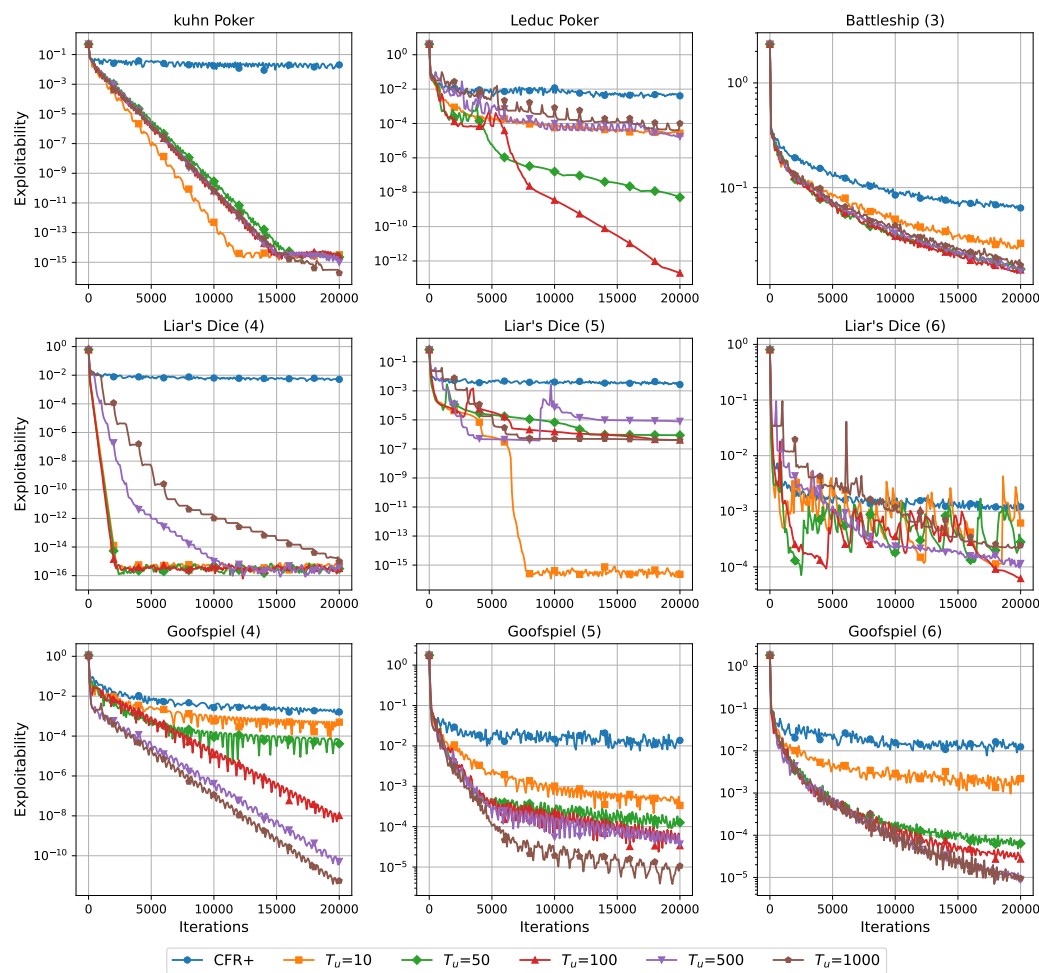

Figure 10: Last-iterate convergence rates of RTCFR$^+$ with $\mu = 0.001$.

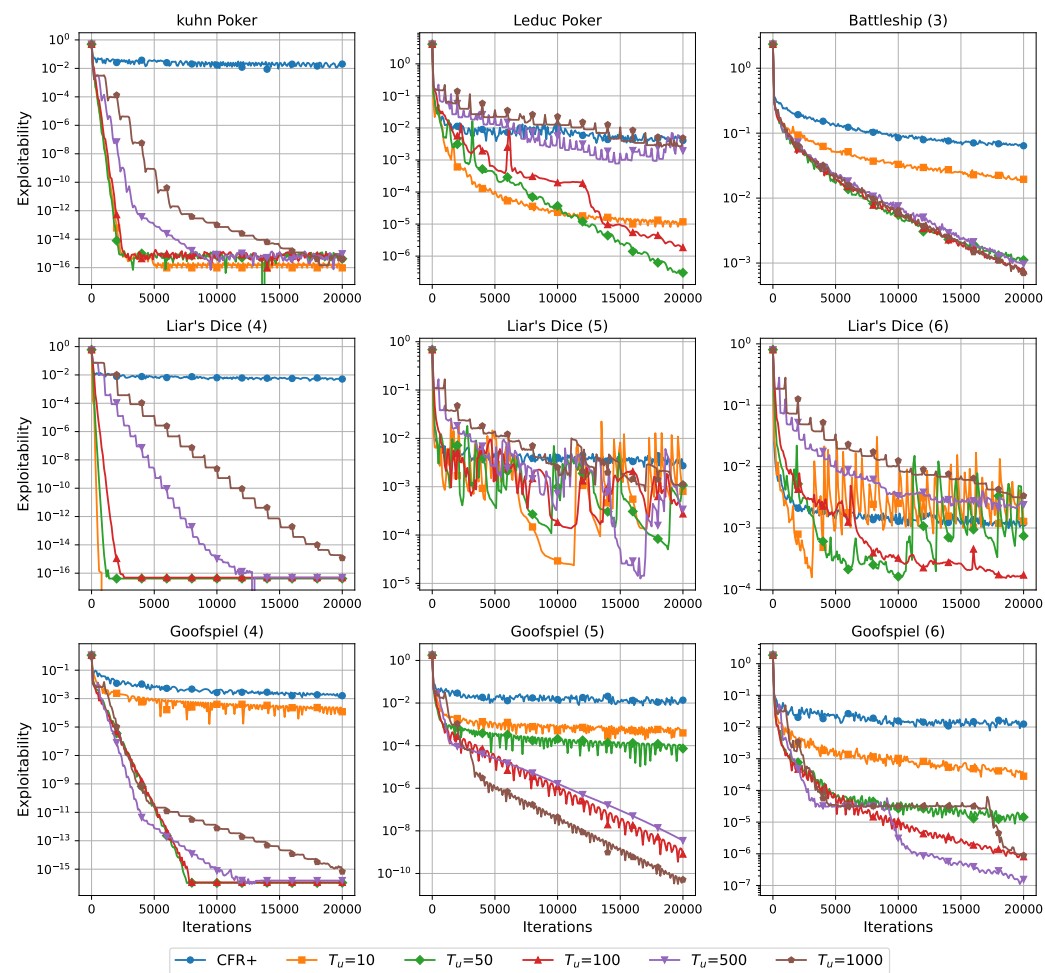

Figure 11: Last-iterate convergence rates of RTCFR$^+$ with $\mu = 0.005$.

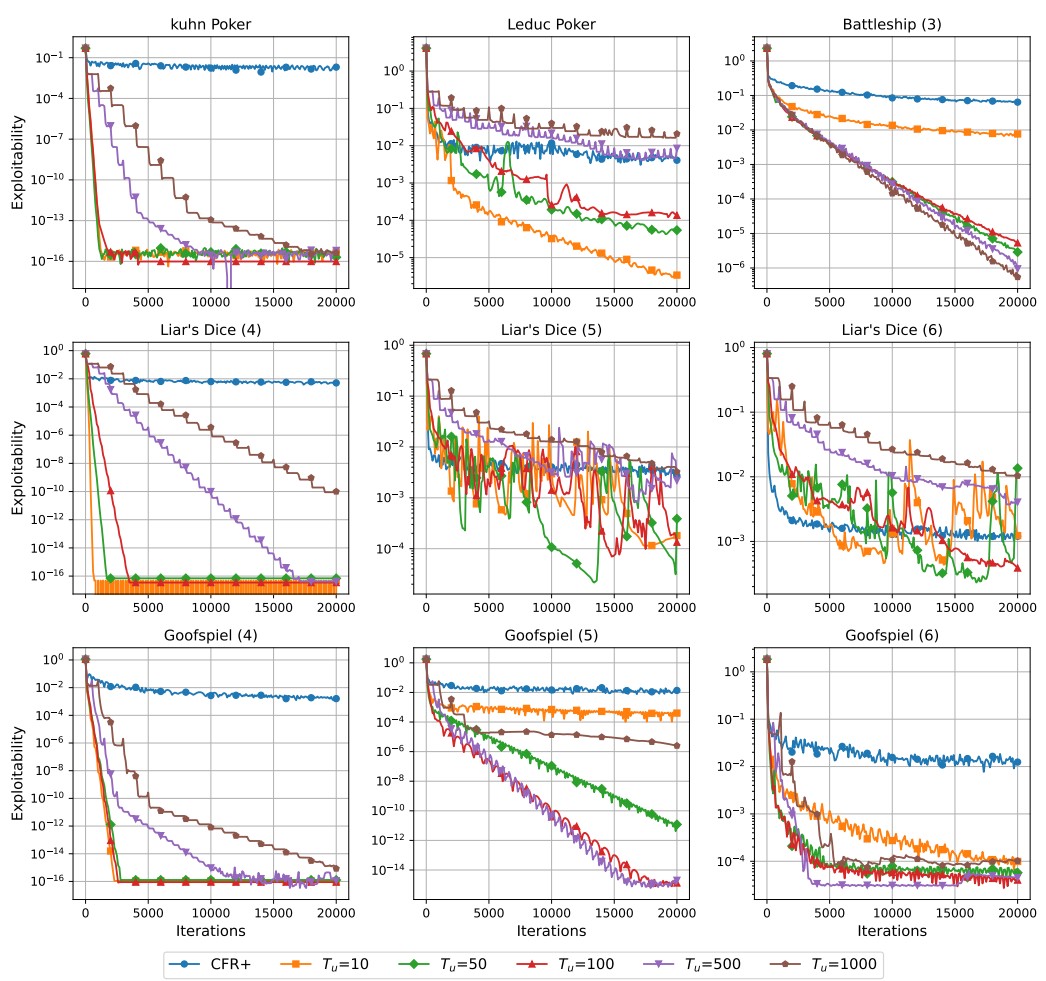

Figure 12: Last-iterate convergence rates of RTCFR$^+$ with $\mu = 0.01$.

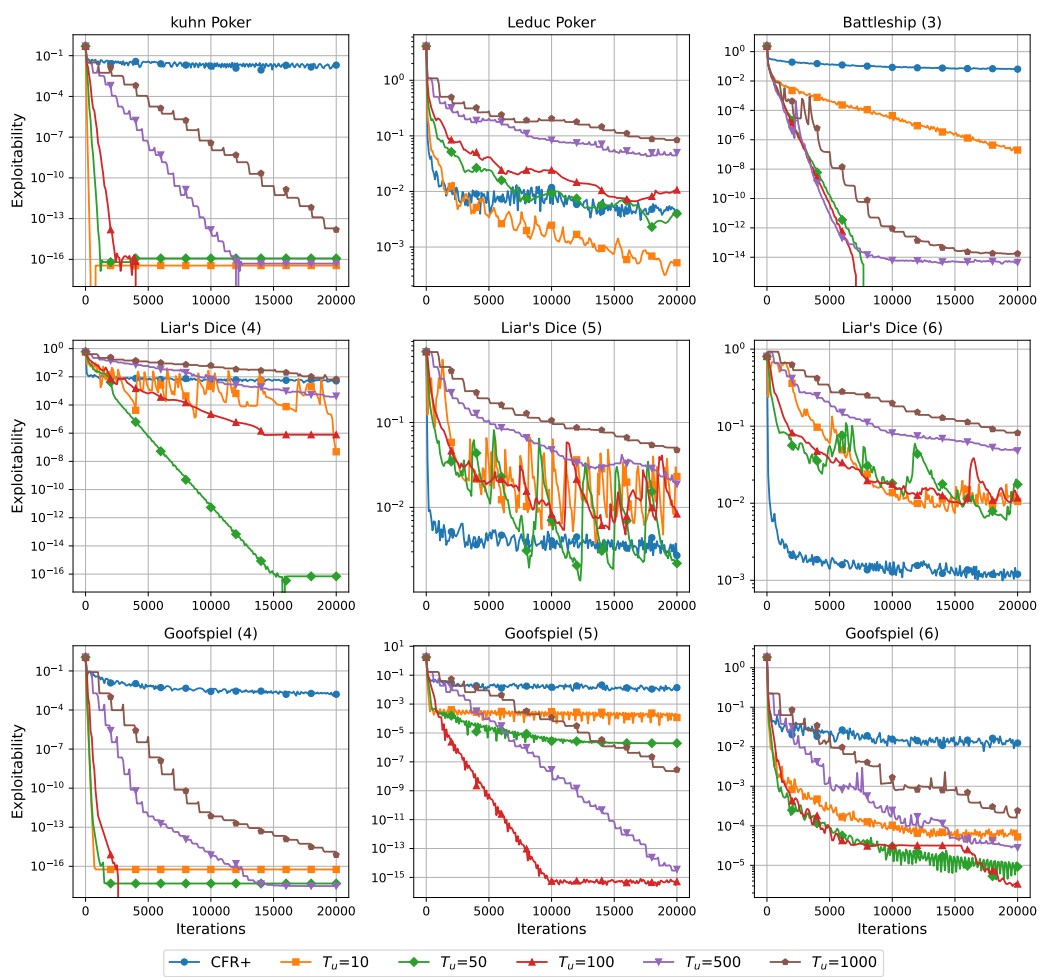

Figure 13: Last-iterate convergence rates of RTCFR$^+$ with $\mu = 0.05$.

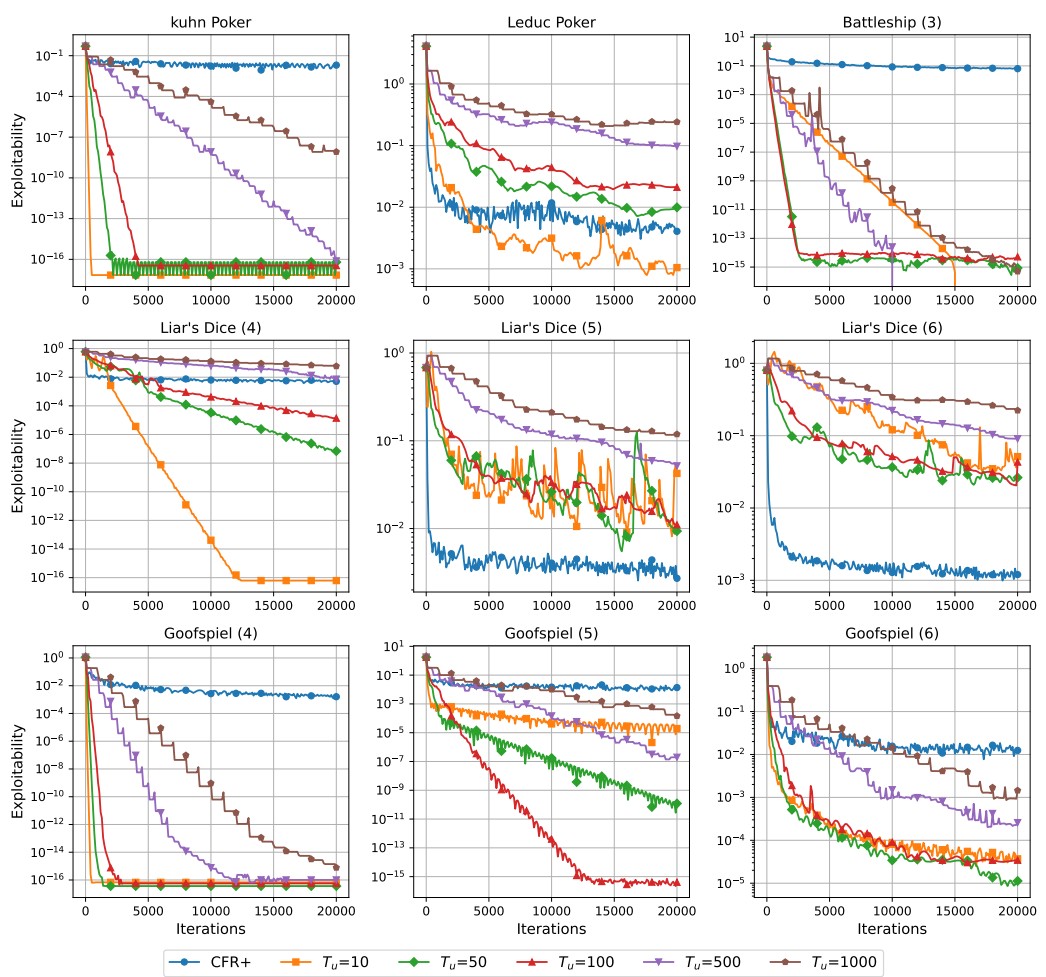

Figure 14: Last-iterate convergence rates of RTCFR$^+$ with $\mu = 0.1$.

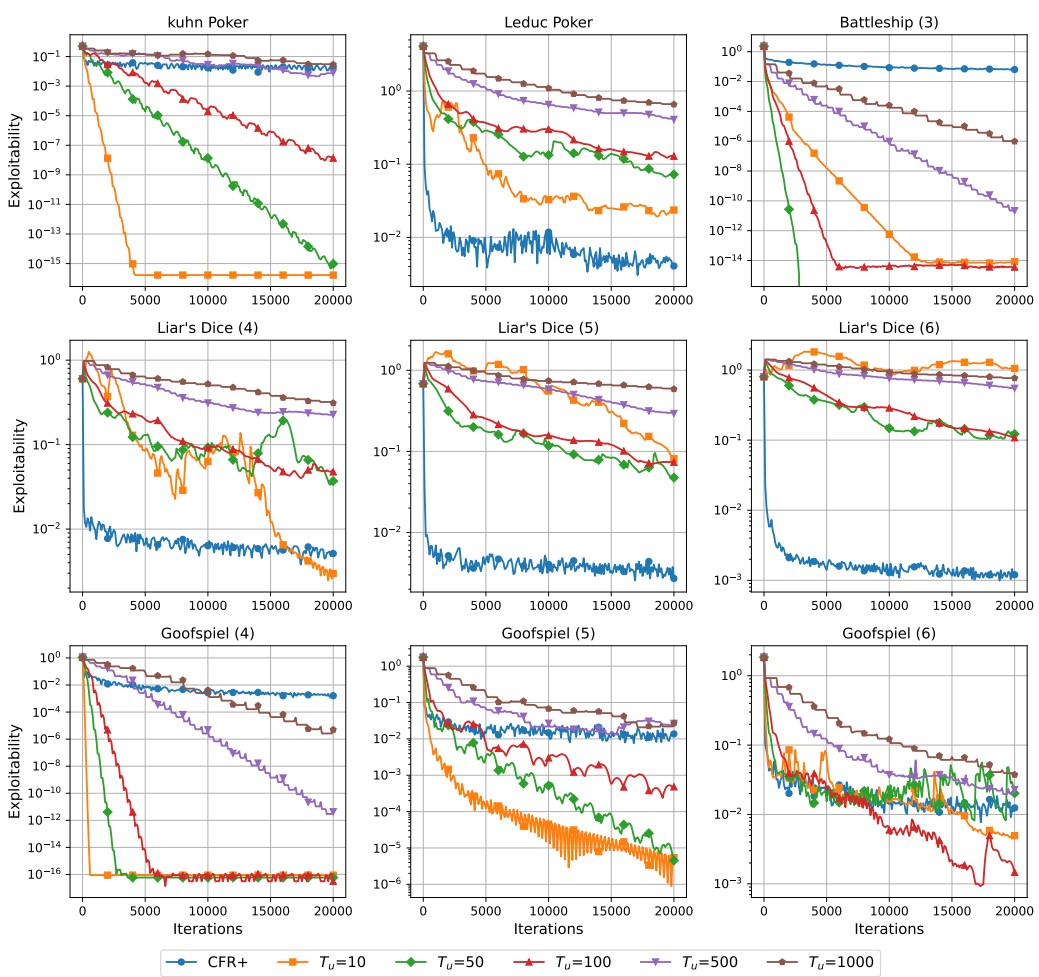

Figure 15: Last-iterate convergence rates of RTCFR$^+$ with $\mu = 0.5$.

# H  Implementation of RTCFR$^+$

In this section, we present a detailed description of the implementation of RTCFR$^+$, which is adapted from the open-source implementation of CFR$^+$ by LiteEFG [Liu et al., 2024].

```python
import LiteEFG
class RTCFRPlusGraph(LiteEFG.Graph):
    def __init__(self, gamma=1e-10, mu=1e-3, shrink_iter=100): #
        default parameters
        super().__init__()
        self.timestep = 0
        self.shrink_iter = shrink_iter # shrink_iter is T_u

        # Initialization of RTCFR+
        with LiteEFG.backward(is_static=True):
            ev = 1.0 * LiteEFG.const(1, 0.0)
            # unperturbed_strategy is \sigma
            self.unperturbed_strategy = LiteEFG.const(self.
                action_set_size, 1.0 / self.action_set_size)
            # perturbed_strategy is \hat{\sigma}
            self.strategy = LiteEFG.const(self.action_set_size,
                1.0 / self.action_set_size)
            # regret_buffer is \bm{\theta}
            self.regret_buffer = LiteEFG.const(self.
                action_set_size, 0.0)

            # ref_strategy is \bm{r}
            self.ref_strategy = LiteEFG.const(self.action_set_size
                , 1.0 / self.action_set_size)
            # the following three variables are used to compute \
                nabla \psi(\bm{r}), note that self.ref_reach_prob(I
                ) = \nabla \psi(\bm{r})(I)
            self.ref_reach_prob = LiteEFG.const(self.
                action_set_size, 1.0)
            self.parent_reach_prob = LiteEFG.const(self.
                action_set_size, 1.0)
            self.parent_to_child_prob = LiteEFG.const(self.
                action_set_size, 1.0)

            self.iteration = LiteEFG.const(1, 0)
            self.mu = LiteEFG.const(1, mu)
            self.gamma = LiteEFG.const(1, gamma)
            self.alpha_I = self.gamma*self.action_set_size

        with LiteEFG.backward(color=0):
            self.iteration.inplace(self.iteration+1)
            # to compute the \hat{\bm{v}}_i^t(I) defined in (4)
            gradient = LiteEFG.aggregate(ev, aggregator="sum") +
                self.utility - self.mu*(self.reach_prob*self.
                strategy - self.ref_reach_prob*self.ref_strategy)
            # to compute the \langle \hat{\bm{v}}_i^t(I), \sigma^
                t_i(I) \rangle defined in (4)
            ev.inplace(LiteEFG.dot(gradient, self.
                unperturbed_strategy))
            # gradient - ev is the instantaneous counterfactual
                regret \hat{\bm{m}}_i^t(I ) defined in (4)
            self.regret_buffer.inplace(LiteEFG.maximum(self.
                regret_buffer + gradient - ev, 0.0))

            # to get \sigma^{t+1}_i(I)
            self.unperturbed_strategy.inplace(LiteEFG.normalize(
                self.regret_buffer, p_norm=1.0, ignore_negative=
                True))
            # to employ PCFR+ to solve the perturbed regularized
                EFGs, please use the following line
```

```python
                # self.unperturbed_strategy.inplace(LiteEFG.normalize(
                #     self.regret_buffer + gradient - ev, p_norm=1.0,
                #     ignore_negative=True))
                # to get \hat{\sigma}^{t+1}_i(I)
                self.strategy.inplace(LiteEFG.normalize((1 - self.
                    alpha_I)*self.unperturbed_strategy + self.gamma,
                    p_norm=1.0, ignore_negative=True))

            # update gamma and the reference strategy profile
            with LiteEFG.backward(color=1):
                self.gamma.inplace(self.gamma * 0.5)
                self.ref_strategy.inplace(self.strategy * 1.0)

            with LiteEFG.forward(color=2):
                # to compute \nabla \psi(\bm{r}) after updating the
                #     reference strategy profile
                self.parent_reach_prob.inplace(LiteEFG.aggregate(self.
                    ref_reach_prob, "sum", object="parent", player="
                    self", padding=1))
                self.parent_to_child_prob.inplace(LiteEFG.aggregate(
                    self.ref_strategy, "sum", object="parent", player="
                    self", padding=1))
                self.ref_reach_prob.inplace(self.parent_reach_prob*
                    self.parent_to_child_prob)

            print("===============Graph is ready for RTCFR
                +===============")

    def update_graph(self, env : LiteEFG.Environment) -> None:
        self.timestep += 1
        if self.timestep==1:
            env.update(self.strategy, upd_color=[2])
        if self.timestep % self.shrink_iter == 0:
            env.update(self.strategy, upd_color=[1])
            env.update(self.strategy, upd_color=[2])
            env.update(self.strategy, upd_color=[0], upd_player=1)
            env.update(self.strategy, upd_color=[0], upd_player=2)
        else:
            env.update(self.strategy, upd_color=[0], upd_player=1)
            env.update(self.strategy, upd_color=[0], upd_player=2)

    def current_strategy(self, type_name="last-iterate") ->
        LiteEFG.GraphNode:
        return self.strategy
```

