# OpenReview forum: "Efficient Last-Iterate Convergence in Solving Extensive-Form Games"
_NeurIPS.cc/2025/Conference — NeurIPS 2025 poster_

### Official Review · Reviewer_L64t · 2025-06-06

**Clarity:** 2
**Significance:** 4
**Originality:** 2
**Rating:** 5
**Confidence:** 4

**Summary:**

This paper introduces RTCFR⁺, a step-size parameter-free RM-based CFR+ variant for last-iterate convergence to Nash equilibria in perturbed extensive-form games (EFGs). It aims to bridge two research gaps: (1) the slow empirical convergence of theoretically sound OMD-based CFR methods (e.g., Reg-CFR), and (2) the lack of last-iterate convergence guarantees for RM-based algorithms (e.g., CFR+) in perturbed regularized EFGs. By proving CFR+ achieves step-size parameter-free last-iterate convergence in this setting and embedding it in a Reward Transformation framework, RTCFR⁺ demonstrates faster empirical convergence than existing last-iterate methods across multiple benchmarks, though without new theoretical convergence guarantees for the overall approach.

**Questions:**

1. In Line 136, computation via the full matrix $\mathbf{A}$ suggests per-iteration complexity scales with histories rather than infosets or sparsification[1]. Does RTCFR⁺ leverage modern CFR optimizations (e.g., polyhedron merging for monotonic infosets) to achieve infoset-linear complexity?

2. Line 152 assumes NEs satisfy γ-perturbed constraints. How are NEs computed when optimal strategies violate γ-bounds (e.g., pure strategies)?

3. How is exploitability calculated under perturbation? Does it measure deviation in the original or perturbed game—particularly critical for Figures 1–3?

4. Algorithm 1 (Line 8) updates reference strategies at fixed intervals. Could pathological strategy sequences at update points prevent convergence? Is there theoretical assurance?

5. Why does Reg-CFR stagnate near 1e-5 exploitability in Kuhn Poker/Goofspiel (Fig 1)? Is this due to numerical precision limits, hyperparameters, or inherent properties?

[1] Farina et al. Fast Payoff Matrix Sparsification Techniques for Structured Extensive-Form Games. AAAI 2022.

**Ethical Concerns:**

["NO or VERY MINOR ethics concerns only"]

**Final Justification:**

I think the work design the first last iterative algorithm in EFG that performs no worse than average iterative algorithms in experiments, which I don't find in my first review. I raise my score accordingly.

**Limitations:**

This paper does not provide a theoretical bound for the number of iterations T.

**Quality:**

3

**Strengths And Weaknesses:**

**Strengths**

1. Establishes the first step-size parameter-free last-iterate convergence proof for RM-based CFR in perturbed regularized EFGs.

2. Comprehensive experiments across multiple EFG benchmarks show RTCFR⁺ significantly outperforms existing last-iterate convergence methods.

3. Directly addresses the pertinent question: "Does CFR+ have parameter-free last-iterate convergence in learning an NE of perturbed regularized EFGs?"

**Weaknesses**

1. Overstated claims: The "parameter-free" assertion applies only to step sizes (η). Critical hyperparameters (γ, μ, Tᵤ) in Algorithm 1 require tuning, contradicting the broader implication of parameter insensitivity. This overshadows the legitimate contribution regarding step-size independence.

2. Narrow applicability: The method operates under restrictive conditions (perturbed EFGs, RM-based local regret minimizers, last-iterate focus, step-size parameter freedom), severely limiting practical deployment scenarios.

3. Unclear practical utility: No comparison to average-iterate performance of CFR+/PCFR+—the dominant practical approaches. If RTCFR⁺ under last-iterate constraints fails to surpass these methods under standard average-iterate metrics, its real-world value diminishes despite theoretical novelty. In addition, I am not quite sure whether some efficient acceleration techniques can be applied to RTCFR+ (see Question 1). If these efficient acceleration techniques are not used, the contribution of the empirical results will be greatly reduced.

4. Insufficient distinction from prior work on parameter-free RM+ last-iterate convergence: The novelty of the proposed method requires clearer articulation, particularly in contrast to Cai et al. (2025) [1], which also established parameter-free last-iterate convergence for RM+ algorithms in games.

Line 132: the --> there

Line 141: This sentence is grammatically incorrect.

[1] Cai et al. Last-iterate convergence properties of regret-matching algorithms in games. ICLR 2025

---

> ### Author Rebuttal · Authors · 2025-07-31
>
> Thank you for your time and for sharing your critical perspective on our manuscript.
>
>
>
> **W1: Overstated claims. The "parameter-free" assertion applies only to step sizes (η). Critical hyperparameters (γ, μ, Tᵤ) in Algorithm 1 require tuning.**
>
> **A:** We agree that the "parameter-free" property specifically refers to the CFR+ solver's independence from step-size tuning (η), not the entire RTCFR+ framework. Our core contribution, as articulated in the Introduction, is indeed the first proof of this step-size-free last-iterate convergence for CFR+ in the context of perturbed regularized EFGs. To ensure this is perfectly clear to the reader, in the final version, we will explicitly state that RTCFR+ is a non-parameter-free algorithm under the limitations section.
>
> Secondly, our algorithm reduces the cost of parameter tuning. Since $\eta$ does not require adjustment, our algorithm is more applicable to solving real-world games compared to those algorithms that necessitate parameter tuning when solving perturbed regularized EFGs.
>
> Lastly, the parameters (γ, μ, Tᵤ) are inherent to the Reward Transformation (RT) framework itself, which our algorithm is built upon. Investigating the parameter-free nature of the RT framework is outside the scope of this paper.
>
>
>
> **W2: The method operates under restrictive conditions (perturbed EFGs, RM-based local regret minimizers, last-iterate focus, step-size parameter freedom)**
>
> **A:**
>
> - Our RTCFR+ is designed to solve original EFGs, rather than perturbed EFGs; the perturbations are introduced only as part of the procedure within RTCFR+. In other words, RTCFR+ is not limited to perturbed EFGs and remains applicable to the original EFGs.
> - A key advantage of RTCFR+ is its use of RM-based local regret minimizers, which replace the OMD-based minimizers employed in prior work (Pérolat et al., 2021; Liu et al., 2023) to improve the empirical last-iterate convergence rate.
> - Last-iterate convergence and a parameter-free design are not limitations; on the contrary, they are highly desirable properties for practical applications, yielding more stable and deployment-friendly algorithms.
>
>
> **W3: No comparison to average-iterate performance of CFR+/PCFR+**
>
> **A:** We compare the last-iterate convergence performance of RTCFR+ with the average-iterate performance of CFR+, PCFR+, and DCFR (Table 1). With fine-tuning, RTCFR+ outperforms the average-iterate performance of CFR+, PCFR+, and DCFR in nearly all tested games, except for Liar's Dice (6). Even without fine-tuning, RTCFR+ achieves superior performance to the average-iterate of CFR+, PCFR+, and DCFR in 5 out of the evaluated 9 games (Kuhn Pker, Leduc Poker, Liar's Dice (4),  Liar's Dice (5), and Goofspiel (4)). In addition, as shown in Table 1, even when considering only CFR+, PCFR+, and DCFR, no single algorithm consistently outperforms the other two across all games.
>
> Table 1: Comparison between the last-iterate convergence performance of RTCFR+ with the average-iterate convergence performance of CFR+, PCFR+, and DCFR.
>
> | Game                | RTCFR$^+$ | RTCFR$^+$ (fine-tuned) | CFR$^+$  | PCFR$^+$ | DCFR     |
> | :------------------ | :-------- | :--------------------- | :------- | :------- | :------- |
> | **Kuhn Poker**      | 2.49e-15  | 6.94e-18               | 1.49e-05 | 8.56e-09 | 1.59e-05 |
> | **Leduc Poker**     | 1.97e-13  | 1.97e-13               | 7.08e-06 | 1.56e-05 | 9.72e-06 |
> | **Battleship (3)**  | 1.64e-02  | 8.60e-16               | 2.74e-04 | 1.74e-07 | 1.58e-04 |
> | **Liar's Dice (4)** | 3.19e-16  | 4.86e-17               | 4.28e-06 | 1.99e-07 | 1.90e-06 |
> | **Liar's Dice (5)** | 4.12e-07  | 5.29e-16               | 6.00e-06 | 1.10e-06 | 3.58e-06 |
> | **Liar's Dice (6)** | 6.17e-05  | 8.49e-06               | 2.47e-06 | 4.80e-05 | 2.56e-06 |
> | **Goofspiel (4)**   | 1.05e-08  | 6.77e-17               | 9.57e-06 | 2.88e-08 | 5.31e-07 |
> | **Goofspiel (5)**   | 3.38e-05  | 5.12e-16               | 4.43e-05 | 1.23e-06 | 3.70e-05 |
> | **Goofspiel (6)**   | 2.73e-05  | 1.55e-07               | 4.50e-05 | 6.23e-06 | 3.62e-05 |
>
>
>
>
>
> **W4: The novelty of the proposed method requires clearer articulation, particularly in contrast to Cai et al. (2025), which also established parameter-free last-iterate convergence for RM+ algorithms in games.**
>
> **A:**
>
> - **Problem Domain: Our work focuses on EFGs, while Cai et al. focus on normal-form games.** This is a critical difference, as the analysis for EFGs is substantially more complex due to the necessity of handling counterfactual values.  For instance, Lemma 4.3, 4.4, B.2, and B.3 in our paper pertain to counterfactual values, a theoretical aspect entirely absent in Cai et al. (2025). And these results cannot be obtained from Cai et al. (2025) directly as counterfactual values significantly increases the complexity of the analysis. In fact, as stated in Appendix A Related Work, their proof techniques related to the parameter-free last-iterate convergence for RM+ primarily follow our proof techniques
> - **Different Levels of Parameter-Free Guarantees**: The parameter-free guarantee of RM+ in Cai et al. (2025) holds only when the initial accumulated counterfactual regrets are all set to zero (see their Algorithm 1). **However, they do not investigate the parameter-free guarantee when the initial accumulated counterfactual regrets are nonzero—a case addressed by our work.** This distinction is central to RTCFR+, as noted in lines 78-79: **RTCFR+ cannot converges to an NE of the original EFG without the parameter-free guarantee for arbitrary initial accumulated counterfactual regrets!** Specifically, as indicated in lines 335-337, if the accumulated counterfactual regrets are reset to zero each time a new perturbed regularized EFG is encountered, experimental results (Figure 3) demonstrates that RTCFR+ fails to converge.
>
>
>
> **Q1: Computation via the full matrix suggests per-iteration complexity scales with histories rather than infosets or sparsification[1].**
>
> **A:** While our theoretical analysis uses matrix notation for clarity, our implementation is built upon the highly optimized LiteEFG framework [Liu et al., 2024]. The operations you mentioned are implemented within LiteEFG. Thus, we are not familiar with the specifics. Notably, the second author of LiteEFG (the advisor of LiteEFG's first author) is also the first author of [1]. Excluding the internal details of LiteEFG, our current code (Appendix F) is indeed infoset-linear.
>
>
>
> **Q2: Line 152 assumes NEs satisfy γ-perturbed constraints. How are NEs computed when optimal strategies violate γ-bounds (e.g., pure strategies)?**
>
> **A:**  Lines 152-153 discuss the NE of γ-perturbed EFGs (line 153 specifies “the set of NEs of γ-perturbed EFGs”). The definition of γ-perturbed EFGs ensures that all strategies within these EFGs adhere to γ-perturbed constraints, which naturally includes NE strategies of γ-perturbed EFGs.
>
> In addition, note that, the perturbation is not related to the game we finally want to solve. The perturbation is only used for the internal operation of our algorithm.
>
>
>
> **Q3: How is exploitability calculated under perturbation? Does it measure deviation in the original or perturbed game—particularly critical for Figures 1–3?**
>
> **A:** In this paper, we do not calculate exploitability in perturbed EFGs. The exploitability shown in all the figures within the paper refers solely to that of the original EFGs.
>
> Again, as we stated in Q2, the perturbation is not related to the game we finally want to solve. The perturbations only used for the internal operation of our algorithm.
>
>
>
> **Q4: Algorithm 1 (Line 8) updates reference strategies at fixed intervals. Could pathological strategy sequences at update points prevent convergence? Is there theoretical assurance?**
>
> **A:** While theory suggests updating the reference strategy only after finding an exact NE of the perturbed regularized EFG, updating at fixed intervals is a common and practical approach used in prior work [Pérolat et al., 2021].
>
> Crucially, our algorithm includes a mechanism to ensure convergence: as detailed in lines 311-322, introducing µ←µ×(1−ς) in line 8 of Algorithm 1—gradually reducing the regularization weight—has minimal impact on the empirical convergence rate of RTCFR+ (Figure 2). However, it ensures that for any fixed intervals, the sequence of NEs for the perturbed regularized EFGs converges to the set of NEs of the original EFG. Therefore, Algorithm 1 guarantees the ability to learn the NE of the original EFGs.
>
>
>
> **Q5: Why does Reg-CFR stagnate near 1e-5 exploitability in Kuhn Poker/Goofspiel (Fig 1)?**
>
> **A:** As stated in line 352, we adopt the hyperparameters from the original Reg-CFR paper. The exploitability curve presented in the original paper also exhibits this result (Fig 1, [https://arxiv.org/pdf/2206.09495v1]).
>
>
>
> **Q6: This paper does not provide a theoretical bound for the number of iterations T.**
>
> **A:**  Currently, we cannot provide a finite-time rate. To our knowledge, no existing research on RM-based CFR algorithms has established a finite-time last-iterate convergence rate. This remains an open question, which arises not only from the intrinsic complexity of RM algorithms but also from the intricacies of both EFGs and the counterfactual regret minimization framework.
>
> Second, to the best of our knowledge, even when restricting attention solely to normal-form games—where the complexities of EFGs and the counterfactual regret minimization framework are absent—finite-time last-iterate convergence of RM algorithms has only been achieved by RS-ExRM+ and RS-SPRM+ [Cai et al., 2025]. However, RS-ExRM+ and RS-SPRM+ sacrifices the parameter-free property. In other words, even in normal-form games, no existing work establishes both finite-time last-iterate convergence and a parameter-free nature for RM algorithms, let alone in the extensive-form games we consider.

---

> ### Comment · Reviewer_L64t · 2025-08-01
>
> I think the author should emphasize a core contribution of the work: designing the first last-iterative algorithm in EFG that performs no worse than average-iterative algorithms in experiments. May I ask if this is a core contribution? If so, I think this work is worth accepting.
>
> As the author also acknowledges that RTCFR+ is not a parameter-free algorithm, I believe the title of the article is inappropriate. I think the title should highlight the last-iterate with strong experimental performance. I'm not sure if the title can be modified, but I suggest changing the title of this work.
>
> Therefore, even if we only consider the experimental part and not the theoretical proof part, I believe this article is still worth accepting. Therefore, I have revised my score to 4. I think this article can receive a higher score if the author can do the following in rebuttal phase:
>
> 1. Apply the RTCFR+method to larger scale games and attempt to combine it with techniques such as depth-limited solving [1,2], impact-recall abstraction [3], action abstraction [4], warm-start[5], sparsification [6], pruning [7], etc. If possible, I would like the author to write a separate section discussing the application prospects of RTCFR+in large-scale games and its combination with these common technologies.
>
> 2. Attempt to apply the RTCFR+ method to multiplayer games, as the convergence of the last-iterate method in multiplayer games is worth studying [8].
>
> [1] Brown et al. Depth-Limited Solving for Imperfect-Information Games. NIPS 2018
>
> [2] Brown et al. Combining Deep Reinforcement Learning and Search for Imperfect-Information Games. NIPS 2020
>
> [3] Ganzfried et al. Potential-Aware Imperfect-Recall Abstraction with Earth Mover’s Distance in Imperfect-Information Games. AAAI 2014
>
> [4] Li et al. RL-CFR: Improving Action Abstraction for Imperfect Information Extensive-Form Games with Reinforcement Learning. ICML 2024
>
> [5] Brown el al. Strategy-Based Warm Starting for Regret Minimization in Games. AAAI 2016
>
> [6] Farina et al. Fast Payoff Matrix Sparsification Techniques for Structured Extensive-Form Games. AAAI 2022
>
> [7] Li et al. Efficient Online Pruning and Abstraction for Imperfect Information Extensive-Form Games. ICLR 2025
>
> [8] Lu et al. Divergence-Regularized Discounted Aggregation: Equilibrium Finding in Multiplayer Partially Observable Stochastic Games. ICLR 2025

---

> ### Author Response · Authors · 2025-08-05
>
> We appreciate your feedback and suggestions.
>
>
>
> **Q1: I think the author should emphasize a core contribution of the work: designing the first last-iterative algorithm in EFG that performs no worse than average-iterative algorithms in experiments. May I ask if this is a core contribution? If so, I think this work is worth accepting.**
>
> **A:** Thank you for your suggestion. This is one of the core contribution of our work. As far as we know, our RTCFR+ is the first last-iterate convergence algorithm in EFG that demonstrates performance no worse than average-iterate algorithms in experiments. We will emphasize this point further in the final version.
>
>
>
> **Q2: As the author also acknowledges that RTCFR+ is not a parameter-free algorithm, I believe the title of the article is inappropriate. I think the title should highlight the last-iterate with strong experimental performance. I'm not sure if the title can be modified, but I suggest changing the title of this work.**
>
> **A:** We acknowledge that the current title is not quite appropriate. In the final version, we will revise the title to "Efficient Last-Iterate Convergence in Solving Extensive-Form Games."
>
>
>
> **Q3.1: Apply the RTCFR+method to larger-scale games and attempt to combine it with techniques such as depth-limited solving [1,2], impact-recall abstraction [3], action abstraction [4], warm-start[5], sparsification [6], pruning [7], etc.**
>
> We attempt to utilize RTCFR+ to address larger-scale games, such as heads-up no-limit Texas Hold’em (HUNL) Subgames, specifically those open-sourced by Libratus [Brown and Sandholm, 2018]. For the aforementioned methods, we note that there is no available open-source code. As a result, we opted for another method with an existing open-source implementation suitable for larger-scale games—Vector CFR [9]: "on the way from the root to the leaves, we will pass forwards two vectors: one containing the probabilities of us and one containing the probabilities of the opponent reaching the current game state, for each player’s $n$ possible private chance outcomes." Theoretically, Vector CFR achieves an $n$-fold speedup, where $n$ denotes the number of chance outcomes (1326 in HUNL as chance outcomes is the possible private hands in HUNL).
>
> We employ the open-source code from Poker RL [Eric Steinberger, 2019; Xu et al., 2024b], which supports Vector CFR and Subgames from Libratus, specifically Subgame 3 and Subgame 4. These Subgames have Histories numbering $4.0 \times 10^8$ and $2.4 \times 10^8$, respectively. These are $400$ and $240$ times greater than the largest game tested in this paper (Goofspiel (6), as shown in Table 1 of our paper). The experimental results after 5,000 iterations are (we present the last-iterate convergence performance for RTCFR+):
>
> |             Algorithms              | Subgame3 | Subgame4 |
> | :---------------------------------: | :------: | :------: |
> |               RTCFR+                | 0.00107  | 0.00136  |
> |   CFR+ (last-iterate performance)   | 0.00116  | 0.00281  |
> |  PCFR+ (last-iterate performance)   | 0.00170  | 0.00258  |
> | CFR+ (average-iterate performance)  | 0.00125  | 0.00101  |
> | PCFR+ (average-iterate performance) | 0.00144  | 0.00104  |
>
> RTCFR+ surpasses the last-iterate convergence performance of CFR+ and PCFR+ in all two HUNL Subgames. Furthermore, in Subgame3, RTCFR+ also outperforms the average-iterate convergence performance of CFR+ and PCFR+. Notably, CFR+ and PCFR+ do not exhibit last-iterate convergence guarantee.
>
> In fact, without the acceleration provided by Vector CFR, it would be infeasible to conduct experiments on HUNL Subgames. For instance, in our current work, the largest game evaluated—Goofspiel (6)—requires approximately **76 minutes** to complete 5,000 iterations of RTCFR+. In contrast, HUNL Subgames are several **hundred times larger** than this benchmark; yet, with Vector CFR, we are able to finish 5,000 iterations of RTCFR+ within just **21 minutes**.
>
> Additional References:
>
> [9] Johanson, Michael, Nolan Bard, Marc Lanctot, Richard G. Gibson, and Michael Bowling. "Efficient Nash equilibrium approximation through Monte Carlo counterfactual regret minimization." In *Aamas*, pp. 837-846. 2012.

---

> ### Author Response · Authors · 2025-08-05
>
> **Q3.2: Discuss the application prospects of RTCFR+ in large-scale games and its combination with these common technologies**
>
> Firstly, RTCFR+ can be directly applied to large-scale games without any modifications. In fact, the modifications introduced by RTCFR+ over CFR+ are minimal. As demonstrated in our implementation provided in Appendix F, RTCFR+ requires fewer than 30 additional lines compared to CFR+ (specifically, lines 33, 40–41, 47–49, 51-55, and 62–66 of the RTCFR+ implementation in Appendix F). The main limitation of applying RTCFR+ to large-scale games lies in the need to tune the hyperparameters $\mu$, $\gamma$, and $T_u$, which can vary significantly across different games. Addressing the dependency on tuning $\mu$, $\gamma$, and $T_u$ remains a central direction for future work. It is important to clarify, however, that this requirement originates from the RT framework itself; all existing algorithms based on the RT framework require tuning of these parameters.
>
> Secondly, integrating RTCFR+ with the mentioned approaches requires case-by-case analysis.
>
> - For algorithms that solely modify the game tree, such as depth-limited solving [1,2], impact-recall abstraction [3], action abstraction [4], and Vector CFR [9], RTCFR+ can be directly applied since RTCFR+ only requires execution on the new game tree. This process is straightforward and presents no significant challenges.
> - Regarding the warm-start approach [5], while its concept of setting initial accumulated counterfactual regrets using an efficient initial strategy is insightful, current integration with RTCFR+ is not feasible. Specifically, [5] is an enhancement tailored for the original CFR. Formally, the analysis presented on the bottom left of page four in [5] demonstrates that the substitute regret is given by $R^{\prime T}(I,a) = T(v^{\prime \sigma}(I,a) - v^{\prime \sigma}(I))$. This formulation implies that $R^{\prime T}(I,a)$ can be negative, a property that does not hold in CFR$^+$ and RTCFR$^+$.
> - As for sparsification [6], which optimizes the computation of loss gradients ($\ell^t_i$, the last line of Eq. (3)), RTCFR+ can seamlessly integrate. This compatibility arises because RTCFR+ solely requires the input of loss gradients, which then facilitates strategy updates through the update rules defined in the first four lines of Eq. (3).
> - The pruning approach in [7] can be directly integrated with RTCFR+. Since this pruning approach modifies the game tree before the algorithm execution (e.g., "permanently and correctly eliminating sub-optimal branches before the CFR begins"), it aligns with our earlier statement on game-tree modification approaches. Hence, RTCFR+ can be directly applied.
>
> As previously discussed in **Q3.1**, since none of the approaches in [1–8] provide open-source code, we adopt Vector CFR [9] to extend our RTCFR+ method for solving larger-scale games as Poker RL provide the open-source code of Vector CFR. In addition, as mentioned earlier, Vector CFR offers significant acceleration, which is crucial for scaling RTCFR+ to larger-scale games.
>
> We also attempt to implement the algorithms you mentioned, but we cannot make it successful.

---

> > ### Author Response · Authors · 2025-08-05
> >
> > **Q4: Attempt to apply the RTCFR+ method to multiplayer games, as the convergence of the last-iterate method in multiplayer games is worth studying [8].**
> >
> > **A:** We agree that last-iterate convergence in multiplayer games presents a compelling research direction.
> >
> > From a theoretical perspective, our results (Theorem 4.1) can be extended to multiplayer games that exhibit monotonicity—an assumption satisfied, for example, by two-player zero-sum games [Pérolat et al., 2021]. In particular, all the lemmas utilized in the proof of Theorem 4.1—namely, Lemmas 4.2, 4.3, 4.4, B.2, B.3, and B.4—hold for arbitrary games, while Lemma B.1 is valid only for games with monotonicity. Notably, most of multiplayer games do not satisfy the monotonicity. Thereofere, we observe that recent works have begun to investigate last-iterate convergence for multiplayer games under properties weaker than monotonicity, such as hypomonotonicity [8] or the weak MVI [10]. Exploring these extensions constitutes one of our intended future research directions.
> >
> > Additionally, we conduct a straightforward experiment to validate the effectiveness of our RTCFR+ in multiplayer games. Specifically, we test RTCFR+, PCFR+, and CFR+ on 3-player Leduc Poker, as implemented in OpenSpiel by using the following string. The experimental results indicate that RTCFR+, PCFR+, and CFR+ all exhibit poor last-iterate convergence performance. Specifically, for these algorithms, the exploitability decreases from 8.59 to only approximately 4.49, after which it plateaus and fails to decrease further.
> >
> > ```
> > leduc_poker_string_3_player = (
> >         "universal_poker(betting=limit,numPlayers=3,numRounds=2,numSuits=2,numRanks=3,"
> >         "blind=1 1 1,firstPlayer=1 1,maxRaises=2 2,raiseSize=2 4,"
> >         "numHoleCards=1,numBoardCards=0 1,stack=20 20)"
> >     )
> > ```
> >
> > Building on our RTCFR+, we will explore algorithms in multiplayer games that not only provide a theoretical last-iterate convergence guarantee but also demonstrate a faster empirical last-iterate convergence rate.
> >
> > -----------------------------------------------------------------
> >
> > Additional References:
> >
> > [10] Pethick, Thomas, Ioannis Mavrothalassitis, and Volkan Cevher. "Efficient Interpolation between Extragradient and Proximal Methods for Weak MVIs." *The Thirteenth International Conference on Learning Representations*, 2025.

---

> ### Comment · Reviewer_L64t · 2025-08-05
>
> Thanks to the author for the detailed explanation and supplementary experiments. The HUNL subgame experiment is excellent. The convergence curve can be displayed in the revision, and the DCFR algorithm can be added. Vector CFR utilises the properties of HUNL-type games, where players' hands do not change, so each information set can be parallelised or vectorised in each iteration. I would like to know whether the HUNL experiment in the author's implementation ran for 500 iterations or 5000 iterations? From the results, it seems to be 500 iterations.
>
> Thank you to the author for the detailed discussion of various optimisation methods in IIG and the exploration of multi-player games. And I have another question: since RTCFR+ solves problems under a perturbed EFG, does this mean that RTCFR+ can handle situations where opponents choose off-tree actions or low-probability actions better than other CFR algorithms? Traditional CFR algorithms do not perform well when handling opponents' low-probability actions[11].
>
> [11] Gabriele Farina, Christian Kroer, Tuomas Sandholm. Regret minimization in behaviorally-constrained zero-sum games. ICML 2017

---

> > ### Author Response · Authors · 2025-08-06
> >
> > Thanks for taking the time to review our response and thoughtful review!
> >
> > **Q1: The convergence curve can be displayed in the revision, and the DCFR algorithm can be added.**
> >
> > **A:** Thanks for your suggestions. We will include the convergence curve and the results of DCFR in the final version.
> >
> >
> >
> > **Q2: I would like to know whether the HUNL experiment in the author's implementation ran for 500 iterations or 5000 iterations? From the results, it seems to be 500 iterations.**
> >
> > **A:** We apologize for the typo in our previous comment—our initial statement mistakenly referred to 500 iterations, when in fact we ran a total of 5,000 iterations. The original comment has now been corrected.
> >
> >
> >
> > **Q3: Since RTCFR+ solves problems under a perturbed EFG, does this mean that RTCFR+ can handle situations where opponents choose off-tree actions or low-probability actions better than other CFR algorithms? Traditional CFR algorithms do not perform well when handling opponents' low-probability actions[11].**
> >
> > **A:** Consistent with [11], if we do not set $\gamma$ (denoted as $\xi$ in [11]) to approach $0$, RTCFR+ can demonstrate superior performance over other CFR algorithms in scenarios where opponents take off-tree or low-probability actions. This advantage arises because the perturbation operation in the RT framework is the "behaviorally-constrained" operator described in [11].

---

> > > ### Comment · Reviewer_L64t · 2025-08-06
> > >
> > > The author's expanded experiments and discussion of related work are excellent. I hope the author will incorporate the rebuttal content into the revision. I have raised my score.

---

> > > > ### Author Response · Authors · 2025-08-06
> > > >
> > > > Thank you for acknowledging our response! We will incude the expanded experiments and discussion of related work in the final version.

---

### Official Review · Reviewer_mv8m · 2025-06-23

**Clarity:** 3
**Significance:** 2
**Originality:** 2
**Rating:** 4
**Confidence:** 4

**Summary:**

The authors introduce methods of applying the *reward transformation* technique of Perolat et al. (2021-22) and Liu et al. (2023) to regret matching in two-player zero-sum extensive-form games, arriving at RTCFR+, an algorithm with guaranteed last-iterate convergence and fast practical performance.

**Questions:**

Some of these questions were alluded to above. I also welcome any other comment regarding the points raised in the main body of the review.

1. Could you compare your Theorem 4.1 to previously-known results (such as Lemma 1 of [1] above)? In particular, why does your proof require anything at all specific about extensive-form games or CFR?
2. Could you provide a rate of last-iterate convergence for Theorem 4.1?

**Ethical Concerns:**

["NO or VERY MINOR ethics concerns only"]

**Final Justification:**

As I stated in the below discussion, I think the changes proposed by the reviewers (including myself) require another round of review, and therefore I will maintain my score. That said, I do not mind if I am outvoted (as it looks like it might be the case.)

----

Edit 8/4: On second thought, the paper is probably interesting enough to warrant an increase in score despite these issues. Increased to 4.

**Limitations:**

yes

**Quality:**

2

**Strengths And Weaknesses:**

The paper claims that RTCFR+ is parameter-free, which to me means that it, like CFR+, works "out-of-the-box" and does not require the user to set any hyperparameters. This is evidently false: in Algorithm 1, I see at least three "interesting'' hyperparameters: $T_u$, $\mu$, $\gamma$. Moreover, these hyperparameters may significantly affect both the theoretical (at least up to constant factors) and practical performance of the algorithm. Thus, I do not think it is valid to call RTCFR+ "parameter-free".

It is well known that algorithms that do exhibit last-iterate convergence, such as OGD, do not perform well in practice for extensive-form games. Moreover, as admitted by the authors on p.25, RTCFR+ does not consistently outperform the prior state of the art, PCFR+. In my opinion, this means that the practical contribution of the paper is not very strong. Along a similar line, it seems weird not to include the PCFR+ results in the main body, given that PCFR+ (or slight variations thereof) is the practical state of the art right now for extensive-form games.

The technical ideas of the paper mostly follow immediately from known previous ideas, namely, regret matching and reward transformation. I am also confused why the proof of Theorem 4.1 is so long. Doesn't that result follow immediately from the fact that CFR+ is no-regret?  See e.g. Lemma 1 of [1]---unless I'm missing something, Theorem 4.1 is a special case of it?

---

> ### Author Rebuttal · Authors · 2025-07-31
>
> We appreciate the time you dedicated to our manuscript.
>
>
>
> **W1: The paper claims that RTCFR+ is parameter-free. This is evidently false.**
>
> **A:** We apologize for the confusion. In this paper, all references to "parameter-free" pertain exclusively to CFR+ for solving perturbed regularized EFGs. At no point do we assert that RTCFR+ is parameter-free. For instance, at the end of Section 3 Problem Statement, we explicitly state, "Our objective is to establish the parameter-free last-iterate convergence for CFR+ in solving Eq. (2)." Moreover, in the limitations section at the end of the paper, we indicate that RTCFR+ requires parameter tuning.
>
> To eliminate any ambiguity, we will add a prominent statement in the introduction and reinforce it in the Limitations section of the final version to state clearly that RTCFR+ is a non-parameter-free algorithm. This will significantly enhances the clarity of our paper.
>
>
>
> **W2: RTCFR+ does not consistently outperform the prior state of the art, PCFR+. In my opinion, this means that the practical contribution of the paper is not very strong.**
>
> **A:**
>
> - PCFR+ lacks the theoretical last-iterate convergence guarantee, whereas our RTCFR+ does offer this guarantee. As we stated in Section 1 Introduction, average-iterate convergence not only increases  computational and memory overhead, but also results approximation errors when strategies are parameterized via function approximation. Therefore, our RTCFR+ shows a promising and practical alternative to the average-iterate convergence algorithms (e.g., PCFR+) for solving large-scale games.
>
> - Compared to existing algorithms that exhibit theoretical last-iterate convergence guarantees—such as Reg-CFR, R-NaD, OMWU, and OGDA—our RTCFR+ demonstrates the fastest empirical last-iterate convergence rate. This makes RTCFR+ more suitable for solving practical applications than these algorithms.
>
>
>
> **W3: The technical ideas of the paper primarily build on established concepts in regret matching and reward transformation.**
>
> **A:** In this paper, our focus is not on combining regret matching and reward transformation, but rather on, after integrating these two methods, whether RM-based CFR algorithms (e.g., CFR+) achieve parameter-free last-iterate convergence when solving perturbed regularized EFGs within the reward transformation framework. Specifically, as stated in lines 69–69, the central question is: “Does CFR+ have parameter-free last-iterate convergence when learning an NE of perturbed regularized EFGs?”
>
> Our primary technical contributions are proving that "CFR+ has parameter-free last-iterate convergence in learning an NE of perturbed regularized EFGs." Specifically, as stated in our introduction: "we propose novel techniques to overcome the challenges in the above two steps of the proof" (line 83-84, beginning of the penultimate paragraph of the introduction). Specifically, our novel contributions are:
>
> - (i) We exploit the fact that an NE represents a best response to others at each infoset in perturbed EFGs to enable the smoothness of the instantaneous counterfactual regrets to be leveraged.
> - (ii) We leverage the linearity of the projection to extend our non-parameter-free convergence result to be parameter-free regime.
>
>
>
> **Q1: I am also confused why the proof of Theorem 4.1 is so long. Doesn't that result follow immediately from the fact that CFR+ is no-regret? See e.g. Lemma 1 of [1]. Could you compare your Theorem 4.1 to previously-known results (such as Lemma 1 of [1] above)?**
>
> **A:** You referenced [1], which appears to be missing from the review. Could you please provide it? We would be pleased to offer a detailed response.
>
> Additionally, it is important to emphasize that, to the best of our knowledge, prior to the NeurIPS 2025 deadline, no study has directly established last-iterate convergence solely from the "no-regret" property. All existing works rely on extensive proofs to demonstrate the last-iterate convergence of no-regret algorithms.
>
>
>
> **Q2: Could you provide a rate of last-iterate convergence for Theorem 4.1?**
>
> **A:** Currently, we cannot provide a finite-time rate. To our knowledge, no existing research on RM-based CFR algorithms has established a finite-time last-iterate convergence rate. This remains an open question, which arises not only from the intrinsic complexity of RM algorithms but also from the intricacies of both EFGs and the counterfactual regret minimization framework.
>
> Second, to the best of our knowledge, even when restricting attention solely to normal-form games—where the complexities of EFGs and the counterfactual regret minimization framework are absent—finite-time last-iterate convergence of RM algorithms has only been achieved by RS-ExRM+ and RS-SPRM+ [Cai et al., 2025]. However, RS-ExRM+ and RS-SPRM+ sacrifices the parameter-free property. In other words, even in normal-form games, no existing work establishes both finite-time last-iterate convergence and a parameter-free nature for RM algorithms, let alone in the extensive-form games we consider.

---

> > ### Comment · Reviewer_mv8m · 2025-08-01
> >
> > On parameter-freeness: I agree with L64t's suggestion to change the title ought to be changed in light of this.
> >
> > -----
> >
> > Regarding [1]: oops, I completely forgot to include it, and actually upon a second look, I seem to have had the wrong reference in mind anyway. A correct reference is Proposition 3.6 of [2]. Basically the argument is the following: Let $R_i$ be the regret of player $i$ for deviating to their average strategy $\bar{\boldsymbol x}\_i := \frac{1}{T} \sum_{t=1}^T \hat{\boldsymbol x}^t_i$. On one hand we have $R_0 + R_1 \lesssim \sqrt{T}$ (where $\lesssim$ hides polynomial game-dependent constants.) because these are no-regret algorithms. On the other hand, by strong convexity of the regularizers, we have $R_0 + R_1 \gtrsim \mu \sum_{t=1}^T \left[ \lVert \bar{\boldsymbol x}\_0 - \hat{\boldsymbol x}^t_0 \rVert_2^2 + \lVert \bar{\boldsymbol x}\_1 - \hat{\boldsymbol x}^t_1 \rVert_2^2 \right]$. Thus there must exist a time $t$ for which $\lVert \bar{\boldsymbol x}\_0 - \hat{\boldsymbol x}^t_0 \rVert_2^2 + \lVert \bar{\boldsymbol x}\_1 - \hat{\boldsymbol x}^t_1 \rVert_2^2 \lesssim 1/(\mu \sqrt{T})$, which implies best-iterate convergence and indeed also gives a rate of best-iterate convergence. Note that this argument doesn't require any property whatsoever about the regret minimizers, except that they have $O_T(\sqrt{T})$ regret.
> >
> > *On this note, actually, I think you also only get best- (not last-)iterate convergence, right? This distinction is minor but worth pointing out.
> >
> > [2] Ioannis Anagnostides, Ioannis Panageas, Gabriele Farina, Tuomas Sandholm (NeurIPS 2023), "On the Convergence of No-Regret Learning Dynamics in Time-Varying Games"

---

> > > ### Author Response · Authors · 2025-08-02
> > >
> > > Thank you very much for continuing the discussion.
> > >
> > >
> > >
> > > **Q1: I agree with L64t's suggestion to change the title ought to be changed in light of this.**
> > >
> > > **A:** Thank you for your suggestion. In the final version, we will change the title to "Efficient Last-Iterate Convergence in Solving Extensive-Form Games."
> > >
> > >
> > >
> > > **Q2: On this note, actually, I think you also only get best- (not last-)iterate convergence, right? This distinction is minor but worth pointing out.**
> > >
> > > **A:** We would like to clarify that our Theorem 4.1 is the last-iterate convegence of CFR+ in learning an NE of the perturbed regularized EFGs.
> > >
> > > If we solely rely on the no-regret property of CFR+, which presents an $O(\sqrt{T})$ regret bound, we indeed achieve only best-iterate convergence, akin to Proposition 3.6 of [2] (also Lemma 1 of [3]). Specifically, the $O(\sqrt{T})$ regret bound allows us to establish that:
> > >
> > > $$\sum_{t=1}^T\Vert x^* - x^t \Vert^2_2 \leq C_1\sqrt{T}, \forall T \geq 1,$$
> > >
> > > where $x^*$ is an NE of the (perturbed) regularized game. This implies an $O(1/\sqrt{T})$ best-iterate convergence rate.
> > >
> > > However, our Theorem 4.1 does not employ the no-regret property of CFR+. Notably, in the intermediate steps of Theorem 4.1 (line 290), we derive:
> > >
> > > $$\sum_{t=1}^T\Vert x^\* - x^t \Vert^2_2 \leq C_2, \forall T \geq 1,$$
> > >
> > > where $x^\*$ is an NE of the (perturbed) regularized game. This implies the last-iterate convergence, meaning $x^t \to x^\*$ as $t \to \infty$. To elaborate, if $x^t$ did not converge to $x^\*$ as $t \to \infty$, there would necessarily exist a $T$ such that $\sum_{t=1}^T \Vert x^* - x^t \Vert^2_2 > C_2$. Thus, Theorem 4.1 demonstrates the last-iterate convergence.
> > >
> > >
> > >
> > > Additional References:
> > >
> > > [3] Wang, Zifan, Yi Shen, Michael Zavlanos, and Karl Henrik Johansson. "No-Regret Learning in Strongly Monotone Games Converges to a Nash Equilibrium." (2023).

---

> > > > ### Comment · Reviewer_mv8m · 2025-08-02
> > > >
> > > > Thank you. This discussion has clarified some of my confusion. I would recommend that the authors explicitly discuss the results of [2, 3] and compare the results of the present paper to those. In particular, I would mention 1) the fact that the present paper gets a better best-iterate convergence guarantee (sum of squared distances is $O_T(1)$ instead of $O_T(\sqrt{T})$), compared to [2, 3]; and 2) the fact that it gets last-iterate convergence (albeit without rate). I'd also explicitly write out the game-dependent factor hidden by the $O_T(1)$ for your algorithm.
> > > >
> > > > Since the number of writing-related changes is growing (including those recommended by other reviewers), I think the paper probably requires another round of review, and as such I'll keep my score. But I am optimistic that a revised version of the paper can be much stronger and worthy of acceptance.

---

> > > > > ### Author Response · Authors · 2025-08-03
> > > > >
> > > > > We appreciate your feedback and hope that, upon reviewing our response, you will reconsider your evaluation of our submission.
> > > > >
> > > > >
> > > > >
> > > > > **Q1:  I would recommend that the authors explicitly discuss the results of [2, 3] and compare the results of the present paper to those. In particular, I would mention 1) the fact that the present paper gets a better best-iterate convergence guarantee (sum of squared distances is instead of ), compared to [2, 3]; and 2) the fact that it gets last-iterate convergence (albeit without rate). I'd also explicitly write out the game-dependent factor hidden by the for your algorithm.**
> > > > >
> > > > > - Regarding the best-iterate convergence, our primary focus is on last-iterate convergence algorithms and do not consider best-iterate convergence in this paper. As stated by Reviewer L64t, our core contribution is presenting the first last-iterate convergence algorithm for EFG, which empirically performs no worse than average-iterate convergence algorithms.
> > > > >
> > > > > - Regarding the last-iterate convergence, Theorem 4.1 articulates this phenomenon. Specifically, it demonstrates that the strategy profile $\hat{{x}}^{t}$ converges to the set of NEs of the perturbed regularized EFGs.
> > > > >
> > > > > - Regarding the game-dependent factor you mentioned, the process is straightforward. We can incorporate $\eta \leq \frac{\mu}{2 |\mathcal{I}|A_{\max}^2 \left( 6 ( L + \mu )^2 + 8 (P+2\mu D)^2{(A_{\max}C_{\max}+1)^2}/{\gamma^{2H}} \right)}$ from lines 282-286, $0 \leq \zeta_I \leq 1$ from line 591, $\hat{m}^{*,\mu,\gamma,{r}}_i(I) = \hat{v}^{\*,\mu,\gamma,{r}}_i(I) - \langle \hat{v}^{\*,\mu,\gamma,{r}}_i(I), \sigma^{\*,\mu,\gamma,{r}}_i(I)\rangle {1}$ from lines 269-270, and $\Vert \hat{v}^{\sigma}_i(I)\Vert_2 \leq P + 2\mu D$ from lemma B2, into the equations following line 286 to achieve our results:
> > > > >
> > > > >   $$\sum_{t=1}^T\Vert x^* - x^t \Vert^2_2 \leq \frac{4|\mathcal{I}|^2A_{max}^2 \left( 6 ( L + \mu )^2 + 8 (P+2\mu D)^2{(A_{max}C_{max}+1)^2}/{\gamma^{2H}} \right)}{\mu^2}(2A_{max}^2+2\Vert \max_{I \in \mathcal{I}}\theta^1_I \Vert^2_2+(A_{max} +1)(P+2\mu D)\Vert \max_{I \in \mathcal{I}}\theta^1_I \Vert_2), \forall T \geq 1$$
> > > > >
> > > > >
> > > > >
> > > > > **Q2:  the number of writing-related changes is growing (including those recommended by other reviewers).**
> > > > >
> > > > > **A:** We would like to clarify that the modifications we're considering, specifically regarding changes from negative scores (i.e., 3) to positive scores (i.e., 4) by other reviewers are minimal:
> > > > >
> > > > > 1. Title modification.
> > > > > 2. Explicitly stating that RTCFR+ is not parameter-free.
> > > > > 3. Adding a comparison to the average-iterate convergence performance of the classical CFR algorithm, emphasizing our main contribution: the first last-iterate convergence algorithm in EFG, which performs no worse than average-iterate convergence algorithms in experiments.
> > > > >
> > > > > The first and second points require very minimal changes, likely less than five lines.
> > > > >
> > > > > As for other suggested modifications by Reviewer L64t, these mainly pertain to potential improvements in our paper, rather than affecting the paper's acceptance at NeurIPS.

---

> > > > > > ### Comment · Reviewer_mv8m · 2025-08-03
> > > > > >
> > > > > > I think the parameter-freeness issue requires a heavier revision than the authors suggest. I'm not sure how interesting the result is that CFR+ is a "parameter-free algorithm" in the perturbed regularized EFG setting. After all, one never actually *directly* cares about the perturbed regularized game; one cares about the EFG, and the perturbed regularized game is really only a tool to accelerate convergence to an equilibrium of the underlying EFG. So whether the algorithm for solving the perturbed regularized game is parameter-free is not really that relevant, and certainly, in my opinion, is not worth mentioning as the "key question" of the paper.
> > > > > >
> > > > > > Ultimately, I'd suggest that basically all references to "parameter-free"ness be removed from the paper. All the reviewers (including myself) are basically quibbling over whether the use of this term is valid, and I agree with L64t that this quibbling overshadows the more interesting contribution of the paper, which is an algorithm that performs similarly to PCFR+ in practice while having a best/last-iterate convergence guarantee--to my knowledge, the first demonstration of such an algorithm. Removing the parameter-free claim would allow a better focus on these interesting contributions. This is a major framing and writing change in the abstract/intro that is not done in a matter of a few lines.
> > > > > >
> > > > > > A few more minor points:
> > > > > >
> > > > > > * I do think best-iterate convergence deserves some space. Why not? That is *actually* a small change requiring only a few lines of discussion, and you get to state an additional result with an actual rate.
> > > > > >
> > > > > > * The expression you gave for the best-iterate bound is quite a mess. Might be better to also state a cleaner version, say, of the form $|I|^a |A_\max|^b / \gamma^{dH} \mu^e$ when $\boldsymbol \theta := 0$ (remove references to D, P, ..., since these can be bounded by some polynomial in $|\Sigma|$).
> > > > > >
> > > > > > * Related to a note that I mentioned in my original review: I would strongly recommend replacing CFR+ in Figure 1 with PCFR+. I don't see any valid reason not to do this, since PCFR+ is better in practice and has basically the same theoretical guarantees (average- but not known best- or last-iterate convergence) as CFR+.

---

> > > > > > > ### Author Response · Authors · 2025-08-04
> > > > > > >
> > > > > > > **Q1: I think the parameter-freeness issue requires a heavier revision than the authors suggest. Removing the parameter-free claim would allow a better focus on these interesting contributions. This is a major framing and writing change in the abstract/intro that is not done in a matter of a few lines.**
> > > > > > >
> > > > > > > **A:** Thank you for your response. We want to emphasize the critical importance of the parameter-free concept. It is essential to note that modifying the title is inevitable; however, eliminating the concept of "parameter-free" entirely is not advisable as the parameter-free result for CFR+ significantly contributes to the excellent performance of RTCFR+.
> > > > > > >
> > > > > > > - Firstly, as mentioned in our abstract (lines 10-12) and introduction (lines 60-62), the RT framework involves solving multiple perturbed regularized EFGs, which makes parameter tuning for each individual perturbed regularized EFG impractical. Hence, for our final algorithm, RTCFR+, to perform effectively, the algorithm used for solving the perturbed regularized EFG must be parameter-free.
> > > > > > > - Secondly, without our parameter-free result for CFR+, RTCFR+ would not achieve its current level of effectiveness and would never converge (lines 78-79, 327-337). Specifically, our parameter-free result applies for any initial accumulated counterfactual regret $\theta^1_I$. This is crucial. If we achieved only the degree of parameter-free status as in Farina et al. [2021]—which holds for $\theta^1_I = 0$—it would require resetting $\theta^1_I$ to 0 for each perturbed regularized EFG. This reset would prevent RTCFR+ from converging, as illustrated in Figure 3. Therefore, we must employ our parameter-free result for CFR+, setting $\theta^1_I$ of the new perturbed regularized EFG to $\theta^{T_u}_I$ of the previous one.
> > > > > > >
> > > > > > > Therefore, for the point that RTCFR+ performs similarly to PCFR+ in practice while offering a last-iterate convergence guarantee, this is the result of our parameter-free result for CFR+. To emphasize this point, it is sufficient to highlight the concept of that RTCFR+ performs similarly to PCFR+ in practice while possessing a last-iterate convergence guarantee at the end of the abstract and the introduction. Specifically, this can added following line 21 and line 107.
> > > > > > >
> > > > > > >
> > > > > > >
> > > > > > > **Q2: I do think best-iterate convergence deserves some space.**
> > > > > > >
> > > > > > > **A:** We do not consider best-iterate convergence because it offers limited utility in real-world games. With best-iterate convergence, computing the exploitability of each iteration's strategy profile is necessary to select an optimal strategy, but this task is typically challenging due to the vast size of real-world games, such as heads-up no-limit Texas Hold’em (HUNL), which reaches a size of $10^{170}$. In contrast, last-iterate convergence circumvents the need to compute exploitability for every iteration; it simply requires the selection of the strategy from the final iteration.
> > > > > > >
> > > > > > > In the final version, we will briefly discuss the distinction between best-iterate convergence and last-iterate convergence, citing the references you mentioned, [2] and [3].
> > > > > > >
> > > > > > >
> > > > > > >
> > > > > > > **Q3: The expression you gave for the best-iterate bound is quite a mess. Might be better to also state a cleaner version, say $\frac{|\mathcal{I}|^a A_{max}^b}{\gamma^{dH}u^e}$ when $\theta=0$, of the form when (remove references to $D$, $P$).**
> > > > > > >
> > > > > > > **A:** As you mentioned, the condition $\theta=0$ is considered. However, as we outlined in our responses to your **Q1**, our parameter-free result holds for any initial accumulated counterfactual regret $\theta^1_I$, not just when it equals $0$. This distinction is critical because RTCFR+ would not converge when we achieved only the degree of parameter-free status that $\theta^1_I=0$.
> > > > > > >
> > > > > > >
> > > > > > >
> > > > > > > **Q4:  I would strongly recommend replacing CFR+ in Figure 1 with PCFR+.**
> > > > > > >
> > > > > > > **A:** Thank you for your suggestion; we will incorporate it. This modification is minor.

---

### Official Review · Reviewer_NuVT · 2025-07-02

**Clarity:** 1
**Significance:** 3
**Originality:** 3
**Rating:** 4
**Confidence:** 2

**Summary:**

The paper addresses the open question of obtaining *last-iterate* convergence guarantees for Counterfactual Regret Minimization (CFR) algorithms without the need to tune a learning-rate schedule.
Prior work has either provided last-iterate convergence by switching to slower Online Mirror Descent–type updates or retained the fast, regret-matching (RM) updates at the cost of only proving convergence of the *average* strategy.
The authors prove that the widely-used CFR\\(^{+} \\) algorithm enjoys *parameter-free* last-iterate convergence when solving perturbed and regularised two-player zero-sum extensive-form games (EFGs).

The technical contribution has two main steps.
First, it is shown that CFR\\(^{+} \\) converges in the last iterate whenever the internal step size exceeds a fixed constant.
The proof leverages a novel term involving counterfactual regrets and best-response utilities to control the dynamics even though RM updates evolve on a cone rather than a simplex.
Second, a coupling argument exploits the linearity of CFR\\(^{+} \\) projections to translate any arbitrary (possibly small) step size to one in the proven range while leaving the sequence of strategies unchanged, thereby delivering a truly parameter-free guarantee.
Combining this insight with the Reward-Transformation framework yields the proposed algorithm, RTCFR\\(^{+} \\), whose perturbation parameter is halved periodically while the reference strategy is updated.

Empirical results on nine benchmark games (including Kuhn, Leduc, Goofspiel, and Battleship) demonstrate that RTCFR\\(^{+} \\) converges substantially faster, in exploitability, than earlier last-iterate methods such as Reg-CFR, R-NaD, Online Mirror Descent with Weighted Updates (OMWU), and Optimistic Gradient Descent Ascent (OGDA).
In many cases it even outperforms the classical, average-iterate CFR\\(^{+} \\), all while keeping default hyper-parameters fixed across games.
These findings suggest that practitioners can now obtain both rapid convergence and last-iterate guarantees without step-size tuning.

**Questions:**

- The density of notation and the clarity of the problem setup may pose challenges for readers.
  In particular, for readers unfamiliar with extensive-form games (EFGs), the exposition makes it difficult to develop a clear understanding of the problem the paper aims to solve.
  I found it nontrivial to grasp how the objective is captured by the Nash equilibrium formulation in equation (1).
  Furthermore, following the discussion on perturbed extensive-form games, I was also left without a clear picture of the overall problem or the specific objective functions being optimized.
  The paper transitions quickly into discussing regret minimization and defines the players’ regrets, but without a firm grounding in the problem setting, it is difficult to contextualize these definitions.
  I understand that space constraints may be a factor, but from the perspective of a non-expert reader, the problem formulation remains opaque.
  A brief high-level overview early on, possibly accompanied by a simple running toy example, would significantly improve accessibility and comprehension.

- Since the paper's main contribution lies in solving perturbed EFGs, it would be helpful to elaborate on how the perturbation and regularization may affect the game’s payoffs.
  In particular, more practical guidance on selecting the parameters \\( \\gamma \\) and \\( \\mu \\) would be valuable, especially to ensure that the resulting equilibrium remains a reasonable approximation of the unperturbed game's solution.
  This would be beneficial for practitioners aiming to apply the method.

- All experiments are conducted on games of moderate size.
  It would strengthen the empirical results to include an evaluation of the method’s scalability to larger, imperfect-information games.

**Ethical Concerns:**

["NO or VERY MINOR ethics concerns only"]

**Final Justification:**

Dear AC and authors,

As I am not very familiar with the topic, I am happy to defer to the AC’s decision based on the other reviewers’ comments.

Best wishes,
Your reviewer

**Limitations:**

The density of notation and the clarity of the problem setup may pose challenges for readers.
In particular, for readers unfamiliar with extensive-form games (EFGs), the exposition makes it difficult to develop a clear understanding of the problem the paper aims to solve.
I found it nontrivial to grasp how the objective is captured by the Nash equilibrium formulation in equation (1).
Furthermore, following the discussion on perturbed extensive-form games, I was also left without a clear picture of the overall problem or the specific objective functions being optimized.
The paper transitions quickly into discussing regret minimization and defines the players’ regrets, but without a firm grounding in the problem setting, it is difficult to contextualize these definitions.
I understand that space constraints may be a factor, but from the perspective of a non-expert reader, the problem formulation remains opaque.

**Quality:**

2

**Strengths And Weaknesses:**

The addressing question is very important and significant. However, the presentation is quite hard to understand. For details, please see the questions.

---

> ### Author Rebuttal · Authors · 2025-07-31
>
> Thank you for your positive feedback and insightful suggestions for improvement.
>
>
>
> **Q1：It nontrivial to grasp how the objective is captured by the Nash equilibrium formulation in equation (1).**
>
> **A:** According to the Minimax theorem, the solution of equation (1), namely $min_{x_0} max_{x_1} x_0^T A x_1$, is also the solution of $max_{x_1} min_{x_0} x_0^T A x_1$. As we consider zero-sum games, this solution implies that each player aims to minimize the maximum possible gain of the opponent. This stable state, where neither player can unilaterally change their strategy to improve their outcome, is NE since the definition of NE is a rational behavior where no player can benefit by unilaterally deviating from the equilibrium (lines 133-134).
>
>
>
> **Q2:  I was also left without a clear picture of the overall problem or the specific objective functions being optimized.**
>
> **A:**  We apologize the confusion about the overall problem. Our primary objective, as stated at the end of Section 3, is: "Our objective is to establish the parameter-free last-iterate convergence for CFR+ in solving Eq. (2)." In other words, we aim to demonstrate that CFR+ achieves parameter-free last-iterate convergence when learning an NE of perturbed regularized EFGs, as specifically emphasized in the introduction, lines 68-69.
>
>
>
>
>
> **Q3: A simple running toy example, would significantly improve accessibility and comprehension.**
>
> **A:** We will include a toy extensive-form game, which will significantly enhance the clarity of our paper.
>
>
>
> **Q4: It would be helpful to elaborate on how the perturbation and regularization may affect the game’s payoffs.**
>
> **A:** As we stated in lines 215-218, "The introduction of perturbation and regularization ensures the smoothness of counterfactual values and strong monotonicity." More precisely:
>
> - Perturbation does not affect the game's payoffs but instead modifies the strategy space to ensure the smoothness of counterfactual values.
> - In contrast, regularization means an additional regularization term is added to the game's payoffs, thereby imparting strong monotonicity.
>
> This smoothness and strong monotonicity reduce the difficulty of proving last-iterate convergence. Specifically:
>
> - Smoothness of counterfactual values allows us to establish a relationship between the gap in counterfactual values and the gap between strategies (line 126).
> - Strong monotonicity guarantees the uniqueness of solutions. Precisely, strong monotonicity ensures that the NE of the perturbed regularized EFG is unique.
>
> Lastly, it is important to note that increasing either the perturbation or regularization weights causes the NE of the perturbed regularized EFG to diverge further from the NE of the original EFG. Therefore, as you mentioned, the selection of $\gamma$ and $\mu$ is critical. We provide a detailed discussion of this issue in **Q5**.
>
>
>
> **Q5: More practical guidance on selecting the parameters and would be valuable.**
>
> **A:** As outlined in Section E (lines 771-788), we have empirically evaluated various values of the parameters of RTCFR+ ($\mu$ and $T_u$; for $\gamma$, we observe that it only needs to be set to a sufficiently small value) and provided selection guidelines to optimize performance. Here, we provide a brief intuition about this parameter.
>
> - Selecting an extremely small $\mu$ may hinder the CFR's ability to learn NE of the perturbed regularized game, resulting in learning an NE of perturbed regularized EFGs becomes impossible. As a result, attaining an NE of the original game becomes impracticable for any $T_u$.
> - When $\mu$ is optimal, neither too small nor too large, this enables CFR+ to learn sufficiently accurate approximate an NE of perturbed regularized EFGs.  These allow RTCFR+ to achieve commendable performance.
> - For large $\mu$ values, although CFR+ are capable of learning the exact NE of perturbed regularized EFGs, the requisite number of reference strategy updates becomes excessively large. Hence, we observe that with large $\mu$ values, a smaller $T_u$ yields better performance.
>
> Based on these analyses, we advocate for the prioritization of determining  $\mu$’s value, followed by the value of $T_u$ , when practically applying RTCFR+.
>
>
>
> **Q6: It would strengthen the empirical results to include an evaluation of the method’s scalability to larger, imperfect-information games.**
>
> **A:** If sufficient computational resources is available, we will extend larger games in the final version. It is worth noting, however, that the games currently evaluated in our paper are already larger in scale than those in related work focused on the last-iterate convergence of CFR algorithms (e.g., Pérolat et al. (2021) and Liu et al. (2023)), demonstrating a step forward in this specific area of research.

---

> > ### Comment · Reviewer_NuVT · 2025-08-08
> > **Thank you for the explanations**
> >
> > I would like to thank the authors for all the comments. I really appreciate it and therefore I would keep my original score.

---

> > > ### Author Response · Authors · 2025-08-09
> > >
> > > Thank you so much for your appreciation!

---

> ### Author Response · Authors · 2025-08-05
>
> **Q6: It would strengthen the empirical results to include an evaluation of the method’s scalability to larger, imperfect-information games.**
>
> **A:** We have conducted experiments on larger-scale games. We utilize RTCFR+ to address larger-scale games, such as heads-up no-limit Texas Hold’em (HUNL) Subgames, specifically those open-sourced by Libratus [Brown and Sandholm, 2018]. We employ the open-source code from Poker RL [Eric Steinberger, 2019; Xu et al., 2024b], which supports Subgames from Libratus, specifically Subgame 3 and Subgame 4. These Subgames have Histories numbering $4.0 \times 10^8$ and $2.4 \times 10^8$, respectively. These are $400$ and $240$ times greater than the largest game tested in this paper (Goofspiel (6), as shown in Table 1 of our paper). We compare with CFR+ and PCFR+ since Poker RL does not support Reg-CFR, R-NaD, OMWU, and OGDA. The experimental results after 5000 iterations demonstrate that RTCFR+ surpasses the last-iterate convergence performance of CFR+ and PCFR+ in all two HUNL Subgames.
>
> | Algorithms | Subgame3 | Subgame4 |
> | :--------: | :------: | :------: |
> |   RTCFR+   | 0.00107  | 0.00136  |
> |    CFR+    | 0.00116  | 0.00281  |
> |   PCFR+    | 0.00170  | 0.00258  |

---

### Official Review · Reviewer_kpKA · 2025-07-03

**Clarity:** 2
**Significance:** 2
**Originality:** 2
**Rating:** 4
**Confidence:** 4

**Summary:**

The paper considers the problem of establishing last-iterate convergence rates for parameter-free CFR-based algorithms in extensive-form zero-sum games. Its main contribution is a last-iterate convergence guarantee for the CFR+ type algorithm, which combines the local-regret decomposition of Zinkevich et al. with the RM+ algorithm as the local no-regret algorithm.

The analysis is conducted within the Reward Transformation (RT) framework, which has been proposed to establish last-iterate convergence guarantees through gradual perturbations of the original game. In particular, the RT framework considers a $\gamma$-perturbed version of the original game, in which the strategy space at each information set is restricted to the $\gamma$-bounded-away simplex. The key insight is that if last-iterate convergence can be proven in the perturbed game and then it cab extended to the original game by gradually reducing $\gamma \to 0$.

While the RT framework has previously been used to provide last-iterate convergence properties for OMD/CFR-type algorithms, the main
contribution of the paper is establishing similar results but for the parameter-free CFR+ type of algorithms. The main technical contribution of the paper is establishing that CFR+ admits last-iterate convergence to the equilibrium of the $\gamma$-perturbed game. Leveraging this result and the structure of the RT framework, the authors can then provide a parameter-free algorithm with last-iterate convergence guarantees.

Additionally, the paper includes experimental results demonstrating that the proposed algorithm outperforms prior approaches on large-scale extensive-form games.

**Questions:**

1. Concerning the experimental evaluations, the RTCFR+ outperforms also the convergence rates of time-average algorithms?

2. Is there a reason for using Nash Equilibrium instead of min-max equilibrium? I think the latter would be more appropriate since your work is on zero-sum games.

**Ethical Concerns:**

["NO or VERY MINOR ethics concerns only"]

**Final Justification:**

The author-reviewer discussion has clarified all of my concerns and thus I believe that the results of the paper will be of interest for the neurips audience.

**Limitations:**

Yes

**Quality:**

2

**Strengths And Weaknesses:**

**Strengths**

1. The considered setting is well-motivated: establishing last-iterate convergence guarantees in zero-sum extensive-form games (EFGs) since step-size selection typically plays a critical role in convergence behavior.

2. To prove that CFR+ exhibits last-iterate convergence in perturbed games, the authors first present a non-parameter-free analysis, showing convergence when the step-size exceeds a certain constant. This two-step approach seems fairly interesing. Overall, the paper appears to possess technical depth.

3. The experimental evaluations are very interesting and standard benchmarks are used.

**Weaknesses**

1. The final algorithms RTCFR+ needs parameters $\mu,\eta$ that makes it a bit questionable whether it can be technically considered as parameter-free. That being said, the authors in the experimental evaluations do not fine-tune making the comparison fair. Nevertheless at least on the theoretical level this creates me some doubts on the \textit{parameter-free take-away message}. I believe that probably the take-away should emphasize more on the practical merits of the approach.

2. Although the introduction is clearly written and effectively situates the paper within the existing literature, the main technical sections are difficult to follow. While this is somewhat expected due to the inherently complex and notation-heavy nature of EFGs, there is room to improve the clarity and accessibility of the theoretical exposition.

---

> ### Author Rebuttal · Authors · 2025-07-31
>
> Thank you for your thoughtful comments and the positive evaluation
>
>
>
> **W1: The final algorithms RTCFR+ needs parameters $\mu,\eta$ that makes it a bit questionable whether it can be technically considered as parameter-free. I believe that probably the take-away should emphasize more on the practical merits of the approach.**
>
> **A:**
>
> - We apologize for any confusion regarding the term "parameter-free." In this paper, all references to "parameter-free" pertain exclusively to CFR+ for solving perturbed regularized EFGs. RTCFR+ itself is not a parameter-free algorithm. We will explicitly state this in the limitations section of the final manuscript to ensure clarity.
> - The practical advantage of our approach is that it eliminates the need for parameter tuning when solving perturbed regularized EFGs. This is beneficial as it reduces the cost associated with parameter tuning, making our algorithm more readily applicable to real-world games compared to algorithms that require parameter tuning for solving perturbed regularized EFGs.
>
>
>
> **Q1: Does RTCFR+ outperforms also the convergence rates of time-average algorithms**
>
> **A:** We compare the last-iterate convergence performance of RTCFR+ with the average-iterate performance of CFR+, PCFR+, and DCFR (Table 1). With fine-tuning, RTCFR+ outperforms the average-iterate performance of CFR+, PCFR+, and DCFR in nearly all tested games, except for Liar's Dice (6). Even without fine-tuning, RTCFR+ achieves superior performance to the average-iterate of CFR+, PCFR+, and DCFR in 5 out of the evaluated 9 games (Kuhn Pker, Leduc Poker, Liar's Dice (4),  Liar's Dice (5), and Goofspiel (4)). In addition, as shown in Table 1, even when considering only CFR+, PCFR+, and DCFR, no single algorithm consistently outperforms the other two across all games.
>
> Table 1: Comparison between the last-iterate convergence performance of RTCFR+ with the average-iterate convergence performance of CFR+, PCFR+, and DCFR.
>
> | Game                | RTCFR$^+$ | RTCFR$^+$ (fine-tuned) | CFR$^+$  | PCFR$^+$ | DCFR     |
> | :------------------ | :-------- | :--------------------- | :------- | :------- | :------- |
> | **Kuhn Poker**      | 2.49e-15  | 6.94e-18               | 1.49e-05 | 8.56e-09 | 1.59e-05 |
> | **Leduc Poker**     | 1.97e-13  | 1.97e-13               | 7.08e-06 | 1.56e-05 | 9.72e-06 |
> | **Battleship (3)**  | 1.64e-02  | 8.60e-16               | 2.74e-04 | 1.74e-07 | 1.58e-04 |
> | **Liar's Dice (4)** | 3.19e-16  | 4.86e-17               | 4.28e-06 | 1.99e-07 | 1.90e-06 |
> | **Liar's Dice (5)** | 4.12e-07  | 5.29e-16               | 6.00e-06 | 1.10e-06 | 3.58e-06 |
> | **Liar's Dice (6)** | 6.17e-05  | 8.49e-06               | 2.47e-06 | 4.80e-05 | 2.56e-06 |
> | **Goofspiel (4)**   | 1.05e-08  | 6.77e-17               | 9.57e-06 | 2.88e-08 | 5.31e-07 |
> | **Goofspiel (5)**   | 3.38e-05  | 5.12e-16               | 4.43e-05 | 1.23e-06 | 3.70e-05 |
> | **Goofspiel (6)**   | 2.73e-05  | 1.55e-07               | 4.50e-05 | 6.23e-06 | 3.62e-05 |
>
>
>
> **Q2: Is there a reason for using Nash Equilibrium instead of min-max equilibrium**
>
> **A:** In this paper, we consider two-player zero-sum games, where the min-max equilibrium is the same as NE. Specifically, as shown in [1], “in two-player zero-sum games the concept of equilibrium, which is based on stability, and the concept of minmax, which is based on security levels, coincide.”
>
>
>
> Additional References:
>
> [1] Maschler, Michael, Shmuel Zamir, and Eilon Solan. *Game theory*. Cambridge University Press, 2020.

---

> > ### Comment · Reviewer_kpKA · 2025-08-02
> > **Reviewer kpKA**
> >
> > Thank you very much for your responses. I maintain my score.

---

> > > ### Author Response · Authors · 2025-08-02
> > >
> > > Thank you for your thoughtful review and consideration!

---

### Decision · Program_Chairs · 2025-09-17

**Decision:**

Accept (poster)

**Comment:**

This paper shows that the classical parameter-free RM-based CFR algorithm achieves last-iterate convergence in learning an NE of perturbed regularized EFGs, while previous studies were only able to show convergence for OMD-based algorithms. Given the practical advantage of the RM algorithm and the lack of good theoretical understanding of this simple algorithm, I believe that the results are significant enough and likely to inspire more future studies. Therefore, despite that this is an asymptotic result and that it does not eventually lead to a parameter-free algorithm for solving the original EFGs, I still support acceptance. All reviewers (including mv8m) have converged to supporting acceptance as well, with the expectation that the authors will revise the paper as promised to remove the overstatement on the "parameter-free" property and incorporate additional empirical results.